# WELFARIST FORMULATIONS FOR DIVERSE SIMILARITY SEARCH

**Siddharth Barman, Nirjhar Das & Shivam Gupta**
Indian Institute of Science Bangalore
`{barman,nirjhardas,shivamgupta2}@iisc.ac.in`

**Kirankumar Shiragur**
Microsoft Research India
`kshiragur@microsoft.com`

## ABSTRACT

Nearest Neighbor Search (NNS) is a fundamental problem in data structures with wide-ranging applications, such as web search, recommendation systems, and, more recently, retrieval-augmented generations (RAG). In such recent applications, in addition to the relevance (similarity) of the returned neighbors, diversity among the neighbors is a central requirement. In this paper, we develop principled welfare-based formulations in NNS for realizing diversity across attributes. Our formulations are based on welfare functions—from mathematical economics—that satisfy central diversity (fairness) and relevance (economic efficiency) axioms. With a particular focus on Nash social welfare, we note that our welfare-based formulations provide objective functions that adaptively balance relevance and diversity in a query-dependent manner. Notably, such a balance was not present in the prior constraint-based approach, which forced a fixed level of diversity and optimized for relevance. In addition, our formulation provides a parametric way to control the trade-off between relevance and diversity, providing practitioners with flexibility to tailor search results to task-specific requirements. We develop efficient nearest neighbor algorithms with provable guarantees for the welfare-based objectives. Notably, our algorithm can be applied on top of any standard ANN method (i.e., use standard ANN method as a subroutine) to efficiently find neighbors that approximately maximize our welfare-based objectives. Experimental results demonstrate that our approach is practical and substantially improves diversity while maintaining high relevance of the retrieved neighbors.

## 1 INTRODUCTION

Nearest Neighbor Search (NNS) is a fundamental problem in computer science with wide-ranging applications in diverse domains, including computer vision (Wang et al., 2012), data mining (Camerra et al., 2010), information retrieval (Manning et al., 2008), classification (Fix & Hodges, 1989), and recommendation systems (Dahiya et al., 2021). The relevance of NNS has grown further in recent years with the advent of retrieval-augmented generation (RAG); see, e.g., (Manohar et al., 2024), (Wu et al., 2024), and references therein. Formally, given vectors $P \subset \mathbb{R}^d$ (in ambient dimension $d$) and a query vector $q \in \mathbb{R}^d$, the objective in NNS is to identify a subset $S$ of $k$ (input) vectors from $P$ that are most similar to $q$ under a similarity function $\sigma : \mathbb{R}^d \times \mathbb{R}^d \to \mathbb{R}_+$. That is, NNS corresponds to the optimization problem $\arg\max_{S \subseteq P : |S| = k} \sum_{v \in S} \sigma(q, v)$. Note that, while most prior works in neighbor search express the problem in terms of minimizing distances, we work with the symmetric version of maximizing similarity.[1]

In practice, the input vectors are high dimensional; in many of the above-mentioned applications the ambient dimension $d$ is close to a thousand. This scale and the large cardinality of $P$ make exact NNS computationally expensive, since applications require, for real-time queries $q$, NNS solutions

---

[1]This enables us to directly apply welfare functions.

in time (sub)linear in the number of input vectors $|P|$. To address this challenge, the widely studied framework of Approximate Nearest Neighbor (ANN) search relaxes the requirement of exactness and instead seeks neighbors whose similarities are approximately close to the optimal ones.

ANN search has received substantial attention over the past three decades. Early techniques relied on space-partitioning methods, including Locality-Sensitive Hashing (LSH) (Indyk & Motwani, 1998; Andoni & Indyk, 2008), k-d trees (Arya et al., 1998), and cover trees (Beygelzimer et al., 2006). More recent industry-scale systems adopt clustering-based (Johnson et al., 2017; Baranchuk et al., 2018) and graph-based (Malkov & Yashunin, 2016; Fu et al., 2019; Sugawara et al., 2016; Subramanya et al., 2019) approaches, along with other practically-efficient methods (Sun et al., 2023; Simhadri et al., 2024).

While relevance—measured in terms of a similarity function $\sigma(\cdot, \cdot)$—is a primary objective in NNS, prior work has shown that *diversity* in the retrieved set of vectors is equally important for user experience, fairness, and reducing redundancy (Carbonell & Goldstein, 1998). For instance, in 2019 Google announced a policy update to limit the number of results from a single domain, thereby reducing redundancy (Liaison, 2019). Similarly, Microsoft recently introduced diversity constraints in ad recommendation systems to ensure that advertisements from a single seller do not dominate the results (Anand et al., 2025). Such an adjustment was crucial for improving user experience and promoting fairness for advertisers.

A natural way to formalize diversity in these settings is to associate each input vector with one or more *attributes*. Diversity can then be measured with respect to these attributes. Building on this idea, the current work develops a principled framework for diversity in neighbor search by drawing on the theory of collective welfare from mathematical economics (Moulin, 2004). This perspective enables the design of optimization criteria that balance similarity-based relevance and attribute-based diversity in a theoretically grounded manner.

This formulation is based on welfare functions, $f : \mathbb{R}^c \mapsto \mathbb{R}$, which provide a principled approach to aggregate the utilities of $c \in \mathbb{Z}_+$ agents. Specifically, among $c$ agents with utilities $u_1, \ldots, u_c$, respectively, the collective welfare is $f(u_1, \ldots, u_c)$. The theory of collective welfare develops meaningful welfare functions, $f$s, and among them, Nash social welfare (NSW) is an exemplar that upholds multiple fairness axioms, including symmetry, independence of unconcerned agents, scale invariance and the Pigou-Dalton principle (Moulin, 2004). Nash social welfare is defined by setting $f$ as the geometric mean, $\mathrm{NSW}(u_1, \ldots, u_c) \coloneqq \left( \prod_{\ell=1}^c u_\ell \right)^{1/c}$. The fact that NSW strikes a balance between fairness and economic efficiency is supported by the observation that it sits between egalitarian and utilitarian welfare: the geometric mean is at least as large as the minimum value, $\min_{1 \le \ell \le c} u_\ell$, and it is also at most the arithmetic mean $\frac{1}{c} \sum_{\ell=1}^c u_\ell$ (the AM-GM inequality).

**Our Formulation.** To achieve diversity across attributes in NNS while maintaining relevance of the returned $k$ vectors, our modeling insight is to equate attributes with agents and apply Nash social welfare. In particular, consider a setting where we have $c \in \mathbb{Z}_+$ different attributes (across the input vectors), and let $S$ be any size-$k$ subset of the input vectors $P$. In our model, each included vector $v \in S$, with attribute $\ell \in [c]$, contributes to the utility $u_\ell$ (see Section 2.1), and the NSW induced by $S$ is the geometric mean of these utilities, $u_1, \ldots, u_c$. Our objective is to find a size-$k$ subset, $S^* \subseteq P$, of input vectors with as large NSW as possible (Definition 1).[2]

Prior work (Anand et al., 2025) imposed constraints for achieving diversity in NNS. These constraints enforced that, for each $\ell \in [c]$, at most $k'$ of the returned vectors can have attribute $\ell$. Such hard constraints rely on a fixed ad hoc quota parameter $k'$ and may fail to adapt to the intent expressed in the query. By contrast, our NSW-based approach balances relevance and diversity in a query-dependent manner. For example, in apparel search, if the query is "blue shirt," then a constraint on the color attribute 'blue' (i.e., when $\ell$ stands for 'blue') would limit the relevance by excluding valid vectors. NSW, however, for the "blue shirt" query, is free to select all the $k$ vectors with attribute 'blue' upholding relevance; see Figure 1 for supporting empirical results. Simultaneously, if the query is just "shirts," then NSW criterion is inclined to select vectors with different color attributes. These features of NSW are further substantiated quantitatively by the stylized instances given in Examples 1 and 2 (Appendix B)

---

[2]The following instantiation highlights the applicability of our model: In an advertising context with $c$ sellers, each selected ad $v \in S$ of a seller $\ell \in [c]$ contributes to $\ell$'s exposure (utility) $u_\ell$.

Figure 1: Neighbor search results ($k = 9$) on the Amazon dataset. From left: **First** and **Second** images - ANN and Nash-based results for query "shirts", respectively. **Third** and **Fourth** images - ANN and Nash-based results for query "blue shirt", respectively. Note that the Nash-based method selects diverse colors for the query "shirts" but conforms to the blue color for the query "blue shirt".

Our welfarist formulation extends further to control the trade-off between relevance and diversity. Specifically, we also consider $p$-mean welfare. Formally, for exponent parameter $p \in (-\infty, 1]$, the $p$th mean $M_p(\cdot)$, of $c$ utilities $u_1, \ldots, u_c \in \mathbb{R}_+$, is defined as $M_p(u_1, \ldots, u_c) := \left( \frac{1}{c} \sum_{\ell=1}^{c} u_\ell^p \right)^{1/p}$. The $p$-mean welfare, $M_p(\cdot)$, captures a range of objectives with different values of $p$: it corresponds to the utilitarian welfare (arithmetic mean) when $p = 1$, the NSW (geometric mean) with $p \to 0$, and the egalitarian welfare when $p \to -\infty$. Notably, setting $p = 1$, we get back the standard nearest neighbor objective (relinquishing diversity). At the other extreme, $p \to -\infty$ aims to find as attribute-diverse a set of $k$ vectors as possible (while paying scarce attention to relevance).

We study, both theoretically and experimentally, two diversity settings: (i) single-attribute and (ii) multi-attribute. In the single-attribute setting, each input vector $v \in P$ is associated with exactly one attribute $\ell \in [c]$.[3] In the more general multi-attribute setting, each input vector $v \in P$ can have more than one attribute.

The constraint-based formulation for diversity considered in Anand et al. (2025) primarily addresses single-attribute setting. In fact, generalizing such constraints to the multi-attributes leads to a formulation wherein it is NP-hard even to determine whether there exist $k$ vectors that satisfy the constraints.[4] The NSW formulation does not run into such a barrier.

**Our Contributions**

• The NSW formulation for diversity, in both single-attribute and multi-attribute settings, is a key contribution of the work (Definition 1). Another contribution is the generalization to $p$-mean welfare.

• We also develop efficient algorithms, with provable guarantees, for the NSW and $p$-mean welfare formulations. For the single-attribute setting, we develop an efficient greedy algorithm for finding $k$ vectors that optimize NSW among the $c$ attributes (Theorem 1). In addition, this algorithm can be provably combined with any sublinear ANN method (as a subroutine) to find near-optimal solutions for the Nash objective in sublinear time (Corollary 2).

• For the multi-attribute setting, we show that the NSW problem is NP-hard (Theorem 3). Also, we develop a polynomial-time 0.63-approximation algorithm for the logarithm of NSW (Theorem 4).

• We complement our theoretical results with experiments on both real-world and semi-synthetic datasets. These experiments demonstrate that the NSW objective effectively captures the trade-off between diversity and relevance in a query-dependent manner. We further analyze the behavior of the $p$-mean welfare objective across different values of $p \in (-\infty, 1]$, observing that it interpolates smoothly between prioritizing for diversity, when $p$ is small, and focusing on relevance, when $p$ is large. Finally, we benchmark the solution quality and running times of various algorithms for solving the NSW and $p$-mean formulations proposed in this work.

## 2  PROBLEM FORMULATION AND MAIN RESULTS

We are interested in neighbor search algorithms that not only achieve a high relevance, but also find a diverse set of vectors for each query. To quantify diversity we work with a model wherein each

---

[3]For instance, in the display-advertisement setup, each advertisement $v$ belongs to exactly one seller $\ell$.

[4]That is, it would be computationally hard to find any size-$k$ constraint-feasible subset $S$, let alone an optimal one. This hardness result follows via a reduction from the Maximum Independent Set problem.

input vector $v \in P$ is assigned one or more attributes from the set $[c] = \{1, 2, \ldots, c\}$. In particular, we write $\mathrm{atb}(v) \subseteq [c]$ to denote the attributes assigned to vector $v \in P$. Also, let $D_\ell \subseteq P$ denote the subset of vectors that are assigned attribute $\ell \in [c]$, i.e., $D_\ell := \{v \in P \mid \ell \in \mathrm{atb}(v)\}$.

## 2.1 OUR RESULTS

An insight of this work is to equate these $c$ attributes with $c$ distinct agents. Here, the output of a neighbor search algorithm—i.e., the selected subset $S \subseteq P$—induces utility among these agents. With this perspective, we define the Nash Nearest Neighbor Search problem (NaNNS) below. This novel formulation for diversity is a key contribution of this work. For any query $q \in \mathbb{R}^d$ and subset $S \subseteq P$, we define utility $u_\ell(S) := \sum_{v \in S \cap D_\ell} \sigma(q, v)$, for each $\ell \in [c]$. That is, $u_\ell(S)$ is equal to the cumulative similarity between $q$ and the vectors in $S$ that belong to $D_\ell$. Equivalently, $u_\ell(S)$ is the cumulative similarity of the vectors in $S$ that have attribute $\ell$.[5]

We employ Nash social welfare to identify size-$k$ subsets $S$ that are both relevant (with respect to similarity) and support diversity among the $c$ attribute classes. The Nash social welfare among $c$ agents is defined as the geometric mean of the agents' utilities. Specifically, in the above-mentioned utility model and with a smoothening parameter $\eta > 0$, the Nash social welfare (NSW) induced by any subset $S \subseteq P$ among the $c$ attributes is defined as $\mathrm{NSW}(S) := \left(\prod_{\ell=1}^{c} (u_\ell(S) + \eta)\right)^{1/c}$. Throughout, $\eta > 0$ will be a fixed smoothing constant that ensures that NSW remains nonzero.

**Definition 1** (NaNNS). Nash nearest neighbor search (NaNNS) corresponds to the optimization problem: $\arg\max_{S \subseteq P : |S|=k} \mathrm{NSW}(S)$, or, equivalently,

$$\arg\max_{S \subseteq P : |S|=k} \log \mathrm{NSW}(S) \tag{1}$$

Here, we have $\log \mathrm{NSW}(S) = \frac{1}{c} \sum_{\ell \in [c]} \log(u_\ell(S) + \eta)$.

To further appreciate the welfarist approach, note that one recovers the standard nearest neighbor problem, NNS, in the single-attribute setting, if—instead of the geometric mean—we maximize the arithmetic mean. That is, maximizing the utilitarian social welfare gives us $\max_{S \subseteq P : |S|=k} \frac{1}{c} \sum_{\ell=1}^{c} u_\ell(S) = \max_{S \subseteq P : |S|=k} \frac{1}{c} \sum_{v \in S} \sigma(q, v)$.

As stated in the introduction, depending on the query and the problem instance, solutions obtained via NaNNS can adjust between the ones obtained through standard NNS and those obtained via hard constraints. This feature is illustrated in the stylized examples 1 and 2 stated in Appendix B.

With the above-mentioned utility model for the $c$ attributes, we also identify an extended formulation based on generalized $p$-means. Specifically, for exponent parameter $p \in (-\infty, 1]$, the $p$th mean $M_p(\cdot)$, of $c$ nonnegative numbers $w_1, w_2, \ldots, w_c \in \mathbb{R}_+$, is defined as $M_p(w_1, \ldots, w_c) := \left(\frac{1}{c} \sum_{\ell=1}^{c} w_\ell^p\right)^{1/p}$. Note that $M_1(w_1, \ldots, w_c)$ is the arithmetic mean $\frac{1}{c} \sum_{\ell=1}^{c} w_\ell$. Here, when $p \to 0$, we obtain the geometric mean (Nash social welfare): $M_0(w_1, \ldots, w_c) = \left(\prod_{\ell=1}^{c} w_\ell\right)^{1/c}$. Further, $p \to -\infty$ gives us egalitarian welfare, $M_{-\infty}(w_1, \ldots, w_\ell) = \min_{1 \le \ell \le c} w_\ell$.

Hence, generalizing both NNS and NaNNS, we have the $p$-mean nearest neighbor search ($p$-NNS) problem defined, for exponent parameters $p \in (-\infty, 1]$, as follows:

$$\max_{S \subseteq P : |S|=k} M_p\big(u_1(S), \ldots, u_c(S)\big)$$

.

**Diversity in Single- and Multi-Attribute Settings.** The current work addresses two diversity settings: the single-attribute setup and, the more general, the multi-attribute one. The single-attribute setting refers to case wherein $|\mathrm{atb}(v)| = 1$ for each input vector $v \in P$ and, hence, the attribute classes $D_\ell$s are pairwise disjoint. In the more general multi-attribute setting, we have $|\mathrm{atb}(v)| \ge 1$; here, the sets $D_\ell$-s intersect.[6] Notably, the NaNNS seamlessly applies to both these settings.

---

[5] Note that in the above-mentioned display-advertising example, $u_\ell(\cdot)$ is the cumulative similarity between the (search) query and the selected advertisements that are from seller $\ell$.

[6] For a motivating instantiation for multi-attributes, note that, in the apparel-search context, it is possible for a product (input vector) $v$ to have multiple attributes based on $v$'s seller and its color(s).

**Algorithmic Results for Single-Attribute NaNNS and $p$-NNS.** In addition to introducing the NaNNS and $p$-NNS formulations for capturing diversity, we develop algorithmic results for these problems, thereby demonstrating the practicality of our approach in neighbor search. In particular, in the single-attribute setting, we show that both NaNNS and $p$-NNS admit efficient algorithms.

**Theorem 1.** *In the single-attribute setting, given any query $q \in \mathbb{R}^d$ and an (exact) oracle ENN for $k$ most similar vectors from any set, Algorithm 1 (Nash-ANN) returns an optimal solution for NaNNS, i.e., it returns a size-$k$ subset $\text{ALG} \subseteq P$ that satisfies $\text{ALG} \in \arg\max_{S \subseteq P: |S|=k} \text{NSW}(S)$. Furthermore, the algorithm runs in time $O(kc) + \sum_{\ell=1}^{c} ENN(D_\ell, q)$, where $ENN(D_\ell, q)$ is the time required by the exact oracle to find $k$ most similar vectors to $q$ in $D_\ell$.*

Further, to establish the practicality of our formulations, we present an approximate algorithm for NaNNS that leverages any standard ANN algorithm as an oracle (subroutine), i.e., works with any $\alpha$-approximate ANN oracle ($\alpha \in (0, 1)$) which returns a subset $S$ containing $k$ vectors satisfying $\sigma(q, v_{(i)}) \geq \alpha \, \sigma(q, v^*_{(i)})$, for all $i \in [k]$, where $v_{(i)}$ and $v^*_{(i)}$ are the $i$-th most similar vectors to $q$ in $S$ and $P$, respectively. Formally,

**Corollary 2.** *In the single-attribute setting, given any query $q \in \mathbb{R}^d$ and an $\alpha$-approximate oracle ANN for $k$ most similar vectors from any set, Algorithm 1 (Nash-ANN) returns an $\alpha$-approximate solution for NaNNS, i.e., it returns a size-$k$ subset $\text{ALG} \subseteq P$ with $\text{NSW}(\text{ALG}) \geq \alpha \max_{S \subseteq P: |S|=k} \text{NSW}(S)$. The algorithm runs in time $O(kc) + \sum_{\ell=1}^{c} ANN(D_\ell, q)$, with $ANN(D_\ell, q)$ denoting the time required by the oracle to find $k$ similar vectors to $q$ in $D_\ell$.*

Furthermore, both Theorem 1 and Corollary 2 generalize to $p$-NNS problem with slight modification in Algorithm 1. Specifically, there exists exact, efficient algorithm (Algorithm 3) for the $p$-NNS problem (Theorem 11 and Corollary 12). Due to space constraints, the algorithm and the analysis for $p$-NNS are deferred to Appendix E.

**Algorithmic Results for Multi-Attribute NaNNS.** Next, we address the multi-attribute setting. While the optimization problem (1) in the single attribute setting can be solved efficiently, the problem is NP-Hard in the the multi-attribute setup (see Appendix C for the proof).

**Theorem 3.** *In the multi-attribute setting, with parameter $\eta = 1$, NaNNS is NP-hard.*

Complementing this hardness result, we show that, considering the logarithm of the objective, NaNNS in the multi-attribute setting admits a polynomial-time $\left(1 - \frac{1}{e}\right)$-approximation algorithm. This result in established in Appendix D.

**Theorem 4.** *In the multi-attribute setting, there exists a polynomial-time algorithm (Algorithm 2) that, given any query $q \in \mathbb{R}^d$, finds a size-$k$ subset $\text{ALG} \subseteq P$ with $\log \text{NSW}(\text{ALG}) \geq \left(1 - \frac{1}{e}\right) \log \text{NSW}(\text{OPT})$; here, OPT denotes an optimal solution of (1).*

**Experimental Validation of our Formulation and Algorithms.** We complement our theoretical results with several experiments on real-world datasets. Our findings highlight that the Nash-based formulation strikes a balance between diversity and relevance.

## 3 ALGORITHM FOR NANNS

This section provides our exact, efficient algorithm (Algorithm 1) for NaNNS in the single-attribute setting. The algorithm has two parts: a preprocessing step and a greedy, iterative selection.

Recall that in the single-attribute setting, the input vectors $P$ are partitioned into subsets $D_1, \ldots, D_c$, where $D_\ell$ denotes the subset of input vectors with attribute $\ell \in [c]$. In the preprocessing step, for each attribute $\ell \in [c]$, we populate $k$ vectors from within $D_\ell$ that are most similar to the given query $q \in \mathbb{R}^d$. Such a size-$k$ subset, for each $\ell \in [c]$, can be obtained by executing any nearest neighbor search algorithm within $D_\ell$ and with respect to query $q$. Alternatively, we can execute any standard ANN algorithm as a subroutine and find sufficiently good approximations for the $k$ nearest neighbors (of $q$) within each $D_\ell$.

Write $\widehat{D}_\ell \subseteq D_\ell$ to denote the $k$—exact or approximate—nearest neighbors of $q \in \mathbb{R}^d$ in $D_\ell$. We note that our algorithm is robust to the choice of the search algorithm (subroutine) used for finding $\widehat{D}_\ell$s: If $\widehat{D}_\ell$s are exact nearest neighbors, then Algorithm 1 optimally solves NaNNS in the single-attribute

---

**Algorithm 1** `Nash-ANN`: Algorithm for NaNNS in the single-attribute setting

---

**Require:** Query $q \in \mathbb{R}^d$ and, for each attribute $\ell \in [c]$, the set of input vectors $D_\ell \subset \mathbb{R}^d$.

1: For each $\ell \in [c]$, fetch $\widehat{D}_\ell$, the $k$ (exact or approximate) nearest neighbors of $q \in \mathbb{R}^d$ from $D_\ell$.
2: For every $\ell \in [c]$ and each index $i \in [k]$, let $v_{(i)}^\ell$ denote the $i$th most similar vector to $q$ in $\widehat{D}_\ell$.
3: Initialize subset $\text{ALG} = \emptyset$, along with count $k_\ell = 0$ and utility $w_\ell = 0$, for each $\ell \in [c]$.
4: **while** $|\text{ALG}| < k$ **do**
5:    Let $a = \underset{\ell \in [c]}{\arg\max} \left( \log \left( w_\ell + \eta + \sigma(q, v_{(k_\ell+1)}^\ell) \right) - \log(w_\ell + \eta) \right)$. {Ties broken arbitrarily.}
6:    Update $\text{ALG} \leftarrow \text{ALG} \cup \left\{ v_{(k_a+1)}^a \right\}$, along with $w_a \leftarrow w_a + \sigma(q, v_{(k_a+1)}^a)$ and $k_a \leftarrow k_a + 1$.
7: **Return** $\text{ALG}$.

---

setting (Theorem 1). Otherwise, if $\widehat{D}_\ell$s are obtained via an ANN algorithm with approximation guarantee $\alpha \in (0, 1)$, then Algorithm 1 achieves an approximation ratio of $\alpha$ (Corollary 2).

The algorithm then considers the vectors with each $\widehat{D}_\ell$ in decreasing order of their similarity with $q$. Confining to this order, the algorithm populates the $k$ desired vectors iteratively. In each iteration, the algorithm greedily selects a new vector based on the marginal increase in $\log \text{NSW}(\cdot)$; see Lines 5-6 in Algorithm 1. Theorem 1 and Corollary 2 provide our main results for Algorithm 1; Proof of Theorem 1 is presented below while the proof of Corollary 2 is deferred to Appendix A.1.

*Proof of Theorem 1.* The runtime of Algorithm 1 can be established by noting that Line 1 requires $\sum_{\ell=1}^c \text{ENN}(D_\ell, q)$ time to populate the subsets $\widehat{D}_\ell$s, and the while-loop (Lines 4-6) iterates $k$ times and each iteration (specifically, Line 5) runs in $O(c)$ time. Hence, as stated, the time complexity of the algorithm is $O(kc) + \sum_{\ell=1}^c \text{ENN}(D_\ell, q)$.

Next, we prove the optimality of the returned set $\text{ALG}$. Let $\text{OPT} \in \arg\max_{S \subseteq P:|S|=k} \text{NSW}(S)$ be an optimal solution with attribute counts $|\text{OPT} \cap D_\ell|$ as close to $|\text{ALG} \cap D_\ell|$ as possible. That is, among the optimal solutions, it is one that minimizes $\sum_{\ell=1}^c |k_\ell^* - k_\ell|$, where $k_\ell^* = |\text{OPT} \cap D_\ell|$ and $k_\ell = |\text{ALG} \cap D_\ell|$, for each $\ell \in [c]$. We will prove that $\text{OPT}$ satisfies $k_\ell^* = k_\ell$ for each $\ell \in [c]$. This guarantee along with Lemma 6 imply that, as desired, $\text{ALG}$ is a Nash optimal solution.

Assume, towards a contradiction, that $k_\ell^* \neq k_\ell$ for some $\ell \in [c]$. Since $|\text{OPT}| = |\text{ALG}| = k$, there exist attributes $x, y \in [c]$ with the property that $k_x^* < k_x$ and $k_y^* > k_y$. For a given attribute $\ell \in [c]$, define the logarithm of cumulative similarity upto the $i^{\text{th}}$ most similar vector as $F_\ell(i) := \log \left( \sum_{j=1}^i \sigma(q, v_{(j)}^\ell) + \eta \right)$, where $v_{(j)}^\ell$ is defined in Line 2 of Algorithm 1.

Next, note that for any attribute $\ell \in [c]$, if Algorithm 1, at any point during its execution, has included $k_\ell'$ vectors of attribute $\ell$ in $\text{ALG}$, then at that point the maintained utility $w_\ell = \sum_{j=1}^{k_\ell'} \sigma(q, v_{(j)}^\ell)$. Hence, at the beginning of any iteration of the algorithm, if the $k_\ell'$ denotes the number of selected vectors of each attribute $\ell \in [c]$, then the marginals considered in Line 5 are $F_\ell(k_\ell' + 1) - F_\ell(k_\ell')$. These observations and the selection criterion in Line 5 of the algorithm give us the following inequality for the counts $k_x = |\text{ALG} \cap D_x|$ and $k_y = |\text{ALG} \cap D_y|$ of the returned solution $\text{ALG}$:

$$F_x(k_x) - F_x(k_x - 1) \geq F_y(k_y + 1) - F_y(k_y) \tag{2}$$

Specifically, equation (2) follows by considering the iteration in which $k_x^{\text{th}}$ (last) vector of attribute $x$ was selected by the algorithm. Before that iteration the algorithm had selected $(k_x - 1)$ vectors of attribute $x$, and let $k_y'$ denote the number of vectors with attribute $y$ that have been selected till that point. Note that $k_y' \leq k_y$. The fact that the $k_x^{\text{th}}$ vector was (greedily) selected in Line 5, instead of including an additional vector of attribute $y$, gives $F_x(k_x) - F_x(k_x - 1) \geq F_y(k_y' + 1) - F_y(k_y') \geq F_y(k_y + 1) - F_y(k_y)$; here, the last inequality follows from Lemma 5. Therefore we have,

$$F_x(k_x^* + 1) - F_x(k_x^*) \overset{\text{(i)}}{\geq} F_x(k_x) - F_x(k_x - 1) \overset{\text{(ii)}}{\geq} F_y(k_y + 1) - F_y(k_y) \overset{\text{(iii)}}{\geq} F_y(k_y^*) - F_y(k_y^* - 1) \tag{3}$$

Here, inequality (i) follows from $k_x^* < k_x$ and Lemma 5, inequality (ii), from equation 2, and inequality (iii), from $k_y^* > k_y$ and Lemma 5.

Next, observe that the definition of $\widehat{D}_\ell$ ensures that $v_{(i)}^\ell$ is in fact the $i^{\text{th}}$ most similar (to $q$) vector among the ones that have attribute $\ell$, i.e., $i^{\text{th}}$ most similar in all of $D_\ell$. Since OPT is an optimal solution, the $k_\ell^* = |\text{OPT} \cap D_\ell|$ vectors of attribute $\ell$ in OPT are the most similar $k_\ell^*$ vectors from $D_\ell$. That is, $\text{OPT} \cap D_\ell = \{v_{(1)}^\ell, \ldots, v_{(k_\ell^*)}^\ell\}$, for each $\ell \in [c]$. This observation and the definition of $F_\ell(\cdot)$ imply that the logarithm of OPT's NSW satisfies $\log \text{NSW}(\text{OPT}) = \frac{1}{c} \sum_{\ell=1}^c F_\ell(k_\ell^*)$. Now, consider a subset of vectors $S$ obtained from OPT by including vector $v_{(k_x^*+1)}^x$ and removing $v_{(k_y^*)}^y$, i.e., $S = \left( \text{OPT} \cup \left\{ v_{(k_x^*+1)}^x \right\} \right) \setminus \left\{ v_{(k_y^*)}^y \right\}$. Note that

$$\log \text{NSW}(S) - \log \text{NSW}(\text{OPT}) = \frac{1}{c}\Big( F_x(k_x^* + 1) - F_x(k_x^*) \Big) + \frac{1}{c}\Big( F_y(k_y^* - 1) - F_y(k_y^*) \Big) \geq 0,$$

where the last inequality is via eqn. (3). Hence, we have shown that $\text{NSW}(S) \geq \text{NSW}(\text{OPT})$. Given that OPT is a Nash optimal solution, the last inequality must hold with an equality, $\text{NSW}(S) = \text{NSW}(\text{OPT})$, i.e., $S$ is an optimal solution as well. This, however, contradicts the choice of OPT as an optimal solution that minimizes $\sum_{\ell=1}^c |k_\ell^* - k_\ell|$; note that $\sum_{\ell=1}^c \left| \widehat{k}_\ell - k_\ell \right| < \sum_{\ell=1}^c |k_\ell^* - k_\ell|$, where $\widehat{k}_\ell := |S \cap D_\ell|$.

Therefore, by way of contradiction, we obtain that $|\text{OPT} \cap D_\ell| = |\text{ALG} \cap D_\ell|$ for each $\ell \in [c]$. As mentioned previously, this guarantee along with Lemma 6 imply that ALG is a Nash optimal solution. This completes the proof of the theorem. $\qquad\square$

## 4 EXPERIMENTAL EVALUATIONS

In this section, we validate the welfare-based formulations and the performance of our proposed algorithms against existing methods on a variety of real and semi-synthetic datasets. We perform three different kinds of experiments. In the first set of experiments (Figure 2, Top row), we compare `Nash-ANN` (Algorithm 1) with prior work on hard-constraint based diversity (Anand et al., 2025). Here, we show that `Nash-ANN` strikes a balance between relevance and diversity both in the single- and multi-attribute settings. In the second set of experiments (Figure 2, Bottom row), we study the effect of varying the exponent parameter $p$ in the $p$-NNS objective on relevance and diversity, in both single- and multi-attribute settings. In the final set of experiments, we compare our algorithm, `Nash-ANN` (with provable guarantees), and a heuristic that improves the runtime of `Nash-ANN`. The heuristic directly utilizes a standard `ANN` algorithm to fetch a sufficiently large candidate set of vectors (irrespective of their attributes). Then, it applies the Nash (or $p$-mean) selection only within this set. Due to space constraints, we defer the details of the third set of experiments to Appendix F.5. Additional plots for the first two experiments appear in Appendices F.3 and F.4.

Below, we outline the metrics, experimental setup, datasets, and algorithms used in our experiments.

**Relevance and Diversity Metrics.** To quantify relevance of an algorithm we consider the ratio of the sum of the similarity scores. Formally, for a query $q$, if $A$ is the set of neighbors returned by an algorithm and $O$ is the set of $k$ nearest neighbors, then the approximation ratio achieved by the algorithm is given by the ratio $\left( \sum_{v \in A} \sigma(v, q) \right) / \left( \sum_{v \in O} \sigma(v, q) \right)$. Hence, this ratio lies in $[0, 1]$, and a higher approximation ratio indicates a more relevant solution. We also report results in terms of recall, which is another metric for relevance; see Appendix F.1 for further details.

To quantify diversity, we use entropy that measures how uniformly an algorithm distributes its selected vectors across the attributes. Entropy of a size-$k$ subset $S \subseteq P$ in the single-attribute setting is given by the quantity $\sum_{\ell \in [c]: p_\ell > 0} -p_\ell \log(p_\ell)$ where $p_\ell = \frac{|S \cap D_\ell|}{|S|}$. Note that a higher entropy value indicates greater diversity. Moreover, it is not hard to see that the highest possible value of entropy is $\log(k)$ (achieved when $S$ contains at most 1 vector from each attribute). A higher value of entropy indicates more attribute-level diversity in the algorithm's output. We also experimentally validate the findings under other diversity metrics, namely the inverse Simpson index and distinct attribute counts; see Appendix F.1.

Note that the diversity metrics defined above are for the single-attribute setting. In the multi-attribute setting, in our experiments, we focus on settings where the attribute set $[c]$ is partitioned into $m$ sets $\{C_i\}_{i=1}^m$ (i.e., $[c] = \sqcup_{i=1}^m C_i$) and every input vector $v \in P$ is associated with exactly one attribute from each $C_i$. In particular, $|\text{atb}(v)| = m$ and $|\text{atb}(v) \cap C_i| = 1$ for each $1 \leq i \leq m$. We call each

Table 1: Summary of considered datasets. For synthetic attributes, we use two strategies: clustering-based (suffixed by `Clus`) and distribution-based (suffixed by `Prob`), see Appendix F.2 for details.

| Dataset | # Input Vectors | # Query Vectors | Dimension | Attributes |
|---|---|---|---|---|
| `Amazon` | $92,092$ | $8,956$ | $768$ | product color |
| `ArXiv` | $200,000$ | $50,000$ | $1536$ | year, paper category |
| `Sift1m` | $1,000,000$ | $10,000$ | $128$ | synthetic |
| `Deep1b` | $9,990,000$ | $10,000$ | $96$ | synthetic |

$C_i$ an attribute class. To measure diversity in the multi-attribute setting, we consider the aforementioned diversity metrics like entropy and inverse Simpson index restricted to an attribute class $C_i$. For instance, the entropy a set $S \subseteq P$ restricted to a particular $C_i$ is given by $\sum_{\ell \in C_i : p_\ell > 0} -p_\ell \log(p_\ell)$, where $p_\ell = \frac{|S \cap D_\ell|}{|S|}$.

**Experimental Setup and Datasets.** We report results on both semi-synthetic and real-world datasets consistent with prior works (Anand et al., 2025) and are summarized in Table 1 and detailed in Appendix F.2.

**Algorithms.** Next, we describe the algorithms we executed.

1. **ANN**: This is the standard ANN algorithm that aims to maximize the similarity of the retrieved vectors to the given query without any diversity considerations. In our experiments, we use the graph based DiskANN method of Subramanya et al. (2019) as the standard ANN algorithm. We instantiate DiskANN with candidate list size $L = 2000$ and the maximum graph degree as $128$.[7] Here, we also set the pruning factor at $1.3$, consistent with the existing recommendation in Anand et al. (2025).

2. **Div-ANN**: This refers to the algorithm of Anand et al. (2025) that solves the hard- constraint-based formulation for diversity in the single-attribute setting. Recall that Anand et al. (2025) aims to maximize the similarity of the retrieved vectors to the given query subject to the constraint that no more than $k'$ vectors in the retrieved set should have the same attribute. Note that the smaller the value of $k'$, the more diverse the retrieved set of vectors. Moreover, $k'$ has to be provided as an input to this algorithm. In our experiments, we set $k' \in \{1, 2, 5\}$ when $k = 10$, and $k' \in \{1, 2, 5, 10\}$ when $k = 50$.

3. **Nash-ANN and $p$-mean-ANN**: `Nash-ANN` refers to Algorithm 1 and $p$-mean-ANN refers to Algorithm 3 (stated in Appendix E). Recall that Algorithm 1 and Algorithm 3 optimally solve the NaNNS and the $p$-NNS problems, respectively, in the single-attribute setting given access to an exact nearest neighbor search oracle (Theorems 1 and 11). Further note that the $p$-mean welfare function, $M_p(\cdot)$, reduces to the Nash social welfare (geometric mean) when the exponent parameter $p \to 0^+$. For readability and at required places, we will write $p = 0$ to denote `Nash-ANN`. We conduct experiments with varying values of $p \in \{-10, -1, -0.5, 0, 0.5, 1\}$.

4. **Multi Nash-ANN and Multi Div-ANN**: In the multi-attribute setting, there are no prior methods to address diversity. Hence, for comparisons, we first fetch $L = 10000$ candidate vectors from $P$ for each query $q$, using the standard `ANN` method, and then apply the following algorithms on the candidate vectors: (i) our algorithm for the NaNNS problem in the multi-attribute setting (Algorithm 2), which we term as `Multi Nash-ANN`, and (ii) an adaptation of the algorithm of Anand et al. (2025), referred hereon as `Multi Div-ANN`, which greedily selects the most similar vectors to the query subject to the constraint that there are no more than $k'$ vectors from each attribute.[8] We compare `Multi Nash-ANN` ($p = 0$) against `Multi Div-ANN` under different choices of $k'$.

5. **Multi $p$-mean-ANN**: In the multi-attribute setting, we also implement an analogue of `Multi Nash-ANN` that (in lieu of NSW) focuses on the $p$-mean welfare, $M_p$. The objective in these experiments is to understand the impact of varying the parameter $p$ and the resulting tradeoff between relevance and diversity. Here, for each given query, we first fetch a set of $L = 10000$ candidate vectors using `ANN`. Then, we populate a set of $k$ vectors by executing a marginal-gains greedy

---

[7]Both these choices are sufficiently larger than standard values, $L = 200$ and maximum graph degree $64$.

[8]Note that one vector can have multiple attributes, hence contributing to the constraint of multiple attributes. Therefore, the issue of identifying an appropriate $k'$ is exacerbated on moving to the multi-attribute setting.

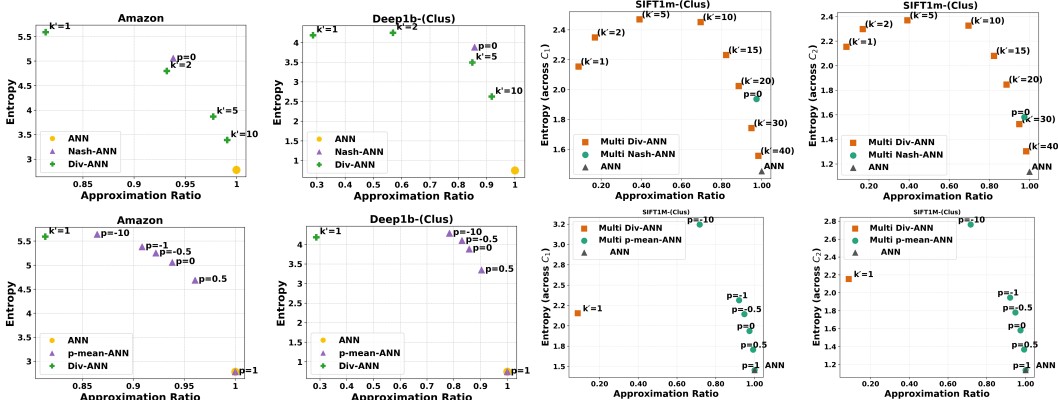

Figure 2: **Top Row: Columns 1 and 2** - Comparison of approximation ratio versus entropy trade-offs between `Nash-ANN`, and `Div-ANN` with varying $k'$, for $k = 50$ on `Amazon` and `Deep1b-(Clus)` datasets in the single-attribute setting. **Columns 3 and 4** - Comparison of approximation ratio versus entropy trade-offs (across attribute classes $C_1$ and $C_2$) between `Multi Nash-ANN`, and `Multi Div-ANN` with varying $k'$ on `Sift1m-(Clus)` dataset with $k = 50$ in the multi-attribute setting. . **Bottom Row: Columns 1 and 2** - Approximation ratio versus entropy trade-offs for $p$-mean-ANN at various $p$ values, for $k = 50$ on `Amazon` and `Deep1b-(Clus)` datasets in the single-attribute setting. **Columns 3 and 4** - Approximation ratio versus entropy trade-offs (across attribute classes $C_1$ and $C_2$) for `Multi` $p$-mean-ANN with varying $p$ on `Sift1m-(Clus)` dataset with $k$=50 in the multi-attribute setting.

method over the $L$ candidate vectors. In particular, we iterate $k$ times, and in each iteration, select a new candidate vector that: (i) for $p \in (0,1]$, yields the maximum increase in $M_p(\cdot)^p$, or (ii) for $p < 0$, leads to the maximum decrease in $M_p(\cdot)^p$.

## 4.1 Results: Balancing Relevance and Diversity

**Single-attribute setting.** We first compare, in the single-attribute setting, the performance of our algorithm, `Nash-ANN`, with `ANN` and `Div-ANN` (with different values of $k'$). The results for the `Amazon` and `Deep1b-(Clus)` datasets with $k = 50$ are shown in Figure 2 (top row, columns one and two). Here, `ANN` finds the most relevant set of neighbors (approximation ratio close to 1), albeit with the lowest entropy (diversity). Moreover, as can be seen in the plots, the most diverse (highest entropy) solution is obtained when we set, in `Div-ANN`, $k' = 1$; this restricts each $\ell \in [c]$ to contribute at most one vector in the output of `Div-ANN`. Also, note that one can increase the approximation ratio (i.e., increase relevance) of `Div-ANN` while incurring a loss in entropy (diversity), by increasing the value of the constraint parameter $k'$. However, selecting a 'right' value for $k'$ is non-obvious, since this choice needs to be tailored to the dataset and, even within it, to queries (recall "blue shirt" query in Figure 1).

By contrast, `Nash-ANN` does not require such ad hoc adjustments and, by design, finds a balance between relevance and diversity. Indeed, as can be seen in Figure 2 (top row), `Nash-ANN` maintains an approximation ratio close to 1 while achieving diversity similar to `Div-ANN` with $k' = 1$. Moreover, `Nash-ANN` Pareto dominates `Div-ANN` with $k' = 2$ for `Amazon` dataset and $k' = 5$ for `Deep1b-(Clus)` dataset on the fronts of approximation ratio and entropy. The results for other datasets and metrics follow similar trends and are given in Appendix F.3.

**Multi-attribute setting.** In the multi-attribute setting, we report results for `Multi Nash-ANN` and `Multi Div-ANN` on the `Sift1m-(Clus)` dataset (Figure 2, Top row–columns 3 and 4) for $k = 50$ and $c = 80$. These eighty attributes are partitioned into four sets, $\{C_i\}_{i=1}^4$, with each set of size $|C_i| = 20$, i.e., $[c] = \cup_{i=1}^4 C_i$. Further, each input vector $v$ is associated with four attributes ($|\text{atb}(v)| = 4$), one from each $C_i$; see Appendix F.2 for further details. Here, to quantify diversity, we separately consider for each $i \in [4]$, the entropy across attributes within a $C_i$. In Figure 2 (Top row–columns 3 and 4), we compare the approximation ratio versus entropy trade-offs of `Multi Nash-ANN` against `Multi Div-ANN` with varying $k'$. Here we show the results for attribute classes $C_1$ (column 1) and $C_2$ (column 2) whereas the results for $C_3$ and $C_4$ are given in

Figure 35. We observe that `Multi Nash-ANN` maintains a high approximation ratio (relevance) while simultaneously achieving significantly higher entropy (higher diversity) than `ANN`. By contrast, in the constraint-based method `Multi Div-ANN`, low values of $k'$ lead to a notable drop in the approximation ratio, whereas increasing $k'$ reduces entropy. For example, for $k'$ below 15, one obtains approximation ratio less than 0.8, and to reach an approximation ratio comparable to `Multi Nash-ANN`, one needs $k'$ as high as 30. Additional results for the `ArXiv` dataset in the multi-attribute setting are provided in Appendix F.4, and they exhibit trends similar to the ones in Figure 2. These findings demonstrate that `Multi Nash-ANN` achieves a balance between relevance and diversity. In summary

*Across datasets, and in both single- and multi-attribute settings, the Nash formulation consistently improves entropy (diversity) over `ANN`, while maintaining an approximation ratio (relevance) of roughly above 0.9. By contrast, the hard-constrained formulation is highly sensitive to the choice of the quota parameter $k'$, and in some cases incurs a substantial drop in approximation ratio (even lower than 0.2).*

**Results for $p$-NNS.** We experiment with $p \in \{-10, -1, -0.5, 0, 0.5, 1\}$ in both single- and multi-attribute settings and show that a trade-off between relevance (approximation ratio) and diversity (entropy) can be achieved by varying $p$. For the single-attribute setting, Figure 2 (Bottom row–columns 1 and 2), and for the multi-attribute setting, Figure 2 (Bottom row–columns 3 and 4) capture this feature on `Sift1m-(Clus)` dataset with $k = 50$: For lower values of $p$, we have higher entropy but lower approximation ratio, while $p = 1$ matches `ANN`. For the multi-attribute setting, we show results for attribute classes $C_1$ (column 3) and $C_2$ (column 4) in Figure 2 bottom row, whereas the results for $C_3$ and $C_4$ are shown in Figure 35. Note that in the multi-attribute setting, `Multi p-mean-ANN` with $p = -10$ Pareto dominates `Multi Div-ANN` with $k' = 1$ in terms of approximation ratio and entropy. Moreover, analogous results are obtained for other datasets and metrics; see Appendix F.3 and F.4.

## 5 CONCLUSION

In this work, we formulated diversity in neighbor search with a welfarist perspective, using Nash social welfare (NSW) and $p$-mean welfare as objectives. Our NSW formulation balances diversity and relevance in a query-dependent manner, satisfies several desirable axiomatic properties, and is naturally applicable in both single-attribute and multi-attribute settings. With these properties, our formulation overcomes key limitations of the prior hard-constrained approach (Anand et al., 2025). Furthermore, the more general $p$-mean welfare interpolates between complete relevance ($p = 1$) and complete diversity ($p = -\infty$), offering practitioners a tunable parameter for real-world needs. Our formulations also admit provable and practical algorithms suited for low-latency scenarios. Experiments on real-world and semi-synthetic datasets validate their effectiveness in balancing diversity and relevance against existing baselines.

An important direction for future work is the design of sublinear-time approximation algorithms, in both single- and multi-attribute settings, that directly optimize our welfare objectives as part of ANN algorithms, thereby further improving efficiency. Another promising avenue is to extend welfare-based diversity objectives to settings without explicit attributes.

### ACKNOWLEDGEMENTS

Siddharth Barman, Nirjhar Das, and Shivam Gupta acknowledge the support of the Walmart Center for Tech Excellence (CSR WMGT-23-0001) and an Ittiam CSR Grant (OD/OTHR-24-0032).

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

# APPENDIX: WELFARIST FORMULATIONS FOR DIVERSE SIMILARITY SEARCH

## Table of Contents

## A   SUPPORTING LEMMAS FOR THE PROOF OF THEOREM 1

As in Algorithm 1, write $\widehat{D}_\ell$ to denote the $k$ nearest neighbors of the given query $q$ in the set $D_\ell$. Recall that in the single-attribute setting the sets $D_\ell$s are disjoint across $\ell \in [c]$. Also, $v^\ell_{(j)} \in \widehat{D}_\ell$ denotes the $j^{\text{th}}$ most similar vector to $q$ in $\widehat{D}_\ell$, for each index $j \in [k]$. We define function $F_\ell(\cdot)$ to

denote the logarithm of the cumulative similarity of prefixes of these vectors; in particular,

$$F_\ell(i) := \log \left( \sum_{j=1}^{i} \sigma(q, v_{(j)}^\ell) + \eta \right) \quad \text{for each } 1 \le i \le k. \tag{4}$$

Note that $F_\ell(i)$ is equal to the logarithm of the cumulative similarity of the $i$ most similar (to $q$) vectors in $D_\ell$. The lemma below shows that $F_\ell(\cdot)$ satisfies a useful decreasing marginals property.

**Lemma 5** (Decreasing Marginals). *For all attributes $\ell \in [c]$ and indices $i', i \in [k]$, with $i' < i$, it holds that*

$$F_\ell(i) - F_\ell(i-1) \le F_\ell(i') - F_\ell(i'-1) .$$

*Proof.* Note that

$$F_\ell(i) - F_\ell(i-1) = \log \left( \frac{\sum_{j=1}^{i} \sigma(q, v_{(j)}^\ell) + \eta}{\sum_{j=1}^{i-1} \sigma(q, v_{(j)}^\ell) + \eta} \right)$$

$$= \log \left( \frac{\sum_{j=1}^{i-1} \sigma(q, v_{(j)}^\ell) + \eta + \sigma(q, v_{(i)}^\ell)}{\sum_{j=1}^{i-1} \sigma(q, v_{(j)}^\ell) + \eta} \right)$$

$$= \log \left( 1 + \frac{\sigma(q, v_{(i)}^\ell)}{\sum_{j=1}^{i-1} \sigma(q, v_{(j)}^\ell) + \eta} \right) .$$

Similarly, we have $F_\ell(i') - F_\ell(i'-1) = \log \left( 1 + \frac{\sigma(q, v_{(i')}^\ell)}{\sum_{j=1}^{i'-1} \sigma(q, v_{(j)}^\ell) + \eta} \right)$.

In addition, the indexing of the vectors $v_{(j)}^\ell$ ensures that $\sigma(q, v_{(i')}^\ell) \ge \sigma(q, v_{(i)}^\ell)$ for $i' < i$. Moreover, since the similarities $\sigma(q, v)$ are non-negative for all $v \in P$, we have $\sum_{j=1}^{i-1} \sigma(q, v_{(j)}^\ell) \ge \sum_{j=1}^{i'-1} \sigma(q, v_{(j)}^\ell)$.

Combining these bounds, we obtain

$$\frac{\sigma(q, v_{(i')}^\ell)}{\sum_{j=1}^{i-1} \sigma(q, v_{(j)}^\ell) + \eta} \ge \frac{\sigma(q, v_{(i)}^\ell)}{\sum_{j=1}^{i'-1} \sigma(q, v_{(j)}^\ell) + \eta} .$$

Adding 1 to both sides of the last equation and taking log (which is an increasing function and, hence, preserves the inequality) gives us the desired inequality. Hence, the lemma stands proved. □

The following lemma asserts the Nash optimality of the subset returned by Algorithm 1, ALG, within a relevant class of solutions.

**Lemma 6.** *In the single-attribute setting, let* ALG *be the subset of vectors returned by Algorithm 1 and $S$ be any subset of input vectors with the property that $|S \cap D_\ell| = |\text{ALG} \cap D_\ell|$, for each $\ell \in [c]$. Then,* $\text{NSW}(\text{ALG}) \ge \text{NSW}(S)$.

*Proof.* Assume, towards a contradiction, that there exists a subset of input vectors $S$ that satisfies $|S \cap D_\ell| = |\text{ALG} \cap D_\ell|$, for each $\ell \in [c]$, and still induces NSW strictly greater than that of ALG. This strict inequality implies that there exists an attribute $a \in [c]$ with the property that the utility $u_a(S) > u_a(\text{ALG})$.[9] That is,

$$\sum_{t \in S \cap D_a} \sigma(q, t) > \sum_{v \in \text{ALG} \cap D_a} \sigma(q, v) \tag{5}$$

On the other hand, note that the construction of Algorithm 1 and the definition of $\widehat{D}_a$ ensure that the vectors in $\text{ALG} \cap D_a$ are in fact the most similar to $q$ among all the vectors in $D_a$. This observation and the fact that $|S \cap D_a| = |\text{ALG} \cap D_a|$ gives us $\sum_{v \in \text{ALG} \cap D_a} \sigma(q, v) \ge \sum_{t \in S \cap D_a} \sigma(q, t)$. This equation, however, contradicts the strict inequality (5).

Therefore, by way of contradiction, we obtain that there does not exist a subset $S$ such that $|S \cap D_\ell| = |\text{ALG} \cap D_\ell|$, for each $\ell \in [c]$, and $\text{NSW}(\text{ALG}) < \text{NSW}(S)$. The lemma stands proved. □

---

[9]Recall the utility model specified in Section 2.1.

## A.1 PROOF OF COROLLARY 2

Here we state the proof of Corollary 2.

**Corollary 2.** *In the single-attribute setting, given any query $q \in \mathbb{R}^d$ and an $\alpha$-approximate oracle ANN for $k$ most similar vectors from any set, Algorithm 1 (Nash-ANN) returns an $\alpha$-approximate solution for NaNNS, i.e., it returns a size-$k$ subset $\mathrm{ALG} \subseteq P$ with $\mathrm{NSW}(\mathrm{ALG}) \geq \alpha \max_{S \subseteq P: |S|=k} \mathrm{NSW}(S)$. The algorithm runs in time $O(kc) + \sum_{\ell=1}^{c} ANN(D_\ell, q)$, with $ANN(D_\ell, q)$ denoting the time required by the oracle to find $k$ similar vectors to $q$ in $D_\ell$.*

*Proof.* The running time of the algorithm is established by an argument similar to that in proof of Theorem 1. Therefore, we only argue correctness.

For every $\ell \in [c]$, let the $\alpha$-approximate oracle return $\widehat{D}_\ell$. Recall that $v_{(i)}^\ell$, $i \in [k]$, denotes the $i^{\text{th}}$ most similar point to $q$ in the set $\widehat{D}_\ell$. Further, for every $\ell \in [c]$, let $D_\ell^*$ be the set of $k$ most similar points to $q$ within $D_\ell$ and define $v_{(i)}^{*\ell}$, $i \in [k]$, to be the $i^{\text{th}}$ most similar point to $q$ in $D_\ell^*$. Recall that by the guarantee of the $\alpha$-approximate NNS oracle, we have $\sigma(q, v_{(i)}^\ell) \geq \alpha \cdot \sigma(q, v_{(i)}^{*\ell})$ for all $i \in [k]$. Let OPT be an optimal solution to the NaNNS problem containing $k_\ell^*$ most similar points of attribute $\ell$ for every $\ell \in [c]$.

Finally, let $\widehat{\mathrm{OPT}}$ be the optimal solution to the NaNNS problem when the set of vectors to search over is $P = \cup_{\ell \in [c]} \widehat{D}_\ell$.

By an argument similar to the proof of Theorem 1, we have $\mathrm{NSW}(\mathrm{ALG}) = \mathrm{NSW}(\widehat{\mathrm{OPT}})$. Therefore

$$\mathrm{NSW}(\mathrm{ALG}) = \mathrm{NSW}(\widehat{\mathrm{OPT}})$$

$$\geq \left( \prod_{\ell \in [c]} \left( \sum_{i=1}^{k_\ell^*} \sigma(q, v_{(i)}^\ell) + \eta \right) \right)^{\frac{1}{c}}$$

$$(\textstyle\bigcup_{\ell \in [c]: k_\ell^* \geq 1} \{v_{(1)}^\ell, \ldots, v_{(k_\ell^*)}^\ell\} \text{ is a feasible solution})$$

$$\geq \left( \prod_{\ell \in [c]} \left( \sum_{i=1}^{k_\ell^*} \alpha \sigma(q, v_{(i)}^{*\ell}) + \eta \right) \right)^{\frac{1}{c}}$$

$$(\text{by } \alpha\text{-approximate guarantee of the oracle; } k_\ell^* \leq k)$$

$$\geq \left( \prod_{\ell \in [c]} \alpha \left( \sum_{i=1}^{k_\ell^*} \sigma(q, v_{(i)}^{*\ell}) + \eta \right) \right)^{\frac{1}{c}} \qquad (\alpha \in (0,1))$$

$$= \alpha \, \mathrm{NSW}(\mathrm{OPT}) \qquad\qquad (\text{definition of OPT})$$

Hence, the corollary stands proved. □

## B STYLIZED EXAMPLES FOR NANNS

As stated in the introduction, depending on the query and the problem instance, solutions obtained via NaNNS can adjust between the ones obtained through standard NNS and those obtained via hard constraints. This is illustrated in the following stylized examples.

The first example shows that if all vectors have same similarity, then an optimal solution, $S^*$, for NaNNS is completely diverse, i.e., all the vectors in $S^*$ have different attributes.

The second example shows that if the vectors of only one attribute have high similarity with the given query, then a Nash optimal solution $S^*$ contains only vectors with that attribute.

**Example 1** (Complete Diversity via NaNNS)**.** Consider an instance in which, for a given query $q \in \mathbb{R}^d$, all vectors in $P$ are equally similar with the query: $\sigma(q, v) = 1$ for all $v \in P$. Also, let $|\mathrm{atb}(v)| = 1$ for all $v \in P$ and write $S^* \in \arg\max_{S \subseteq P: |S|=k} \mathrm{NSW}(S)$. If $c \geq k$, then here it holds that $|S^* \cap D_\ell| \leq 1$ for all $\ell \in [c]$.

*Proof.* Towards a contradiction, suppose there exists $T \in \arg\max_{S \subseteq P: |S|=k} \text{NSW}(S)$ such that $|T \cap D_{\ell^*}| > 1$ for some $\ell^* \in [c]$. Note that according to the setting specified in the example, $u_\ell(T) = |T \cap D_\ell| + \eta$ for all $\ell \in [c]$.

Since $c \geq k$ and $|T \cap D_{\ell^*}| > 1$, there exists $\ell' \in [c]$ such that $|T \cap D_{\ell'}| = 0$. Let $v^* \in T \cap D_{\ell^*}$ and $v' \in D_{\ell'}$ be two vectors. Consider the set $T' = (T \setminus \{v^*\}) \cup \{v'\}$. We have,

$$\frac{\text{NSW}(T')}{\text{NSW}(T)} = \left( \frac{(u_{\ell'}(T') + \eta)}{(u_{\ell'}(T) + \eta)} \cdot \frac{(u_{\ell^*}(T') + \eta)}{(u_{\ell^*}(T) + \eta)} \prod_{\ell \in [c] \setminus \{\ell^*, \ell'\}} \frac{(u_\ell(T') + \eta)}{(u_\ell(T) + \eta)} \right)^{\frac{1}{c}}$$

$$= \left( \frac{(1 + \eta)}{\eta} \cdot \frac{(u_{\ell^*}(T) - 1 + \eta)}{(u_{\ell^*}(T) + \eta)} \prod_{\ell \in [c] \setminus \{\ell^*, \ell'\}} \frac{(u_\ell(T) + \eta)}{(u_\ell(T) + \eta)} \right)^{\frac{1}{c}}$$

$$= \left( \frac{(1 + \eta)}{\eta} \cdot \frac{(u_{\ell^*}(T) - 1 + \eta)}{(u_{\ell^*}(T) + \eta)} \right)^{\frac{1}{c}}$$

$$= \left( \frac{u_{\ell^*}(T) - 1 + \eta u_{\ell^*}(T) + \eta^2}{\eta u_{\ell^*}(T) + \eta^2} \right)^{\frac{1}{c}} > 1 \qquad\qquad (u_{\ell^*}(T) \geq 2)$$

Therefore, we have $\text{NSW}(T') > \text{NSW}(T)$, which contradicts the optimality of $T$. Hence, we must have $|T \cap D_\ell| \leq 1$ for all $\ell \in [c]$, which proves the claim. $\qquad\square$

**Example 2** (Complete Relevance via NaNNS). Consider an instance in which for a given query $q \in \mathbb{R}^d$ and for a particular $\ell^* \in [c]$, only vectors $v \in D_{\ell^*}$ have similarity $\sigma(q, v) = 1$ and all other vectors $p' \in P \setminus D_{\ell^*}$ have similarity $\sigma(q, p') = 0$. Also, suppose that $|\text{atb}(p)| = 1$ for each $p \in P$, along with $|D_{\ell^*}| \geq k$. Then, for a Nash optimal solution $S^* \in \arg\max_{S \subseteq P, |S|=k} \text{NSW}(S)$ it holds that $|S^* \cap D_{\ell^*}| = k$. That is, for all other $\ell \in [c] \setminus \{\ell^*\}$ we have $|S^* \cap D_\ell| = 0$.

*Proof.* Towards a contradiction, suppose there exists $T \in \arg\max_{S \subseteq P: |S|=k} \text{NSW}(S)$ such that $|T \cap D_{\ell^*}| < k$. Therefore, there exists $\ell' \in [c] \setminus \{\ell^*\}$ such that $|T \cap D_{\ell'}| \geq 1$. Let $v^* \in D_{\ell^*} \setminus T$ and let $v' \in T \cap D_{\ell'}$. Note that $u_{\ell'}(T) = 0$ since $\sigma(q, v) = 0$ for all $v \in D_\ell$ for any $\ell \in [c] \setminus \{\ell^*\}$. Moreover, we also have $u_{\ell^*}(T) = |T \cap D_{\ell^*}|$.

Consider the set $T' = (T \setminus \{v'\}) \cup \{v^*\}$. We have,

$$\frac{\text{NSW}(T')}{\text{NSW}(T)} = \left( \frac{(u_{\ell'}(T') + \eta)}{(u_{\ell'}(T) + \eta)} \cdot \frac{(u_{\ell^*}(T') + \eta)}{(u_{\ell^*}(T) + \eta)} \prod_{\ell \in [c] \setminus \{\ell^*, \ell'\}} \frac{(u_\ell(T') + \eta)}{(u_\ell(T) + \eta)} \right)^{\frac{1}{c}}$$

$$= \left( \frac{(u_{\ell'}(T) - \sigma(q, v') + \eta)}{(u_{\ell'}(T) + \eta)} \cdot \frac{(u_{\ell^*}(T) + \sigma(q, v^*) + \eta)}{(u_{\ell^*}(T) + \eta)} \prod_{\ell \in [c] \setminus \{\ell^*, \ell'\}} \frac{(u_\ell(T) + \eta)}{(u_\ell(T) + \eta)} \right)^{\frac{1}{c}}$$

$$= \left( \frac{(0 - 0 + \eta)}{0 + \eta} \cdot \frac{(|T \cap D_{\ell^*}| + 1 + \eta)}{(|T \cap D_{\ell^*}| + \eta)} \right)^{\frac{1}{c}}$$

$$= \left( 1 + \frac{1}{|T \cap D_{\ell^*}| + \eta} \right)^{\frac{1}{c}} > 1 \,.$$

Therefore, we have obtained $\text{NSW}(T') > \text{NSW}(T)$, which contradicts the optimality of $T$. Hence, it must be the case that $|T \cap D_{\ell^*}| = k$, which proves the claim. $\qquad\square$

## C  PROOF OF THEOREM 3

This section restates and proves Theorem 3. Recall that in the multi-attribute setting, input vectors $v \in P$ are associated one or more attributes, $|\text{atb}(v)| \geq 1$.

**Theorem 3.** *In the multi-attribute setting, with parameter $\eta = 1$, NaNNS is NP-hard.*

*Proof.* Consider the decision version of the optimization problem: given a real $W$, decide whether there exists $S \subseteq P$, $|S| = k$ such that $\log \text{NSW}(S) \geq W$. We will refer to this problem as NaNNS. Note that the input to a NaNNS instance is the following: a set of vectors $P \subset \mathbb{R}^d$, $|P| = n$, similarity function $\sigma : \mathbb{R}^d \times \mathbb{R}^d \to \mathbb{R}_+$, integer $k \in \mathbb{N}$, the sets $D_\ell = \{p \in P : \ell \in \text{atb}(p)\}$ for every color $\ell \in [c]$, a query point $q \in \mathbb{R}^d$ and a real $W$. We will show that NaNNS is NP-Complete by reducing EXACT REGULAR SET PACKING (ERSP)[10] to NaNNS.

In ERSP, we are given a universe of $n$ elements that we denote by $\mathcal{U} = \{1, 2, \dots, n\}$, a collection of subsets $\mathcal{S} = \{S_1, \dots, S_m\}$ where $S_i \subseteq \mathcal{U}$, $|S_i| = \tau$, for all $i \in [m]$ and an integer $k \in \mathbb{N}$. The problem is to decide if there is a sub-collection $I \subseteq \mathcal{S}$, $|I| = k$, such that for all distinct $S, S' \in I$ $S \cap S' = \emptyset$.

To reduce ERSP to an instance of NaNNS, we view $\mathcal{U}$ as the set of attributes, i.e., $c = n$. The set of vectors $P$ is embedded in $\mathbb{R}^n$ and is given by $P = \{\frac{1}{\tau} \cdot \mathbf{1}_S \mid S \in \mathcal{S}\}$ and the query vector is $q = \mathbf{1}$. Here, $\mathbf{1}$ is the all ones vector in $\mathbb{R}^n$, and $\mathbf{1}_S$ is the vector in $\mathbb{R}^n$ whose $i$-th coordinate is $\mathbb{1}\{i \in S\}$ for all $i \in [n]$. Therefore, the set of vectors $P$ is of size $m$. Moreover, the set of vectors having attribute $\ell \in [n]$ is denoted by $D_\ell = \{\frac{1}{\tau} \cdot \mathbf{1}_S \mid S \in \mathcal{S}, \ell \in S\}$. The size of the solution set of the NaNNS is equal to the $k$ of the ERSP instance. Finally, the similarity function $\sigma : \mathbb{R}^n \times \mathbb{R}^n \to \mathbb{R}$ is taken to be the usual dot-product. Finally, we set $W = \tau k \log 2$. Note that the reduction takes time polynomial in $n$ and $m$.

Also note that for any $v \in P$, $\sigma(q, v) = \langle \frac{1}{\tau} \cdot \mathbf{1}_S, \mathbf{1} \rangle = 1$ where $v = \frac{1}{\tau} \cdot \mathbf{1}_S$ for some $S \in \mathcal{S}$.

Now we prove the correctness of the reduction.

"$\Rightarrow$": Suppose $I^* \subset \mathcal{S}$, $|I^*| = k$ is a solution to ERSP instance. Consider the set $N^* := \{\frac{1}{\tau} \cdot \mathbf{1}_S : S \in I^*\}$. Clearly, $N^* \subseteq P$ and $|N^*| = k$, hence $N^*$ is a feasible set of the NaNNS problem. Now, since $I^*$ is a solution to the ERSP instance, for distinct $S, S' \in I^*$ we have $S \cap S' = \emptyset$. Particularly, if for an attribute $\ell \in [c]$, we have $\ell \in S$ for some $S \in I^*$, then $\ell \notin S'$ for all $S' \in I^* \setminus \{S\}$. Therefore, $|N^* \cap D_\ell| \leq 1$ for all $\ell \in [c]$ which in turn implies that $u_\ell(N^*)$ is either 1 or 0 for all $\ell \in [c]$. Finally, note that any point $v \in P$ belongs to exactly $\tau$ attributes, i.e., $|\text{atb}(v)| = \tau$. Hence,

$$\log \text{NSW}(N^*) = \frac{1}{c} \sum_{\ell=1}^{c} \log(1 + u_\ell(N^*)) = \frac{1}{c} \sum_{v \in N^*} \sum_{\ell \in \text{atb}(v)} \log(1 + 1) = \frac{\tau k \log 2}{c} .$$

Therefore, if there is a solution to the ERSP instance, then the corresponding NaNNS instance also has a solution.

"$\Leftarrow$": Suppose $N^* \subseteq P$, $|N^*| = k$, is a solution to the NaNNS instance (i.e., $\log \text{NSW}(N^*) \geq W$) corresponding to the ERSP instance. Define $I^* := \{S \mid \frac{1}{\tau} \cdot \mathbf{1}_S \in N^*\}$. Note that $|I^*| = k$. We will show that $I^*$ is a solution to the ERSP instance. First observe that,

$$\sum_{\ell \in [c]} u_\ell(N^*) = \sum_{\ell \in [c]} \sum_{v \in N^* \cap D_\ell} \sigma(q, v) = \sum_{v \in N^*} \sum_{\ell \in \text{atb}(v)} \sigma(q, v) = \tau k .$$

We also have the set of attributes with non-zero utility is given by $\mathcal{A} = \cup_{S \in I^*} S$. Clearly, $1 \leq |\mathcal{A}| \leq \tau k$ via Union Bound. Hence,

$$\begin{aligned}
W = \frac{\tau k \log 2}{c} \leq \log \text{NSW}(N^*) &= \frac{1}{c} \sum_{\ell \in [c]} \log(1 + u_\ell(N^*)) \\
&= \frac{1}{c} \sum_{\ell \in \mathcal{A}} \log(1 + u_\ell(N^*)) \\
&= \frac{|\mathcal{A}|}{c} \cdot \frac{1}{|\mathcal{A}|} \sum_{\ell \in \mathcal{A}} \log(1 + u_\ell(N^*)) \\
&\leq \frac{|\mathcal{A}|}{c} \cdot \log \left( \frac{1}{|\mathcal{A}|} \sum_{\ell \in \mathcal{A}} 1 + u_\ell(N^*) \right) \quad \text{(concavity of log)}
\end{aligned}$$

---

[10]ERSP is known to be NP-Complete due to Karp (1972) and W[1] hard with respect to solution size due to Ausiello et al. (1980); see also Garey & Johnson (1990)

$$= \frac{|\mathcal{A}|}{c} \cdot \log\left(1 + \frac{\sum_{\ell \in \mathcal{A}} u_\ell(N^*)}{|\mathcal{A}|}\right)$$

$$= \frac{|\mathcal{A}|}{c} \cdot \log\left(1 + \frac{\tau k}{|\mathcal{A}|}\right)$$

$$\leq \frac{\tau k \log 2}{c}$$

Here, the last inequality follows from Lemma 7 (stated and proved below). Hence, all the inequalities in the derivation above must hold with equality. Particularly, we must have $|\mathcal{A}| = \tau k$ by equality condition of Lemma 7. Hence, for distinct sets $S, S' \in I^*$, we must have $S \cap S' = \emptyset$. Therefore, $I^*$ is a solution of the ERSP instance.

$\square$

**Lemma 7.** *For any $a > 0$ and for all $x \in (0, a]$, $x \log(1 + \frac{a}{x}) \leq a \log 2$. Moreover, the equality holds when $x = a$.*

*Proof.* Let $f(x) := x \log(1 + \frac{a}{x})$. We have $f(a) = a \log(2)$ and,

$$\lim_{x \to 0^+} f(x) = \lim_{x \to 0^+} x \log(a + x) - x \log x = \lim_{x \to 0^+} x \log(a + x) - \lim_{x \to 0^+} x \log(x) = 0 - 0 = 0 .$$

Note that $f'(x) = \log(1 + \frac{a}{x}) - \frac{a}{a+x}$. We will show that $f'(x) > 0$ for all $x \in (0, a]$ which will conclude the proof.

**Case 1**: $x \in (0, \frac{a}{2}]$. We have $\log(1 + \frac{a}{x}) \geq \log(1 + \frac{a}{a/2}) = \log(3) > 1$. On the other hand, $\frac{a}{a+x} \leq 1$.

**Case 2**: $x \in (\frac{a}{2}, a]$. In this case, $\log(1 + \frac{a}{x}) \geq \log(1 + \frac{a}{a}) = \log(2) > 0.693$. However, $\frac{a}{a+x} < \frac{a}{a+\frac{a}{2}} = \frac{2}{3} \leq 0.667$.

Therefore, $f'(x) = \log(1 + \frac{a}{x}) - \frac{a}{a+x} > 0$ for all $x \in (0, a]$ which concludes the proof. $\square$

## D    PROOF OF THEOREM 4

This section details Algorithm 2, based on which we obtain Theorem 4. We establish this theorem by showing that the $\log \text{NSW}(\cdot)$ objective is submodular. Hence, we obtain the stated $\left(1 - \frac{1}{e}\right)$-approximation by applying the approximation algorithm for cardinality-constrained submodular maximization (Nemhauser et al., 1978).

---
**Algorithm 2** `MultiNashANN`: Algorithm for approximate solution in the multi-attribute case
---
**Require:** Query $q \in \mathbb{R}^d$.
1: Initialize $\text{ALG} = \emptyset$.
2: **for** $i = 1, \ldots, k$ **do**
3:    $\widehat{v} = \arg\max_{v \in P \setminus \text{ALG}} \log \text{NSW}(\text{ALG} \cup \{v\})$.
4:    $\text{ALG} \leftarrow \text{ALG} \cup \{\widehat{v}\}$.
5: **Return** $\text{ALG}$.

---

**Theorem 4.** *In the multi-attribute setting, there exists a polynomial-time algorithm (Algorithm 2) that, given any query $q \in \mathbb{R}^d$, finds a size-$k$ subset $\text{ALG} \subseteq P$ with $\log \text{NSW}(\text{ALG}) \geq \left(1 - \frac{1}{e}\right) \log \text{NSW}(\text{OPT})$; here, $\text{OPT}$ denotes an optimal solution of (1).*

*Proof.* We will show that function $f : 2^P \to \mathbb{R}_+$, $f(S) = \log \text{NSW}(S)$, $S \subseteq P$, is monotone submodular. Observe that for $S \subseteq T \subseteq P$, $D_\ell \cap S \subseteq D_\ell \cap T$, hence $u_\ell(S) \leq u_\ell(T)$ for all $\ell \in [c]$. Moreover, since $\log$ is an increasing function, $\log(u_\ell(S) + 1) \leq \log(u_\ell(T) + 1)$ for all $\ell \in [c]$. Therefore, we can conclude that $f(S) \leq f(T)$, hence $f$ is monotone.

For submodularity, let $S \subseteq T \subseteq P$ be two subsets and let $w \in P \setminus T$. We will denote by $S + w$ and $T + w$ the sets $S \cup \{w\}$ and $T \cup \{w\}$. Now, we have

$$f(S + w) - f(S) - f(T + w) + f(T)$$

$$= \frac{1}{c} \sum_{\ell \in [c]} \log \left( \frac{1 + \sum_{v \in D_\ell \cap (S+w)} \sigma(q, v)}{1 + \sum_{v \in D_\ell \cap S} \sigma(q, v)} \cdot \frac{1 + \sum_{v \in D_\ell \cap T} \sigma(q, v)}{1 + \sum_{v \in D_\ell \cap (T+w)} \sigma(q, v)} \right)$$

$$= \frac{1}{c} \sum_{\ell \in \mathrm{atb}(w)} \log \left( \left( 1 + \frac{\sigma(q, w)}{1 + \sum_{v \in D_\ell \cap S} \sigma(q, v)} \right) \cdot \left( 1 + \frac{\sigma(q, w)}{1 + \sum_{v \in D_\ell \cap T} \sigma(q, v)} \right)^{-1} \right)$$

$$= \frac{1}{c} \sum_{\ell \in \mathrm{atb}(w)} \log \left( \left( 1 + \frac{\sigma(q, w)}{1 + u_\ell(S)} \right) \cdot \left( 1 + \frac{\sigma(q, w)}{1 + u_\ell(T)} \right)^{-1} \right)$$

$$\geq 0 \qquad\qquad (u_\ell(S) \leq u_\ell(T) \text{ since } S \subseteq T)$$

Therefore, upon rearranging, we have $f(S + w) - f(S) \geq f(T + w) - f(T)$ which is a characterization of submodularity.

Hence, Algorithm 2, which, in every iteration, greedily picks the element with maximum marginal contribution, achieves a $(1 - \frac{1}{e})$-approximation (Nemhauser et al., 1978). $\qquad\square$

## E  EXTENSIONS FOR $p$-NNS

In this section, we discuss the extension of results for NaNNS to the $p$-NNS problem. We state an algorithm (Algorithm 3) and present its guarantee (Theorem 11 and Corollary 12) in finding the exact optimal solution for the $p$-NNS problem. Recall that $p$-mean welfare of $c$ agents with utilities $(w_1, \ldots, w_c)$ is given by $M_p(w_1, \ldots, w_c) = \left( \frac{1}{c} \sum_{\ell=1}^c w_\ell^p \right)^{\frac{1}{p}}$ for $p \in (-\infty, 1]$. The $p$-NNS problem is stated as follows:

$$\max_{S \subseteq P : |S| = k} M_p\left( u_1(S), \ldots, u_c(S) \right)$$

Here, as in Section 2.1, the utility $u_\ell(S) = \sum_{v \in S \cap D_\ell} \sigma(q, v)$, for any subset of vectors $S$ and attribute $\ell \in [c]$. Also, we will write $M_p(S) := M_p\left( u_1(S), \ldots, u_c(S) \right)$.

---

**Algorithm 3** p-Mean-ANN: Algorithm for $p$-NNS in the single-attribute setting

---

**Require:** Query $q \in \mathbb{R}^d$ and, for each attribute $\ell \in [c]$, the set of input vectors $D_\ell \subset \mathbb{R}^d$ and $p \in (-\infty, 1] \setminus \{0\}$.
1: For each $\ell \in [c]$, fetch the $k$ (exact or approximate) nearest neighbors of $q \in \mathbb{R}^d$ from $D_\ell$. Write $\widehat{D}_\ell \subseteq D_\ell$ to denote these sets.
2: For every $\ell \in [c]$ and each index $i \in [k]$, let $v_{(i)}^\ell$ denote the $i$th most similar vector to $q$ in $\widehat{D}_\ell$.
3: Initialize subset $\mathrm{ALG} = \emptyset$, along with count $k_\ell = 0$ and utility $w_\ell = 0$, for each $\ell \in [c]$.
4: **while** $|\mathrm{ALG}| < k$ **do**
5:    **if** $p \in (0, 1]$ **then**
6:       Let $a = \arg\max_{\ell \in [c]} \left( (w_\ell + \eta + \sigma(q, v_{(k_\ell + 1)}^\ell))^p - (w_\ell + \eta)^p \right)$. {Ties broken arbitrarily.}
7:    **else if** $p < 0$ **then**
8:       Let $a = \arg\min_{\ell \in [c]} \left( (w_\ell + \eta + \sigma(q, v_{(k_\ell + 1)}^\ell))^p - (w_\ell + \eta)^p \right)$. {Ties broken arbitrarily.}
9:    Update $\mathrm{ALG} \leftarrow \mathrm{ALG} \cup \left\{ v_{(k_a + 1)}^a \right\}$, along with $w_a \leftarrow w_a + \sigma(q, v_{(k_a + 1)}^a)$ and $k_a \leftarrow k_a + 1$.
10: **Return** $\mathrm{ALG}$.

---

For a given query $q$, let $f_\ell(i) = \sum_{j=1}^i \sigma(q, v_{(i)}^\ell)$ for any attribute $\ell \in [c]$ and index $i \in [k]$.

**Lemma 8** (Decreasing Marginal for $p > 0$). *Fix a $p \in (0, 1]$ and attribute $\ell \in [c]$. Let $F_\ell(i) = (f_\ell(i) + \eta)^p$. Then, for $1 \leq i' < i \leq k$, we have*

$$F_\ell(i') - F_\ell(i' - 1) \geq F_\ell(i) - F_\ell(i - 1) .$$

*Proof.* Let $G(j) := F_\ell(j) - F_\ell(j-1)$ for all $j \in [k]$. We will show that $G(j)$ is decreasing in $j$. Towards this, we have the following inequalities for $j \geq 2$:

$$G(j-1) - G(j)$$
$$= F_\ell(j-1) - F_\ell(j-2) - F_\ell(j) + F_\ell(j-1)$$
$$= 2F_\ell(j-1) - (F_\ell(j) + F_\ell(j-2))$$
$$= 2(f_\ell(j-1) + \eta)^p - ((f_\ell(j) + \eta)^p + (f_\ell(j-2) + \eta)^p)$$
$$= 2(f_\ell(j-2) + \eta)^p \left( \left( 1 + \frac{\sigma(q, v_{(j-1)}^\ell)}{f_\ell(j-2) + \eta} \right)^p - \frac{1}{2} \left( 1 + \left( 1 + \frac{\sigma(q, v_{(j-1)}^\ell) + \sigma(q, v_{(j)}^\ell)}{f_\ell(j-2) + \eta} \right)^p \right) \right)$$
$$\geq 2(f_\ell(j-2) + \eta)^p \left( \left( 1 + \frac{\sigma(q, v_{(j-1)}^\ell)}{f_\ell(j-2) + \eta} \right)^p - \frac{1}{2} \left( 1 + \left( 1 + \frac{2\sigma(q, v_{(j-1)}^\ell)}{f_\ell(j-2) + \eta} \right)^p \right) \right)$$
$$\qquad (\sigma(q, v_{(j-1)}^\ell) \geq \sigma(q, v_{(j)}^\ell); x \mapsto x^p \text{ is increasing for } p \in (0, 1] \text{ and } x \geq 0)$$
$$\geq 2(f_\ell(j-2) + \eta)^p \left( \left( 1 + \frac{\sigma(q, v_{(j-1)}^\ell)}{f_\ell(j-2) + \eta} \right)^p - \left( \frac{1}{2} \cdot 1 + \frac{1}{2} \cdot \left( 1 + \frac{2\sigma(q, v_{(j-1)}^\ell)}{f_\ell(j-2) + \eta} \right) \right)^p \right)$$
$$\qquad (x \mapsto x^p \text{ is concave for } p \in (0, 1] \text{ and } x \geq 0)$$
$$= 0 .$$

Therefore, we have $G(j) \leq G(j-1)$ for all $2 \leq j \leq k$. Particularly, for $1 \leq i' < i \leq k$, we have $G(i') \geq G(i)$, which is the claimed inequality. $\qquad\square$

**Lemma 9** (Increasing Marginals for $p < 0$). *Fix a $p \in (-\infty, 0)$ and attribute $\ell \in [c]$. Let $F_\ell(i) = (f_\ell(i) + \eta)^p$. Then, for $1 \leq i' < i \leq k$, we have*

$$F_\ell(i') - F_\ell(i' - 1) \leq F_\ell(i) - F_\ell(i-1) .$$

*Proof.* The proof proceeds similarly to the proof of Lemma 8, except that we now seek the reverse inequality. More precisely, with $G(j)$ same as defined in proof of Lemma 8, we wish to show that $G(j) \geq G(j-1)$ for all $2 \leq j \leq k$. Towards this, we have,

$$G(j-1) - G(j)$$
$$= 2(f_\ell(j-2) + \eta)^p \left( \left( 1 + \frac{\sigma(q, v_{(j-1)}^\ell)}{f_\ell(j-2) + \eta} \right)^p - \frac{1}{2} \left( 1 + \left( 1 + \frac{\sigma(q, v_{(j-1)}^\ell) + \sigma(q, v_{(j)}^\ell)}{f_\ell(j-2) + \eta} \right)^p \right) \right)$$
$$\leq 2(f_\ell(j-2) + \eta)^p \left( \left( 1 + \frac{\sigma(q, v_{(j-1)}^\ell)}{f_\ell(j-2) + \eta} \right)^p - \frac{1}{2} \left( 1 + \left( 1 + \frac{2\sigma(q, v_{(j-1)}^\ell)}{f_\ell(j-2) + \eta} \right)^p \right) \right)$$
$$\qquad (\sigma(q, v_{(j-1)}^\ell) \geq \sigma(q, v_{(j)}^\ell); x \mapsto x^p \text{ is decreasing for } p \in (-\infty, 0) \text{ and } x \geq 0)$$
$$\leq 2(f_\ell(j-2) + \eta)^p \left( \left( 1 + \frac{\sigma(q, v_{(j-1)}^\ell)}{f_\ell(j-2) + \eta} \right)^p - \left( \frac{1}{2} \cdot 1 + \frac{1}{2} \cdot \left( 1 + \frac{2\sigma(q, v_{(j-1)}^\ell)}{f_\ell(j-2) + \eta} \right) \right)^p \right)$$
$$\qquad (x \mapsto x^p \text{ is convex for } p \in (-\infty, 0) \text{ and } x \geq 0)$$
$$= 0 .$$

$\qquad\square$

**Lemma 10.** *In the single-attribute setting, let* ALG *be the subset of vectors returned by Algorithm 3 and $S$ be any subset of input vectors with the property that $|S \cap D_\ell| = |$ALG$\cap D_\ell|$, for each $\ell \in [c]$. Then, $M_p($ALG$) \geq M_p(S)$.*

*Proof.* Assume, towards a contradiction, that there exists a subset of input vectors $S$ that satisfies $|S \cap D_\ell| = |$ALG$\cap D_\ell|$, for each $\ell \in [c]$, and still induces p-mean welfare strictly greater than that of ALG. This strict inequality combined with the fact that $M_p(w_1, \ldots, w_c)$ is an increasing function of $w_i$s implies that there exists an attribute $a \in [c]$ with the property that the utility $u_a(S) >$

$u_a(\text{ALG})$.[11] That is,

$$\sum_{t \in S \cap D_a} \sigma(q,t) > \sum_{v \in \text{ALG} \cap D_a} \sigma(q,v) \tag{6}$$

On the other hand, note that the construction of Algorithm 3 and the definition of $\widehat{D}_a$ ensure that the vectors in $\text{ALG} \cap D_a$ are in fact the most similar to $q$ among all the vectors in $D_a$. This observation and the fact that $|S \cap D_a| = |\text{ALG} \cap D_a|$ gives us $\sum_{v \in \text{ALG} \cap D_a} \sigma(q,v) \geq \sum_{t \in S \cap D_a} \sigma(q,t)$. This equation, however, contradicts the strict inequality (6).

Therefore, by way of contradiction, we obtain that there does not exist a subset $S$ such that $|S \cap D_\ell| = |\text{ALG} \cap D_\ell|$, for each $\ell \in [c]$, and $M_p(\text{ALG}) < M_p(S)$. The lemma stands proved. $\square$

**Theorem 11.** *In the single-attribute setting, given any query $q \in \mathbb{R}^d$ and an (exact) oracle ENN for $k$ most similar vectors from any set, Algorithm 3 (p-mean-ANN) returns an optimal solution for p-NNS, i.e., it returns a size-$k$ subset $\text{ALG} \subseteq P$ that satisfies $\text{ALG} \in \arg\max_{S \subseteq P:|S|=k} M_p(S)$. Furthermore, the algorithm runs in time $O(kc) + \sum_{\ell=1}^{c} ENN(D_\ell, q)$, where $ENN(D_\ell, q)$ is the time required by the exact oracle to find $k$ most similar vectors to $q$ in $D_\ell$.*

*Proof.* The running time of the algorithm is established by the same arguments as in the running time analysis of Theorem 1.

For the correctness analysis, we divide the proof into two cases: $p < 0$ and $p \in (0,1]$.

**Case 1**: $p \in (0,1]$. Note that $x \mapsto x^p$ is an increasing function for $x \geq 0$. Hence, the problem $\max_{S \subseteq P,|S|=k} M_p(S)$ is equivalent to the problem $\max_{S \subseteq P,|S|=k} M_p(S)^p$ or in other words, $\max_{S \subseteq P,|S|=k} \frac{1}{c} \sum_{\ell=1}^{c} u_\ell(S)^p$. The proof hereafter proceeds essentially similar to the proof of Theorem 1. Let $k_\ell = |\text{ALG} \cap D_\ell|$ for all $\ell \in [c]$. Further, let $\text{OPT} \in \arg\max_{S \subseteq P,|S|=k} \frac{1}{c} \sum_{\ell=1}^{c} u_\ell(S)^p$ and $k_\ell^* = |\text{OPT} \cap D_\ell|$ for all $\ell \in [c]$, where OPT is chosen such that $\sum_{\ell=1}^{c} |k_\ell^* - k_\ell|$ is minimized.

We will prove that OPT satisfies $k_\ell^* = k_\ell$ for each $\ell \in [c]$. This guarantee along with Lemma 6 imply that, as desired, ALG is a p-mean welfare optimal solution.

Assume, towards a contradiction, that $k_\ell^* \neq k_\ell$ for some $\ell \in [c]$. Since $|\text{OPT}| = |\text{ALG}| = k$, there exists attributes $x, y \in [c]$ with the property that

$$k_x^* < k_x \qquad \text{and} \qquad k_y^* > k_y \tag{7}$$

With $F_\ell(i)$ as defined in Lemma 8, we have for any pair of indices $1 \leq i' < i \leq k$,

$$F_\ell(i') - F_\ell(i'-1) \geq F_\ell(i) - F_\ell(i-1) \tag{8}$$

Next, note that for any attribute $\ell \in [c]$, if Algorithm 3, at any point during its execution, has included $k_\ell'$ vectors of attribute $\ell$ in ALG, then at that point the maintained utility $w_\ell = f_\ell(k_\ell')$. Hence, at the beginning of any iteration of the algorithm, if the $k_\ell'$ denotes the number of selected vectors of each attribute $\ell \in [c]$, then the marginals considered in Line 6 are $F_\ell(k_\ell' + 1) - F_\ell(k_\ell')$. These observations and the selection criterion in Line 6 of the algorithm give us the following inequality for the counts $k_x = |\text{ALG} \cap D_x|$ and $k_y = |\text{ALG} \cap D_y|$ of the returned solution ALG:

$$F_x(k_x) - F_x(k_x - 1) \geq F_y(k_y + 1) - F_y(k_y) \tag{9}$$

Specifically, equation (9) follows by considering the iteration in which $k_x^{\text{th}}$ (last) vector of attribute $x$ was selected by the algorithm. Before that iteration the algorithm had selected $(k_x - 1)$ vectors of attribute $x$, and let $k_y'$ denote the number of vectors with attribute $y$ that have been selected till that point. Note that $k_y' \leq k_y$. The fact that the $k_x^{\text{th}}$ vector was (greedily) selected in Line 6, instead of including an additional vector of attribute $y$, gives $F_x(k_x) - F_x(k_x - 1) \geq F_y(k_y' + 1) - F_y(k_y') \geq F_y(k_y + 1) - F_y(k_y)$; here, the last inequality follows from equation (8). Hence, equation (9) holds.

Moreover,

$$F_x(k_x^* + 1) - F_x(k_x^*) \geq F_x(k_x) - F_x(k_x - 1) \qquad \text{(via eqns. (7) and (8))}$$
$$\geq F_y(k_y + 1) - F_y(k_y) \qquad \text{(via eqn. (9))}$$
$$\geq F_y(k_y^*) - F_y(k_y^* - 1) \tag{10}$$

---

[11]Recall the utility model specified in Section 2.1.

The last inequality follows from equations (7) and (8).

Recall that $v_{(i)}^\ell$ denotes the $i^{\text{th}}$ most similar (to $q$) vector in the set $\widehat{D}_\ell$. The definition of $\widehat{D}_\ell$ ensures that $v_{(i)}^\ell$ is in fact the $i^{\text{th}}$ most similar (to $q$) vector among the ones that have attribute $\ell$, i.e., $i^{\text{th}}$ most similar in all of $D_\ell$. Since OPT is an optimal solution, the $k_\ell^* = |\text{OPT} \cap D_\ell|$ vectors of attribute $\ell$ in OPT are the most similar $k_\ell^*$ vectors from $D_\ell$. That is, $\text{OPT} \cap D_\ell = \left\{ v_{(1)}^\ell, \ldots, v_{(k_\ell^*)}^\ell \right\}$, for each $\ell \in [c]$. This observation and the definition of $F_\ell(\cdot)$ imply that the $p$-th power of OPT's $p$-mean welfare satisfies

$$M_p(\text{OPT})^p = \frac{1}{c} \sum_{\ell=1}^{c} F_\ell(k_\ell^*). \tag{11}$$

Now, consider a subset of vectors $S$ obtained from OPT by including vector $v_{(k_x^*+1)}^x$ and removing $v_{(k_y^*)}^y$, i.e., $S = \left( \text{OPT} \cup \left\{ v_{(k_x^*+1)}^x \right\} \right) \setminus \left\{ v_{(k_y^*)}^y \right\}$. Note that

$$M_p(S)^p - M_p(\text{OPT})^p = \frac{1}{c} \Big( F_x(k_x^* + 1) - F_x(k_x^*) \Big) + \frac{1}{c} \Big( F_y(k_y^* - 1) - F_y(k_y^*) \Big)$$
$$\geq 0 \qquad\qquad\qquad \text{(via eqn. (10))}$$

Hence, $M_p(S) \geq M_p(\text{OPT})$. Given that OPT is a $p$-mean welfare optimal allocation, the last inequality must hold with an equality, $M_p(S) = M_p(\text{OPT})$, i.e., $S$ is an optimal solution as well. This, however, contradicts the choice of OPT as an optimal solution that minimizes $\sum_{\ell=1}^{c} |k_\ell^* - k_\ell|$ – note that $\sum_{\ell=1}^{c} \left| \widehat{k}_\ell - k_\ell \right| < \sum_{\ell=1}^{c} |k_\ell^* - k_\ell|$, where $\widehat{k}_\ell := |S \cap D_\ell|$.

Therefore, by way of contradiction, we obtain that $|\text{OPT} \cap D_\ell| = |\text{ALG} \cap D_\ell|$ for each $\ell \in [c]$. As mentioned previously, this guarantee along with Lemma 10 imply that ALG is a p-mean welfare optimal solution. This completes the proof of the theorem for the case $p \in (0, 1]$.

**Case 2**: $p < 0$. In this case, the proof follows a similar argument as the previous case. However, due to $p$ being negative, the key inequalities are reversed. We present the proof formally below for the sake of completeness.

Note that $x \mapsto x^p$ is a decreasing function for $x \geq 0$. Hence, the problem $\max_{S \subseteq P, |S|=k} M_p(S)$ is equivalent to the problem $\min_{S \subseteq P, |S|=k} M_p(S)^p$ or in other words, $\min_{S \subseteq P, |S|=k} \frac{1}{c} \sum_{\ell=1}^{c} u_\ell(S)^p$. Let $k_\ell = |\text{ALG} \cap D_\ell|$ for all $\ell \in [c]$. Further, let $\text{OPT} \in \arg\min_{S \subseteq P, |S|=k} \frac{1}{c} \sum_{\ell=1}^{c} u_\ell(S)^p$ and $k_\ell^* = |\text{OPT} \cap D_\ell|$ for all $\ell \in [c]$, where OPT is chosen such that $\sum_{\ell=1}^{c} |k_\ell^* - k_\ell|$ is minimized.

We will prove that OPT satisfies $k_\ell^* = k_\ell$ for each $\ell \in [c]$. This guarantee along with Lemma 10 imply that, as desired, ALG is a p-mean welfare optimal solution.

Assume, towards a contradiction, that $k_\ell^* \neq k_\ell$ for some $\ell \in [c]$. Since $|\text{OPT}| = |\text{ALG}| = k$, there exists attributes $x, y \in [c]$ with the property that

$$k_x^* < k_x \qquad \text{and} \qquad k_y^* > k_y \tag{12}$$

With $F_\ell(i)$ as defined in Lemma 9, we have for any pair of indices $1 \leq i' < i \leq k$,

$$F_\ell(i') - F_\ell(i' - 1) \leq F_\ell(i) - F_\ell(i - 1) \tag{13}$$

Next, note that for any attribute $\ell \in [c]$, if Algorithm 3, at any point during its execution, has included $k_\ell'$ vectors of attribute $\ell$ in ALG, then at that point the maintained utility $w_\ell = f_\ell(k_\ell')$. Hence, at the beginning of any iteration of the algorithm, if the $k_\ell'$ denotes the number of selected vectors of each attribute $\ell \in [c]$, then the marginals considered in Line 8 are $F_\ell(k_\ell' + 1) - F_\ell(k_\ell')$. These observations and the selection criterion in Line 8 of the algorithm give us the following inequality for the counts $k_x = |\text{ALG} \cap D_x|$ and $k_y = |\text{ALG} \cap D_y|$ of the returned solution ALG:

$$F_x(k_x) - F_x(k_x - 1) \leq F_y(k_y + 1) - F_y(k_y) \tag{14}$$

Specifically, equation (14) follows by considering the iteration in which $k_x^{\text{th}}$ (last) vector of attribute $x$ was selected by the algorithm. Before that iteration the algorithm had selected $(k_x - 1)$ vectors of attribute $x$, and let $k_y'$ denote the number of vectors with attribute $y$ that have been selected till that

point. Note that $k'_y \leq k_y$. The fact that the $k_x^{\text{th}}$ vector was (greedily) selected in Line 8, instead of including an additional vector of attribute $y$, gives $F_x(k_x) - F_x(k_x - 1) \leq F_y(k'_y + 1) - F_y(k'_y) \leq F_y(k_y + 1) - F_y(k_y)$; here, the last inequality follows from equation (13). Hence, equation (14) holds.

Moreover,

$$
\begin{aligned}
F_x(k_x^* + 1) - F_x(k_x^*) &\leq F_x(k_x) - F_x(k_x - 1) && \text{(via eqns. (12) and (13))} \\
&\leq F_y(k_y + 1) - F_y(k_y) && \text{(via eqn. (14))} \\
&\leq F_y(k_y^*) - F_y(k_y^* - 1) && (15)
\end{aligned}
$$

The last inequality follows from equations (12) and (13).

Recall that $v_{(i)}^\ell$ denotes the $i^{\text{th}}$ most similar (to $q$) vector in the set $\widehat{D}_\ell$. The definition of $\widehat{D}_\ell$ ensures that $v_{(i)}^\ell$ is in fact the $i^{\text{th}}$ most similar (to $q$) vector among the ones that have attribute $\ell$, i.e., $i^{\text{th}}$ most similar in all of $D_\ell$. Since OPT is an optimal solution, the $k_\ell^* = |\text{OPT} \cap D_\ell|$ vectors of attribute $\ell$ in OPT are the most similar $k_\ell^*$ vectors from $D_\ell$. That is, $\text{OPT} \cap D_\ell = \left\{ v_{(1)}^\ell, \ldots, v_{(k_\ell^*)}^\ell \right\}$, for each $\ell \in [c]$. This observation and the definition of $F_\ell(\cdot)$ imply that the $p$-th power of OPT's $p$-mean welfare satisfies

$$
M_p(\text{OPT})^p = \frac{1}{c} \sum_{\ell=1}^c F_\ell(k_\ell^*). \tag{16}
$$

Now, consider a subset of vectors $S$ obtained from OPT by including vector $v_{(k_x^*+1)}^x$ and removing $v_{(k_y^*)}^y$, i.e., $S = \left( \text{OPT} \cup \left\{ v_{(k_x^*+1)}^x \right\} \right) \setminus \left\{ v_{(k_y^*)}^y \right\}$. Note that

$$
\begin{aligned}
M_p(S)^p - M_p(\text{OPT})^p &= \frac{1}{c} \Big( F_x(k_x^* + 1) - F_x(k_x^*) \Big) + \frac{1}{c} \Big( F_y(k_y^* - 1) - F_y(k_y^*) \Big) \\
&\leq 0 && \text{(via eqn. (15))}
\end{aligned}
$$

Hence, $M_p(S) \geq M_p(\text{OPT})$. Given that OPT is a $p$-mean welfare optimal allocation, the last inequality must hold with an equality, $M_p(S) = M_p(\text{OPT})$, i.e., $S$ is an optimal solution as well. This, however, contradicts the choice of OPT as an optimal solution that minimizes $\sum_{\ell=1}^c |k_\ell^* - k_\ell|$ – note that $\sum_{\ell=1}^c \left| \widehat{k}_\ell - k_\ell \right| < \sum_{\ell=1}^c |k_\ell^* - k_\ell|$, where $\widehat{k}_\ell := |S \cap D_\ell|$.

Therefore, by way of contradiction, we obtain that $|\text{OPT} \cap D_\ell| = |\text{ALG} \cap D_\ell|$ for each $\ell \in [c]$. As mentioned previously, this guarantee along with Lemma 10 imply that ALG is a p-mean welfare optimal solution. This completes the proof of the theorem for the case $p < 0$.

Combining the two cases we have the proof of the theorem for all $p \in (-\infty, 1] \setminus \{0\}$. □

**Corollary 12.** *In the single-attribute setting, given any query $q \in \mathbb{R}^d$ and an $\alpha$-approximate oracle ANN for $k$ most similar vectors from any set, Algorithm 3 (p-mean-ANN) returns an $\alpha$-approximate solution for $p$-NNS, i.e., it returns a size-$k$ subset $\text{ALG} \subseteq P$ with $M_p(\text{ALG}) \geq \alpha \max_{S \subseteq P: |S|=k} M_p(S)$. The algorithm runs in time $O(kc) + \sum_{\ell=1}^c \text{ANN}(D_\ell, q)$, with $\text{ANN}(D_\ell, q)$ being the time required by the approximate oracle to find $k$ similar vectors to $q$ in $D_\ell$.*

*Proof.* The running time of the algorithm is established by an argument similar to that in proof of Theorem 11. Therefore, we only argue correctness.

For every $\ell \in [c]$, let the $\alpha$-approximate oracle return $\widehat{D}_\ell$. Recall that $v_{(i)}^\ell$, $i \in [k]$, denotes the $i^{\text{th}}$ most similar point to $q$ in the set $\widehat{D}_\ell$. Further, for every $\ell \in [c]$, let $D_\ell^*$ be the set of $k$ most similar points to $q$ within $D_\ell$ and define $v_{(i)}^{*\ell}$, $i \in [k]$, to be the $i^{\text{th}}$ most similar point to $q$ in $D_\ell^*$. Recall that by the guarantee of the $\alpha$-approximate NNS oracle, we have $\sigma(q, v_{(i)}^\ell) \geq \alpha \cdot \sigma(q, v_{(i)}^{*\ell})$ for all $i \in [k]$. Let $\text{ALG}^*$ be the solution obtained by running p-mean-ANN with an exact NNS oracle, and let $\text{ALG}^*$ contain $k_\ell^*$ most similar points of attribute $\ell$ for every $\ell \in [c]$. Moreover, let OPT be the optimal solution to the $p$-NNS problem. Note that we have by Theorem 11, $M_p(\text{ALG}^*) = M_p(\text{OPT})$.

Finally, let $\widehat{\text{OPT}}$ be the optimal solution to the $p$-NNS problem when the set of vectors to search over is $P = \cup_{\ell \in [c]} \widehat{D}_\ell$.

By an argument similar to the proof of Theorem 11, we have $M_p(\text{ALG}) = M_p(\widehat{\text{OPT}})$. Therefore, we can write,

$$M_p(\text{ALG}) = M_p(\widehat{\text{OPT}})$$

$$\geq \left( \frac{1}{c} \sum_{\ell \in [c]} \left( \sum_{i=1}^{k_\ell^*} \sigma(q, v_{(i)}^\ell) + \eta \right)^p \right)^{\frac{1}{p}}$$

$$(\cup_{\ell \in [c] : k_\ell^* \geq 1} \{v_{(1)}^\ell, \ldots, v_{(k_\ell^*)}^\ell\} \text{ is a feasible solution})$$

$$\geq \left( \frac{1}{c} \sum_{\ell \in [c]} \left( \sum_{i=1}^{k_\ell^*} \alpha \sigma(q, v_{(i)}^{*\ell}) + \eta \right)^p \right)^{\frac{1}{p}}$$

(by $\alpha$-approximate guarantee of the oracle, and $M_p$ is increasing in its argument)

$$= \alpha \left( \frac{1}{c} \sum_{\ell \in [c]} \left( \sum_{i=1}^{k_\ell^*} \sigma(q, v_{(i)}^{*\ell}) + \frac{\eta}{\alpha} \right)^p \right)^{\frac{1}{p}}$$

$$\geq \alpha \left( \frac{1}{c} \sum_{\ell \in [c]} \left( \sum_{i=1}^{k_\ell^*} \sigma(q, v_{(i)}^{*\ell}) + \eta \right)^p \right)^{\frac{1}{p}} \qquad (\alpha \in (0,1) \text{ and } M_p \text{ is increasing})$$

$$= \alpha \, M_p(\text{ALG}^*)$$

$$= \alpha \, M_p(\text{OPT}) \qquad\qquad\qquad\qquad\qquad\qquad\qquad \text{(by Theorem 11)}$$

Hence, the corollary stands proved. $\qquad\qquad\qquad\qquad\qquad\qquad\qquad\qquad\qquad\qquad\qquad\qquad$ $\square$

## F   EXPERIMENTAL EVALUATION AND ANALYSIS

In this section, we present additional experimental results to further validate the performance of `Nash-ANN` in comparison with the existing methods. We begin with a detailed discussion of the evaluation metrics (Appendix F.1), followed by a description of the datasets used in our study (Appendix F.2). Next, we report results for the single-attribute setting (Appendix F.3), where we compare the approximation ratio alongside all diversity metrics for $k = 10$ and $k = 50$. We also include recall values for both $k = 10$ and $k = 50$ (Appendix F.3.5). The key observation in all these plots is that the NSW objective effectively strikes a balance between relevance and diversity without having to specify any ad hoc constraints like quotas. Furthermore, we report experimental results for the multi-attribute setting on both a synthetic dataset (`Sift1m`) and a real-world dataset (`ArXiv`). Finally, we experimentally validate the performance-efficiency trade-offs of a faster heuristic variant of $p$-`mean-ANN` that can be used in addition to any existing (standard) ANN algorithm.

### F.1   EVALUATION METRICS

We evaluate the performance of our proposed methods against baseline algorithms using the following metrics:

**Relevance Metrics**:

1. **Approximation Ratio**: For a given query $q$, let an algorithm of choice return the set $S_1$ and let a (standard) ANN algorithm return the set $S_2$. Then the approximation ratio of the algorithm is defined as the ratio $\frac{\sum_{p \in S_1} \sigma(q,p)}{\sum_{p \in S_2} \sigma(q,p)}$. Therefore, a higher approximation ratio indicates a more relevant solution.

2. **Recall**: For a given query $q$, let $S^*$ be the set of exact nearest neighbors of $q$ and let $S_1$ be the output of an algorithm. Then the recall of the algorithm is the quantity $\frac{|S_1 \cap S^*|}{|S^*|}$. Therefore, a higher recall indicates a more relevant solution.

It is important to note that recall is a fragile metric when the objective is to retrieve a relevant-cum-diverse set of vectors for a given query. This can be illustrated with the following stylized example in the single-attribute setting. Suppose for a given query $q$, all the vectors in the set of exact nearest neighbors $S^*$ have similarity 1, i.e., for all $p \in S^*$, $\sigma(q, p) = 1$. However, let all vectors $p \in S^*$ be associated with the same attribute $\ell^* \in [c]$, i.e., $\mathrm{atb}(p) = \ell^*$ for all $p \in S^*$. Therefore, the set of exact nearest neighbors is not at all diverse. However, it may be the case that there is a set $S'$ of $k$ vectors, all having different attributes (i.e. $\mathrm{atb}(p) \neq \mathrm{atb}(p')$ and $\mathrm{atb}(p) \neq \ell^*$ for $p, p' \in S'$, $p \neq p'$), such that $\sigma(q, p') = 0.99$ for all $p' \in S'$. In other words, there is a highly relevant set of vectors that is also completely diverse. Note that for the set $S'$, the recall is actually 0 but the approximation ratio is 0.99. Hence, in the context of diverse neighbor search problem, instead of recall, approximation ratio may be a more meaningful relevance metric.

**Diversity Metrics**:

- **Entropy**: Let $S \subseteq P$, $|S| = k$, be the output of an algorithm. Then the entropy of the set of $S$ in the single-attribute setting is given by the quantity $\sum_{\ell \in [c]: p_\ell > 0} -p_\ell \log(p_\ell)$ where $p_\ell = \frac{|S \cap D_\ell|}{|S|}$. Note that a higher entropy value indicates greater diversity.

- **Inverse Simpson Index**: For a given set $S \subseteq P$, $|S| = k$ in the single-attribute setting, the inverse Simpson index index is defined as $\frac{1}{\sum_{\ell=1}^{c} p_\ell^2}$ where $p_\ell$ is as defined in the definition of entropy above. A higher value indicates greater diversity.

- **Distinct Attribute Count**: In the single-attribute setting, the distinct attribute count of a set $S \subseteq P$, $|S| = k$ is the number $|\{\ell \in [c] : |S \cap D_\ell| > 0\}|$.

In the multi-attribute setting, in this work, we focus on settings where the attribute set $[c]$ is partitioned into $m$ sets $\{C_i\}_{i=1}^{m}$ (i.e., $[c] = \sqcup_{i=1}^{m} C_i$) and every input vector $v \in P$ is associated with one attribute from each $C_i$, i.e., $|\mathrm{atb}(v)| = m$ and $|\mathrm{atb}(v) \cap C_i| = 1$. To measure diversity in the multi-attribute setting, we consider the aforementioned diversity metrics like entropy and inverse Simpson index restricted to a $C_i$. More precisely, the entropy a set $S \subseteq P$ restricted to a particular $C_i$ is given by $\sum_{\ell \in C_i} -p_\ell \log(p_\ell)$ where $p_\ell = \frac{|S \cap D_\ell|}{|S|}$. Similarly, the inverse Simpson index of a set $S \subseteq P$ restricted to $C_i$ is given by $\frac{1}{\sum_{\ell \in C_i} p_\ell^2}$ where $p_\ell$ is as defined before.

## F.2 Datasets

1. **Amazon Products Dataset** (`Amazon`): The dataset also known as the Shopping Queries Image Dataset (SQID) (Ghossein et al., 2024), is based on the Amazon Shopping Queries dataset (Reddy et al., 2022) that is publicly available on the KDD Cup 2022 Challenge website[12]. The SQID includes image embeddings for about $190,000$ products listed in the Amazon Shopping Queries dataset along with the text embeddings of user queries present in the same dataset. The image and text embeddings are obtained via the use of OpenAI's CLIP model (Radford et al., 2021) which maps both images and texts into a shared vector space. The task is to retrieve product images relevant to a given text query. The SQID also contains metadata such as product image url, product id, product description, product title, product color, etc. The dataset is publicly available on Hugging Face platform.[13]

   We directly use the embeddings from the Hugging Face repository and map product id-s to retrieve additional metadata from the Amazon KDD dataset. We use $\sigma(u, v) = 1 + \frac{u^\top v}{\|u\| \cdot \|v\|}$ as the similarity function between two vectors $u$ and $v$. Note that the image and text embeddings in the dataset were generated using the cosine similarity metric in the loss function (see (Ghossein et al., 2024), Section 4.2) hence the similarity function defined in this work is a natural choice. We choose the set of product colors as our set of attributes $[c]$. To obtain a clean label for the product color of a given product in the dataset, we apply majority voting among the colors listed in the product color, description, and title of the product. In the event of a tie, we assign the item to a separate color class labeled 'color_mix' (e.g., if the title says 'blue' but the color column says 'red'). Product entries in

---

[12] https://amazonkddcup.github.io
[13] https://huggingface.co/datasets/crossingminds/shopping-queries-image-dataset

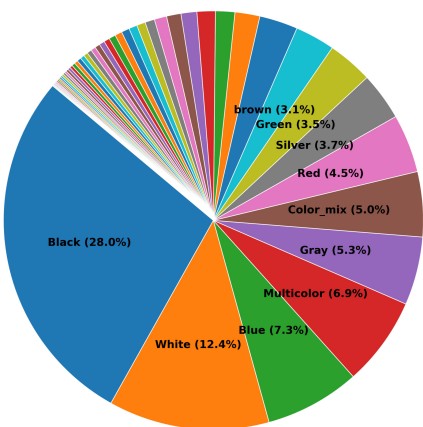

Figure 3: Distribution of product colors in the processed (cleaned) `Amazon` dataset.

the dataset whose metadata does not contain any valid color names are removed. The pre-processing script will be shared with reviewers as an anonymous repository during the open discussion phase and will be publicly released in the camera-ready version. The processed dataset contains approximately $92,092$ vector embeddings of products. Note that we do not apply any pre-processing to the query set which contains $8,956$ vector. The vector embeddings of both images and queries are 768 dimensional. Note that the dataset exhibits a skewed color distribution, shown in Figure 3), with some dominant colors such as black and white.

2. **ArXiv OpenAI Embedding** (`ArXiv`): The dataset published by Cornell University consists of vector embeddings for approximately $250,000$ machine learning papers available through the arXiv search engine (Wester, 2022). The embedding of a given paper was generated using OpenAI's `text-embedding-ada-002` model on the augmented abstract of the paper that combined the paper's title, authors, year, and abstract. The dataset is publicly available on Kaggle[14] (Wester, 2022).

   We consider the year in which a paper was last updated as the attribute in the single-attribute setting, and additionally consider the arXiv category the paper belongs to as a second attribute in the multi-attribute setting. Note that this dataset does not contain a predefined query set; hence, we randomly split 20% of the total vector embeddings to serve as queries. Such queries simulate the task of finding papers similar to a given query paper. The similarity function used for this dataset is the reciprocal of the Euclidean distance between two vectors, i.e, for two vectors $u$ and $v$, $\sigma(u, v) = \frac{1}{\|u-v\|+\mu}$, where $\mu$ is a small constant to avoid issues for the case when $\|u - v\| = 0$. Typically, we set $\mu = \eta$ (recall that $\eta$ is the smoothening parameter in the definition of NSW($\cdot$)). The distribution of the input vectors across update-year and arXiv category are shown in Figure 4.

   For our experiments, we only consider papers with update-year between 2012 and 2025 (both inclusive) and belonging to one or more of the following arXiv categories: `cs.ai`, `math.oc`, `cs.lg`, `cs.cv`, `stat.ml`, `cs.ro`, `cs.cl`, `cs.ne`, `cs.ir`, `cs.sy`, `cs.hc`, `cs.cr`, `cs.cy`, `cs.sd`, `eess.as`, and `eess.iv`. The pre-processing script will be shared with reviewers as an anonymous repository during the open discussion phase and will be publicly released in the camera-ready version.

3. **SIFT Embeddings**: It is a popular benchmarking dataset for approximate nearest neighbor search using the Euclidean distance (TensorFlow, 2025). The dataset consists of pre-trained SIFT embeddings, with $1,000,000$ vectors for indexing and a separate set of $10,000$ vectors as the query set, both in a 128 dimensional space. The embeddings are publicly available[15] (TensorFlow, 2025). Note that this dataset does not contain any metadata that can be naturally adapted as attributes to model diversity. Therefore, we adopt two strategies for synthetic attribute generation:

---

[14]https://www.kaggle.com/datasets/awester/arxiv-embeddings
[15]https://www.tensorflow.org/datasets/catalog/sift1m

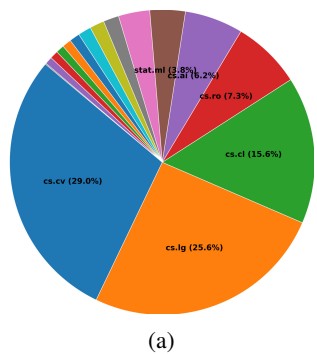 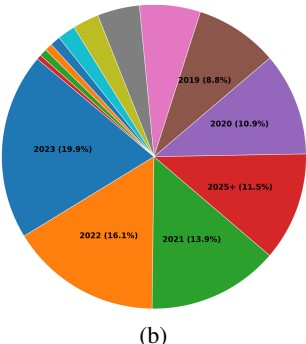

Figure 4: Distribution of (a) paper categories (b) last update year in the `ArXiv` dataset.

- **Clustering-based** (`Sift1m-(Clus)`): Since attributes such as color often occupy distinct regions in the embedding space, we follow a similar idea and apply $k$-means clustering to identify 20 clusters. Each cluster is then assigned a unique color, which serves as our synthetic attribute. Therefore, in this case $c = 20$.
- **Probability distribution-based** (`Sift1m-(Prob)`): To remain consistent with the prior work (Anand et al., 2025), we also adopt a randomized approach to color (attribute) assignment. For each vector, we assign one of three majority colors uniformly at random with probability 0.9, and with the remaining probability 0.1, one of the remaining 17 colors is assigned. This results in a skewed distribution over colors that mimics real-world settings (e.g., market dominance by a few sellers). The preprocessing script will be shared with reviewers as an anonymous repository during the open discussion phase and will be publicly released in the camera-ready version.

**Multi-attribute setting**: We extend the clustering-based attribute generation strategy to the multi-attribute setting as follows. We divide each 128 dimensional input vector $v$ into four equal segments of 32 dimensions $\{v^i\}_{i=1}^4$, i.e., $v^1 = v[1, \ldots, 32]$, $v^2 = v[33, \ldots, 64]$ etc. We then separately apply k-means clustering to compute 20 clusters on each segment, i.e., on the set of vectors $\{v^i : v \in P\}$ for each $i \in [4]$. Let the set of cluster ids be $C_i$ for the set of vectors $\{v^i : v \in P\}$, $i \in [4]$. Note that $|C_i| = 20$. Thereafter, the set of attributes assigned to the original input vector $v$ is the union of the cluster ids of $v^i$s. In other words, $\mathrm{atb}(v) = \sqcup_{i=1}^4 \{C_i(v^i)\}$ where $C_i(v^i)$ is the cluster id of $v^i$.

4. **Deep Descriptor Embeddings**: It is another benchmarking dataset for nearest neighbor search, evaluated using cosine distance (TensorFlow, 2025). The version used in this study contains approximately $9,990,000$ vectors for indexing and $10,000$ separate query vectors, both residing in a 96 dimensional space. These embeddings are publicly available[16] (TensorFlow, 2025), and we adopt the same synthetic attribute generation procedure as in the SIFT dataset to produce `Deep1b-(Clus)` and `Deep1b-(Prob)` variants.

**Choice of Parameter** $\eta$**:**  For our methods, we tune and set the smoothing parameter, $\eta$, to 0.01 for the `ArXiv`, `Sift1m-(Clus)` and `Sift1m-(Prob)` datasets in comparing relevance with diversity, and set it to 0.0001 to analyze performance at different values $p$. For other datasets, namely `Amazon`, `Deep1b-(Clus)` and `Deep1b-(Prob)`, we set $\eta$ to 50 for both experiments.

F.3    BALANCING RELEVANCE AND DIVERSITY: SINGLE-ATTRIBUTE SETTING

In this experiment, we evaluate the performance of $p$-`mean-ANN` (and the special case of $p = 0$, `Nash-ANN`) in its ability to balance relevance and diversity in the $p$-NNS (and NaNNS) problem in the single-attribute setting. We begin by examining the tradeoff between approximation ratio and entropy achieved by our algorithms on additional datasets beyond those used in the main paper. Moreover, we also report results for other diversity metrics such as the inverse Simpson index (Appendix F.3.3) and the number of distinct attributes appearing in the $k$ neighbors (Appendix F.3.4)

---

[16]https://www.tensorflow.org/datasets/catalog/deep1b

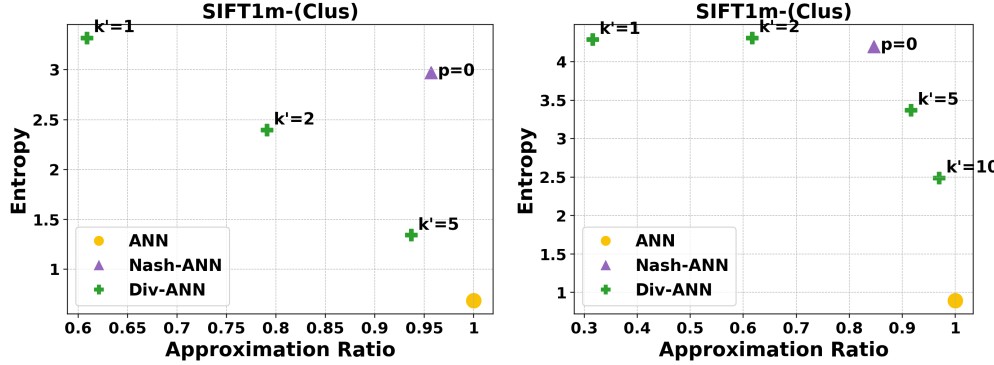

Figure 5: The plots show results on approximation ratio and entropy for **(Left)** $k = 10$; **(Right)** $k = 50$ in single-attribute setting on `Sift1m-(Clus)` dataset.

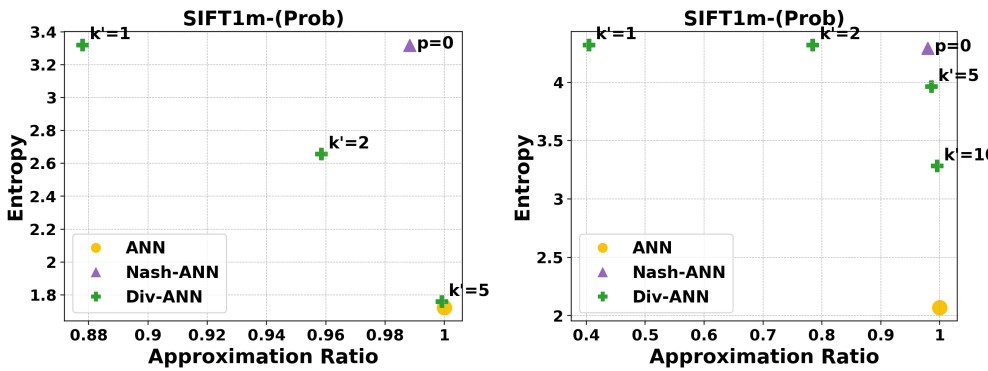

Figure 6: The plots show results on approximation ratio and entropy for **(Left)** $k = 10$; **(Right)** $k = 50$ in single-attribute setting on `Sift1m-(Prob)` dataset.

retrieved by our algorithms. These experiments corroborate the findings in the main paper, namely, `Nash-ANN` and $p$-mean-`ANN` are able to strike a balance between relevance and diversity whereas `ANN` only optimizes for relevance (hence low diversity) and `Div-ANN` only optimizes for diversity (hence low relevance).

### F.3.1 APPROXIMATION RATIO VERSUS ENTROPY

We report the results for different datasets in Figures 5, 6, 7, 8, and 9. On the `Sift1m-(Clus)` dataset (Figure 5), `Nash-ANN` achieves entropy close to that of the most diverse solution (`Div-ANN` with $k' = 1$) in both $k = 10$ and $k = 50$ cases. Moreover, `Nash-ANN` achieves significantly higher approximation ratio than `Div-ANN` in both $k = 10$ and $k = 50$ cases when $k' = 1$. For $k = 10$ case, `Nash-ANN` Pareto dominates `Div-ANN` even with the relaxed constraint of $k' = 5$ for $k = 10$. When the number of required neighbors is increased to $k = 50$, no other method Pareto dominates `Nash-ANN`. Similar observations hold for the `Sift1m-(Prob)` (Figure 6), and `Deep1b-(Prob)` (Figure 7) datasets. In the results on the `ArXiv` dataset (Figure 8) with $k = 10$, we observe that `Div-ANN` already achieves a high approximation ratio. However, `Nash-ANN` matches the entropy of `Div-ANN` with $k' = 1$ while improving on the approximation ratio. For $k = 50$, `Nash-ANN` nearly matches the entropy of `Div-ANN` with $k' = 1, 2$ whereas it significantly improves on the approximation ratio. The results for $k = 10$ for datasets in the main paper are shown in Figure 9 and have similar observations. In summary, the experimental results clearly demonstrate the ability of `Nash-ANN` to adapt to the varying nature of queries and consistently strike a balance between relevance and diversity.

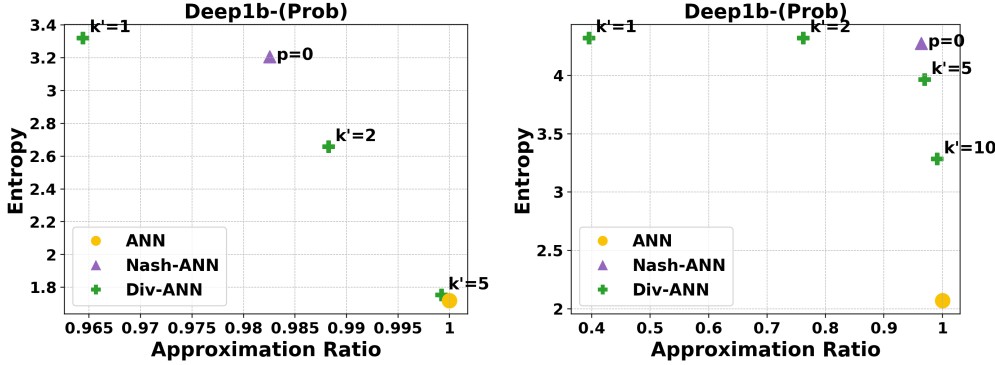

Figure 7: The plots show results on approximation ratio and entropy for **(Left)** $k = 10$; **(Right)** $k = 50$ in single-attribute setting on `Deep1b-(Prob)` dataset.

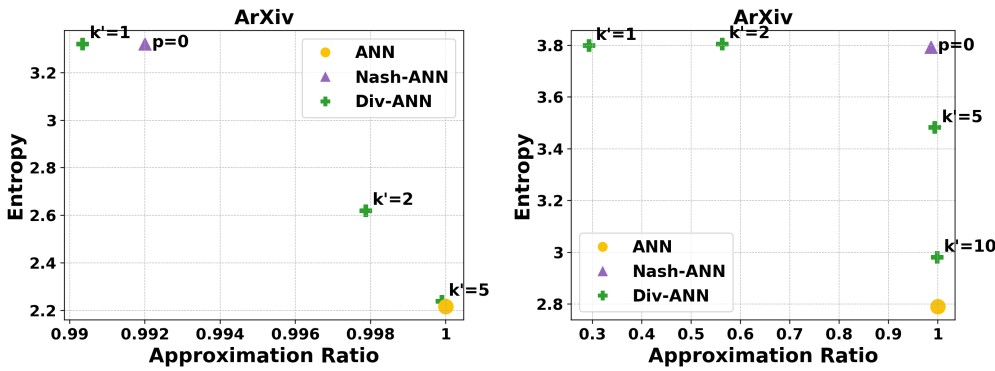

Figure 8: The plots show results on approximation ratio and entropy for **(Left)** $k = 10$; **(Right)** $k = 50$ in single-attribute setting on `ArXiv` dataset.

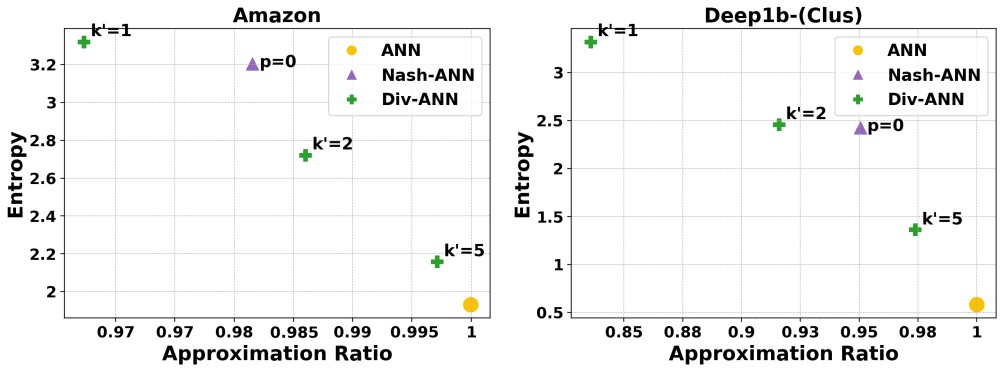

Figure 9: The plots show results on approximation ratio and entropy for **(Left)** $k = 10$; **(Right)** $k = 10$ in single-attribute setting on `Amazon` and `Deep1b-(Clus)` dataset.

### F.3.2 PERFORMANCE ON $p$-MEAN-ANN

In this set of experiments, we study the effect on trade-off between approximation ratio and entropy when the parameter $p$ in the $p$-NNS objective is varied over a range. Recall that the $p$-NNS problem with $p \to 0$ corresponds to the NaNNS problem and with $p = 1$ corresponds to the NNS problem. We experiment with values of $p \in \{-10, -1, -0.5, 0, 0.5, 1\}$ by running our algorithm

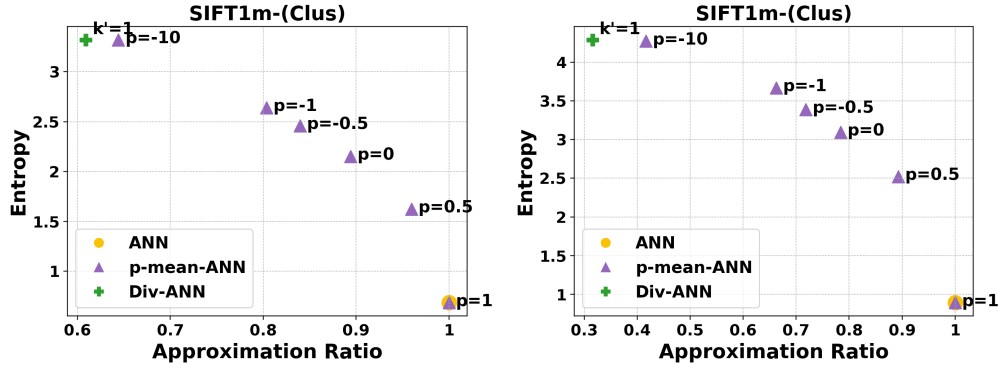

Figure 10: The plot reports the performance of $p$-mean-ANN with varying $p$ values for **(Left)** $k = 10$; **(Right)** $k = 50$ on Sift1m-(Clus) dataset in the single-attribute setting.

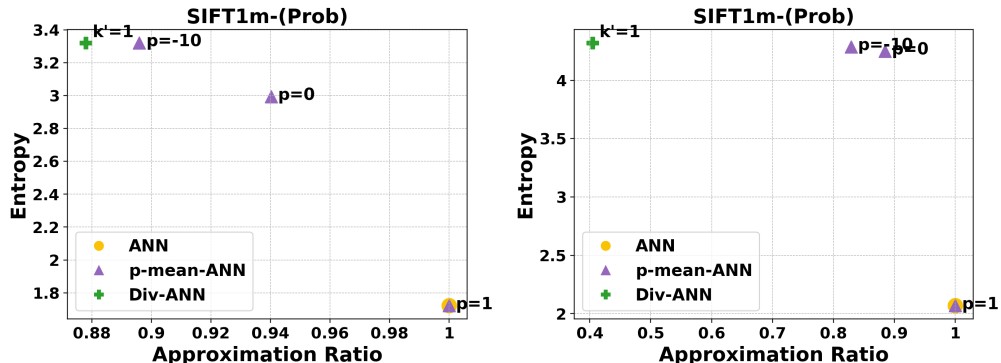

Figure 11: The plot reports the performance of $p$-mean-ANN with varying $p$ values for **(Left)** $k = 10$; **(Right)** $k = 50$ on Sift1m-(Prob) dataset in single-attribute setting. Note that for all other $p \in \{-1, -0.5, 0.5, 1\}$ the approximation ratio and entropy were extremely close to ones of $p = -10, 0$. To avoid clutter in plot we only show $p = -10, 0$.

$p$-mean-ANN (Algorithm 3) on the various datasets. The results are shown in Figures 10, 11, 12, 13, and 14. We observe across all datasets for both $k = 10$ and $k = 50$ that as $p$ decreases from 1, the entropy increases but approximation ratio decreases. This highlights the key intuition that as $p$ decreases, the behavior changes from utilitarian welfare ($p = 1$ aligns exactly with ANN) to egalitarian welfare (more attribute-diverse). In other words, the parameter $p$ allows us to smoothly interpolate between complete relevance (the standard NNS with $p = 1$) and complete diversity ($p \to -\infty$).

### F.3.3 Approximation Ratio Versus Inverse Simpson Index

We also report results (Figures 15, 16, 17, 18, 19, and 20) on approximation ratio versus inverse Simpson index for all the aforementioned datasets, comparing Nash-ANN with Div-ANN with various choices of quota parameter $k'$. The trends are similar to those for approximation ratio vs. entropy.

### F.3.4 Approximation Ratio Versus Distinct Attribute Count

We also report the number of distinct attributes appearing in the set of neighbors returned by different algorithms. Note that Div-ANN by design always returns a set where the number of distinct attributes is at least $(k/k')$. We plot approximation ratio versus number of distinct attributes and the results are shown in Figures 21, 22, 23, and 24. The results show that while Div-ANN with $k' = 1$ has high number of distinct attributes (by design) its approximation ratio is quite low. On the other

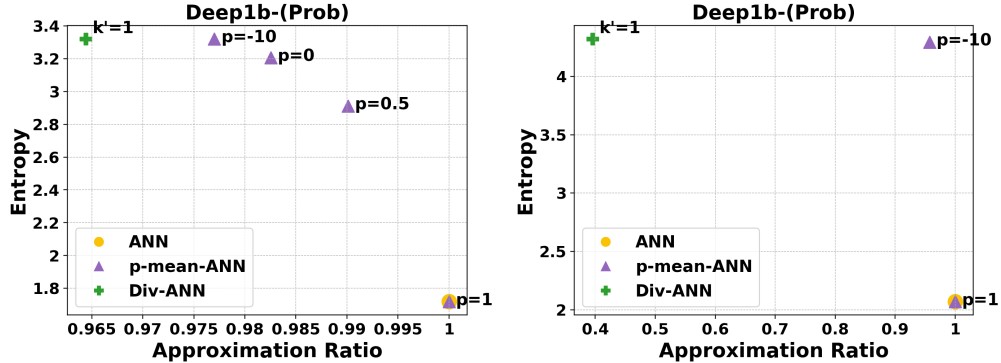

Figure 12: The plot reports the performance of $p$-mean-ANN with varying $p$ values for **(Left)** $k$ = 10; **(Right)** $k$ = 50 on Deep1b-(Prob) dataset in single-attribute setting. Note that for all other $p \in \{-1, -0.5, 0, 0.5, 1\}$ the approximation ratio and entropy were extremely close to ones of $p=-10$ in $k$ = 50. To avoid clutter in plot we only show $p = -10$. Due to same reasons we omit $p = -1, -0.5$ for $k$ = 10.

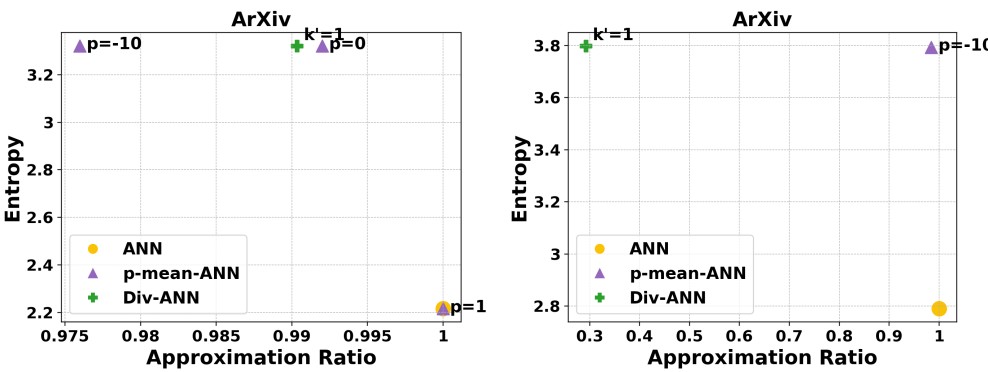

Figure 13: The plot reports the performance of $p$-mean-ANN with varying $p$ values for **(Left)** $k$ = 10; **(Right)** $k$ = 50 on ArXiv dataset in single-attribute setting. Note that for all other $p \in \{-1, -0.5, 0, 0.5\}$ the approximation ratio and entropy were extremely close to ones of $p=-10$. To avoid clutter in plot we only show $p = 0$.

hand, Nash-ANN has almost equal or slightly lower number of distinct attributes but achieves very high approximation ratio.

### F.3.5   RECALL VERSUS ENTROPY

We also report results for another popular relevance metric in the nearest neighbor search literature, namely, recall. The results for different datasets are shown in Figures 27, 28, 29, 30, 31, and 32. Note that as discussed earlier (Appendix F.1), recall can be a fragile metric when the goal is to balance between diversity and relevance. However, we still report recall to be consistent with prior literature and to demonstrate that Nash-ANN does not perform poorly. In fact, it is evident from the plots that Nash-ANN's recall value (relevance) surpasses that of Div-ANN with $k' = 1$ (most attribute diverse solution) while achieving almost similar entropy. As already noted, the approximation ratio for Nash-ANN remains sufficiently high, indicating that the retrieved set of neighbors lies within a reasonably good neighborhood of the true nearest neighbors of a given query.

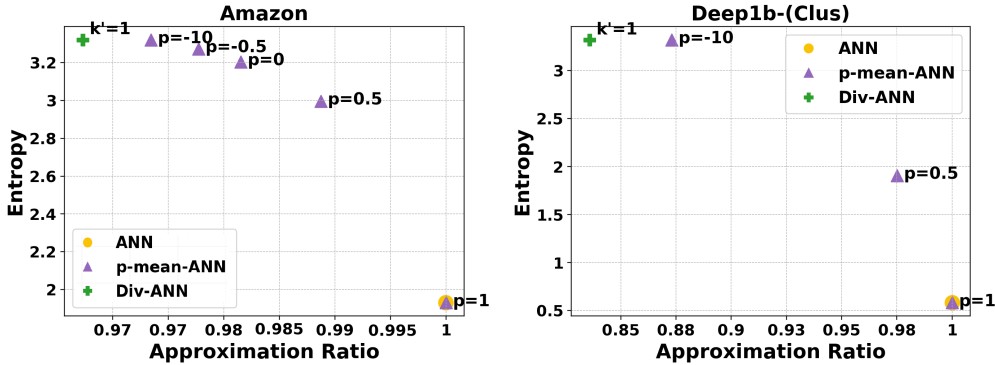

Figure 14: The plot reports the performance of $p$-mean-ANN with varying $p$ values for **(Left)** $k = 10$ on `Amazon`; **(Right)** $k = 10$ on `Deep1b-(Clus)` dataset in single-attribute setting. Note that for all other $p \in \{-1, -0.5, 0, 0.5\}$ the approximation ratio and entropy were extremely close to ones of $p=-10$ in `Deep1b-(Clus)` dataset. To avoid clutter in plot we only show $p = 0$. Due to similar reasons we omit $p = -1$ in `Amazon` dataset as it was close to $p = \{-10, -0.5\}$

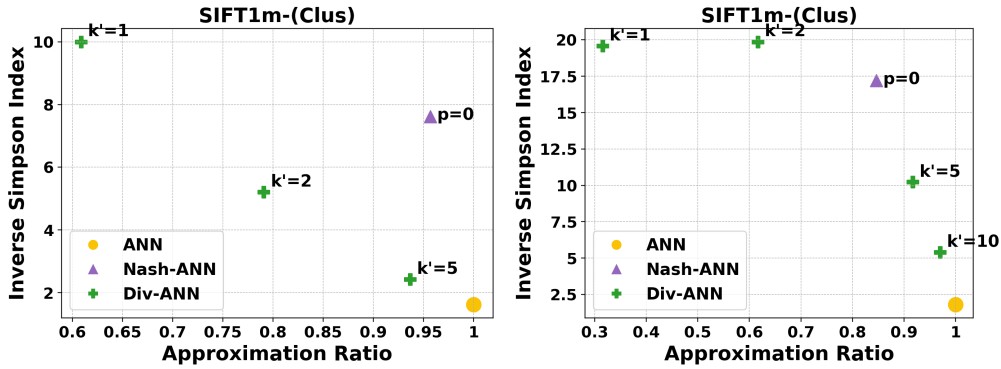

Figure 15: The plots show results on approximation ratio and inverse Simpson index for **(Left)** $k = 10$; **(Right)** $k = 50$ in single-attribute setting on `Sift1m-(Clus)` dataset.

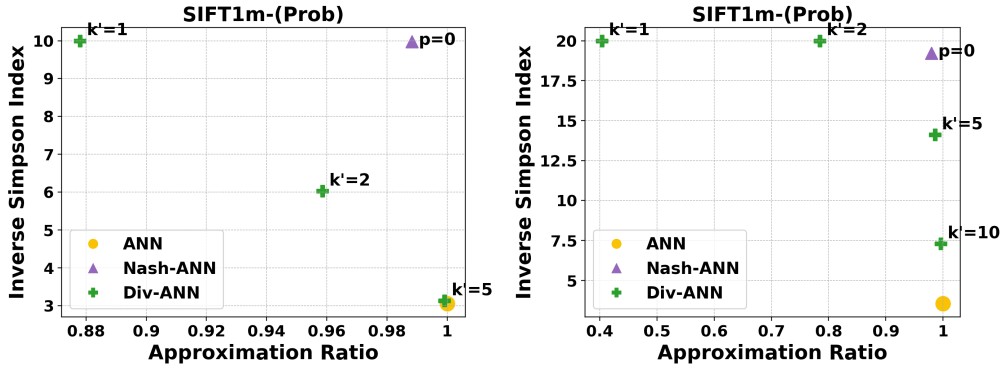

Figure 16: The plots show results on approximation ratio and inverse Simpson index for **(Left)** $k = 10$; **(Right)** $k = 50$ in single-attribute setting on `Sift1m-(Prob)` dataset.

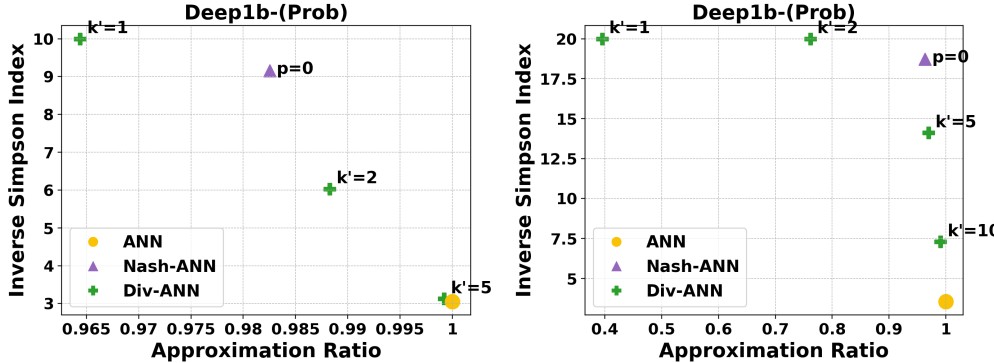

Figure 17: The plots show results on approximation ratio and inverse Simpson index for **(Left)** $k = 10$; **(Right)** $k = 50$ in single-attribute setting on `Deep1b-(Prob)` dataset.

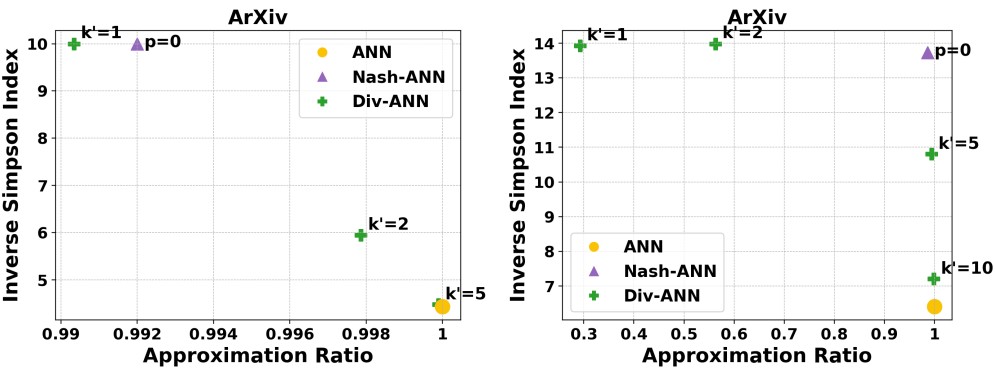

Figure 18: The plots show results on approximation ratio and inverse Simpson index for **(Left)** $k = 10$; **(Right)** $k = 50$ in single-attribute setting on `ArXiv` dataset.

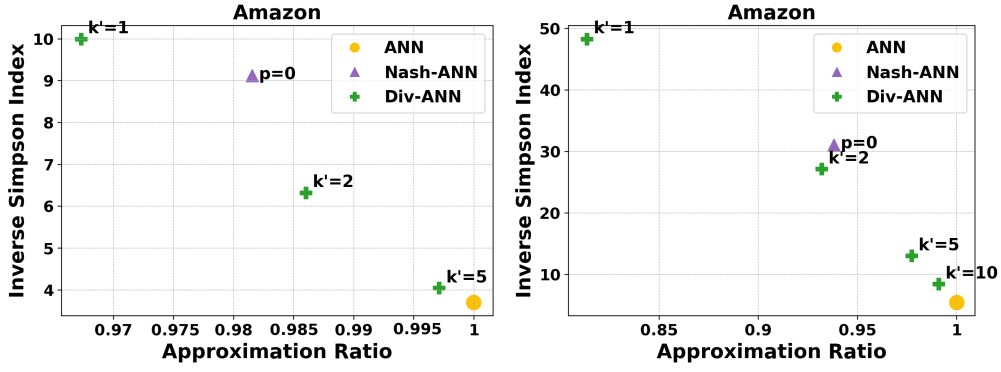

Figure 19: The plots show results on approximation ratio and inverse Simpson index for **(Left)** $k = 10$; **(Right)** $k = 50$ in single-attribute setting on `Amazon` dataset.

## F.4 BALANCING RELEVANCE AND DIVERSITY: MULTI-ATTRIBUTE SETTING

Recall that our welfarist formulation seamlessly extends to the multi-attribute setting. In Section 4, we discussed the performance of `Multi Nash-ANN` and `Multi Div-ANN` on `Sift1m-(Clus)`, where each input vector was associated with four attributes. In this section, we repeat the same set of experiments on one of the real-world dataset, namely `ArXiv`, which nat-

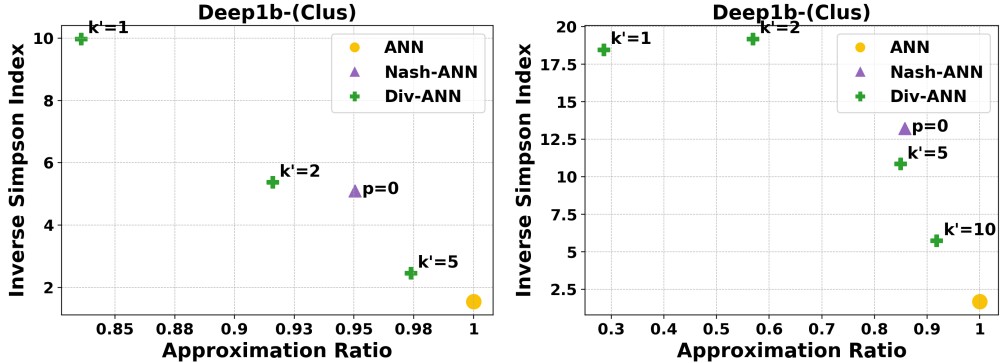

Figure 20: The plots show results on approximation ratio and inverse Simpson index for **(Left)** $k = 10$; **(Right)** $k = 50$ in single-attribute setting on `Deep1b-(Clus)` dataset.

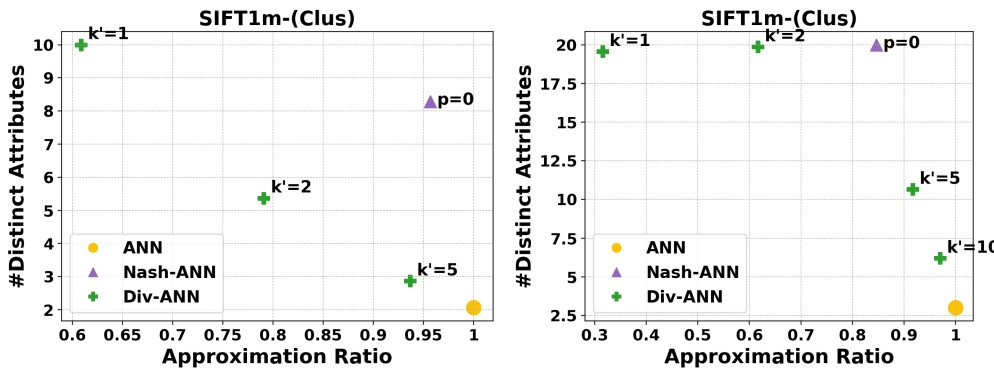

Figure 21: The plots show results on approximation ratio and distinct counts for **(Left)** $k = 10$; **(Right)** $k = 50$ in single-attribute setting on `Sift1m-(Clus)` dataset.

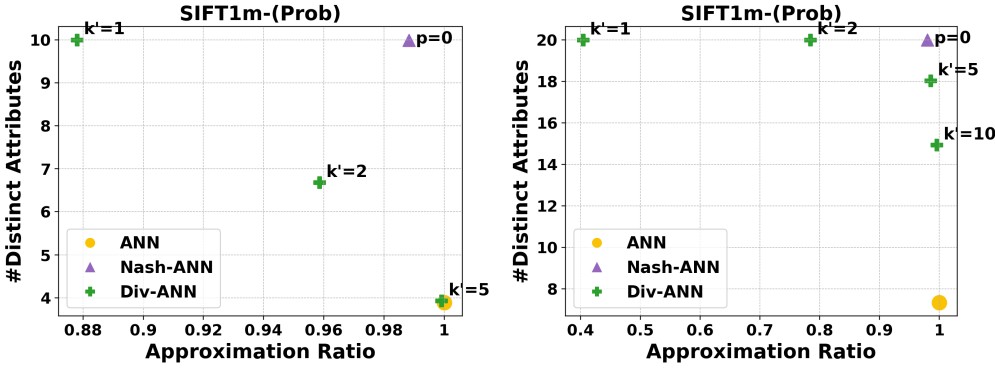

Figure 22: The plots show results on approximation ratio and distinct counts for **(Left)** $k = 10$; **(Right)** $k = 50$ in single-attribute setting on `Sift1m-(Prob)` dataset.

urally contains two partition sets of the attributes ($m = 2$; see Appendix F.1, Diversity Metrics): update year ($|C_1| = 14$) and paper category ($|C_2| = 16$). Therefore, $c = |C_1| + |C_2| = 30$. The results for $k = 50$ are presented in Figure 33. Note that in each plot we restrict the entropy to one of the attribute partitions ($C_1$ and $C_2$) so that the diversity within a partition set can be understood from these plots. The results indicate that `Multi Nash-ANN` achieves an approximation ratio very close to one while maintaining entropy levels comparable to `Multi Div-ANN` with $k' = 1$ or 2 for

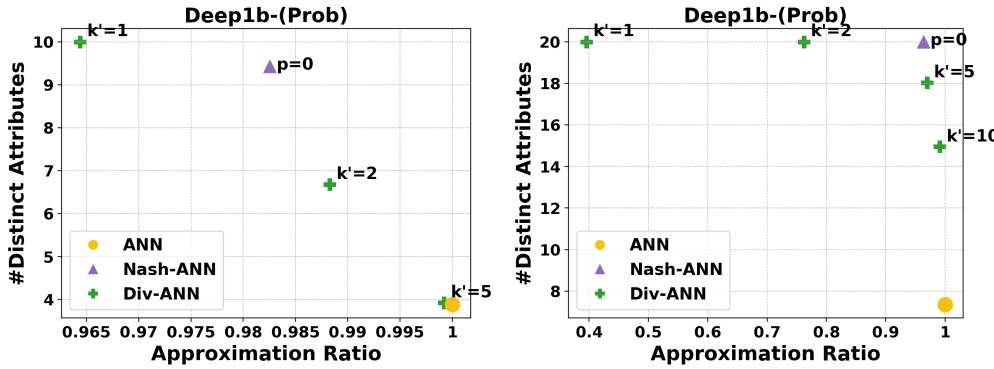

Figure 23: The plots show results on approximation ratio and distinct counts for **(Left)** $k = 10$; **(Right)** $k = 50$ in single-attribute setting on `Deep1b-(Prob)` dataset.

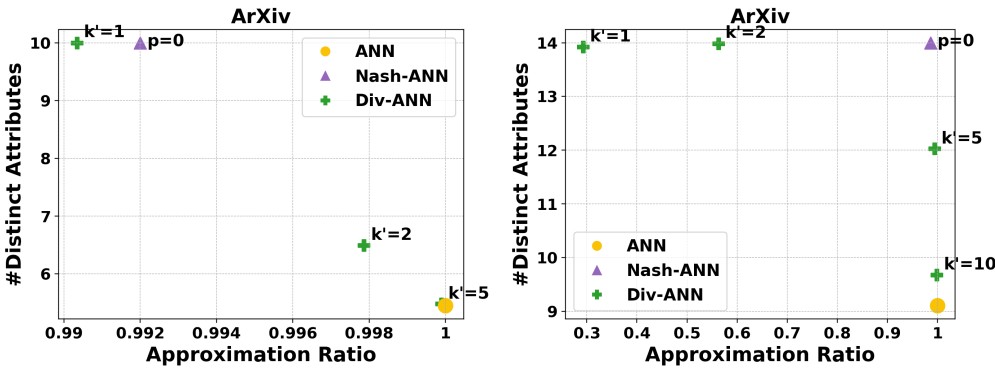

Figure 24: The plots show results on approximation ratio and distinct counts for **(Left)** $k = 10$; **(Right)** $k = 50$ in single-attribute setting on `ArXiv` dataset.

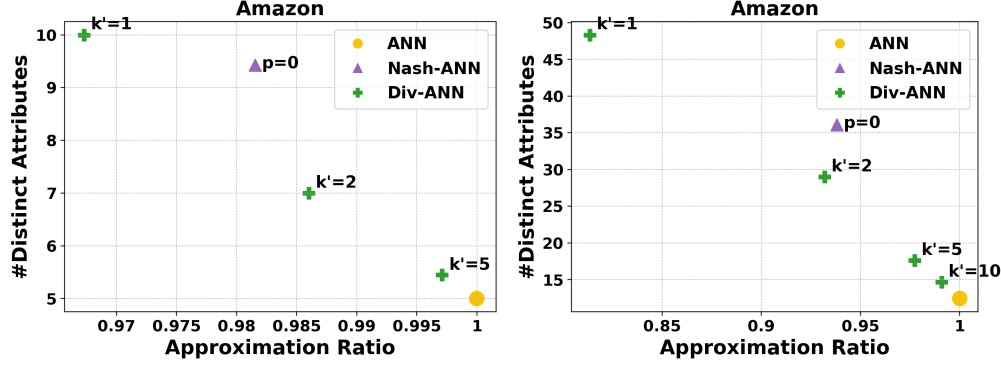

Figure 25: The plots show results on approximation ratio and distinct counts for **(Left)** $k = 10$; **(Right)** $k = 50$ in single-attribute setting on `Amazon` dataset.

both the attribute partition sets. In fact `Multi Nash-ANN` Pareto dominates `Multi Div-ANN` with $k' = 5$.

We also study the effect of varying $p$ in $p$-NNS problem in the multi-attribute setting. The results for performance of `Multi p-mean-ANN` (an analogue of `Multi Nash-ANN`) for $p \in \{-10, -1, -0.5, 0, 0.5, 1\}$ are shown in Figures 34 and 35. Interestingly, we observe that with de-

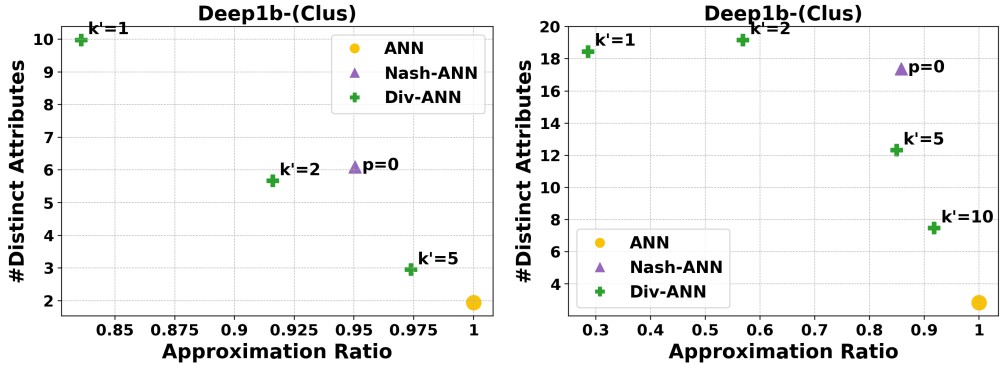

Figure 26: The plots show results on approximation ratio and distinct counts for **(Left)** $k = 10$; **(Right)** $k = 50$ in single-attribute setting on `Deep1b-(Clus)` dataset.

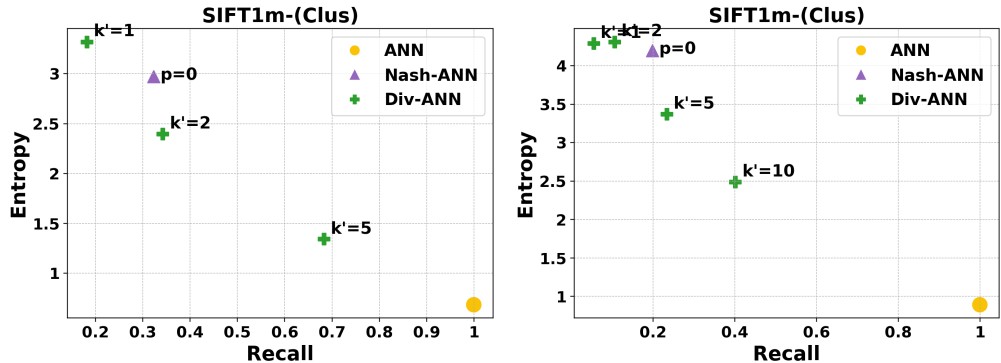

Figure 27: The plots show results on recall and entropy for **(Left)** $k = 10$ ; **(Right)** $k = 50$ in single-attribute setting on `Sift1m-(Clus)` dataset.

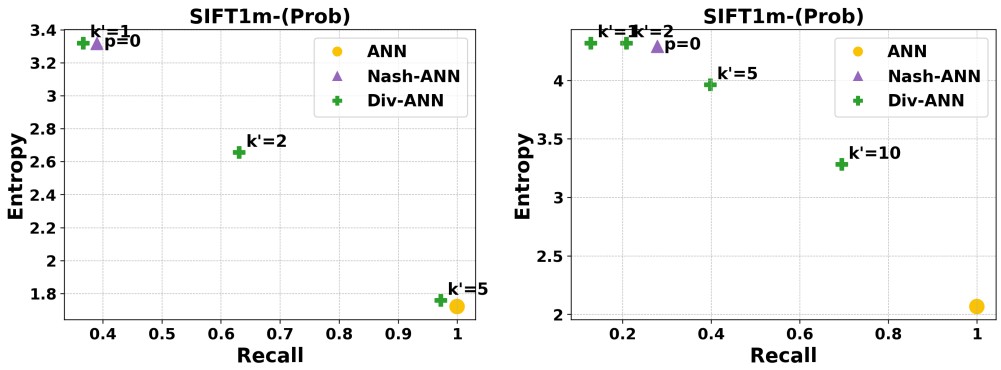

Figure 28: The plots show results on recall and entropy for **(Left)** $k = 10$ ; **(Right)** $k = 50$ in single-attribute setting on `Sift1m-(Prob)` dataset.

creasing $p$, the entropy (across $C_1$ or $C_2$) increases but the approximation ratio remains nearly the same and very close to 1. On the other hand, `Multi Div-ANN` with $k' = 1$ has very low approximation ratio. In fact, `Multi p-mean-ANN` with $p = -1$ and $-10$ Pareto dominates `Multi Div-ANN` with $k' = 1$.

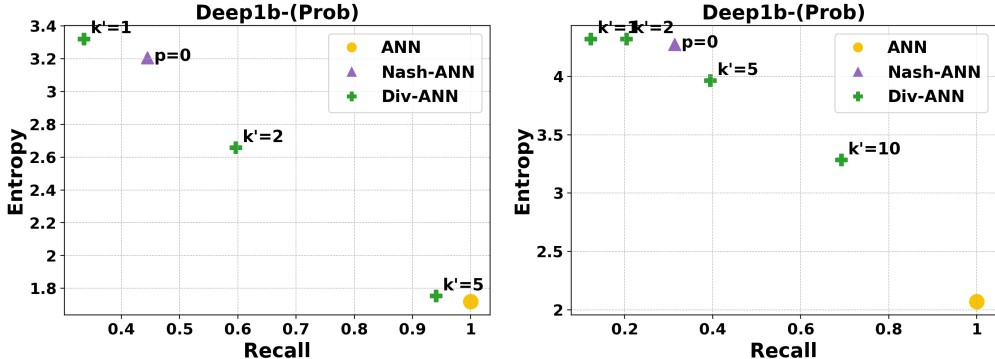

Figure 29: The plots show results on recall and entropy for **(Left)** $k = 10$ ; **(Right)** $k = 50$ in single-attribute setting on `Deep1b-(Prob)` dataset.

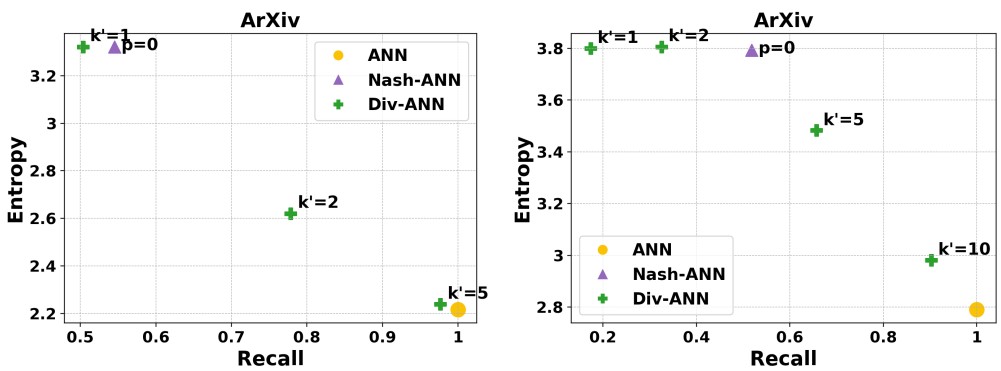

Figure 30: The plots show results on recall and entropy for **(Left)** $k = 10$; **(Right)** $k = 50$ in single-attribute setting on `ArXiv` dataset.

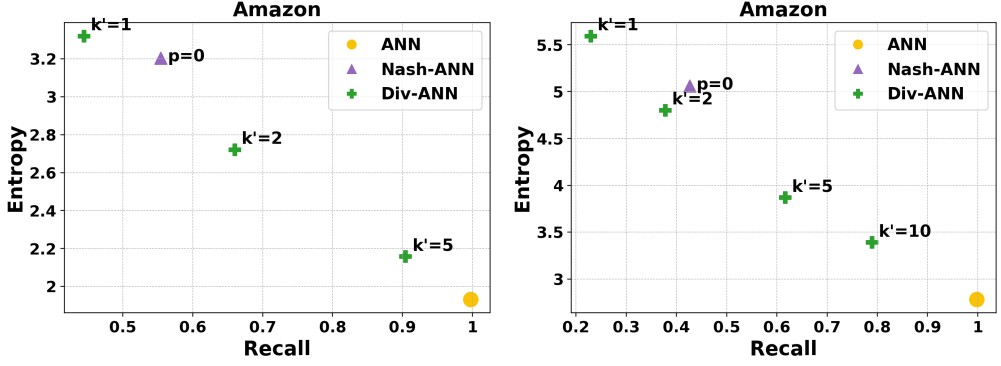

Figure 31: The plots show results on recall and entropy for **(Left)** $k = 10$ ; **(Right)** $k = 50$ in single-attribute setting on `Amazon` dataset.

## F.5 A FASTER HEURISTIC FOR THE SINGLE ATTRIBUTE SETTING: $p$-FETCHUNION-ANN

In this section, we empirically study a faster heuristic algorithm for NSW and $p$-mean welfare formulations. Specifically, the heuristic—called $p$-FetchUnion-ANN—first fetches a sufficiently large candidate set of vectors (irrespective of their attributes) using the ANN algorithm. Then, it applies the Nash (or $p$-mean) selection (similar to Line 5 in Algorithm 1 or Lines 6-8 in Algorithm 3)

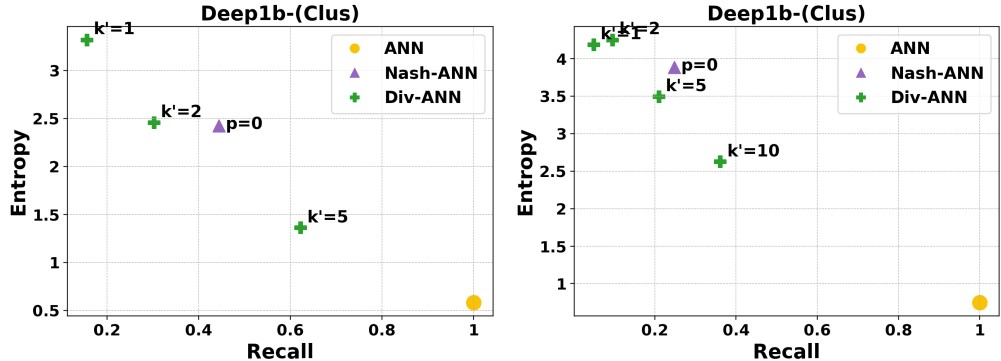

Figure 32: The plots show results on recall and entropy for **(Left)** $k = 10$ ; **(Right)** $k = 50$ in single-attribute setting on `Deep1b-(Clus)` dataset.

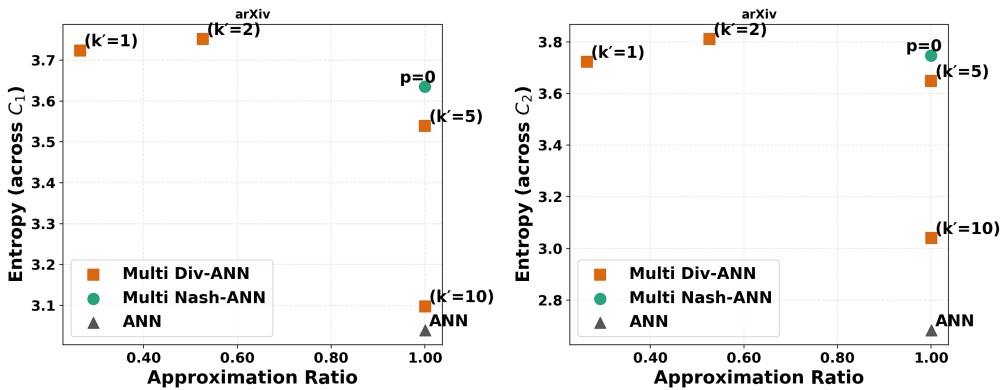

Figure 33: The plot shows approximation ratio and entropy trade-off on `ArXiv` dataset in multi-attribute setting.

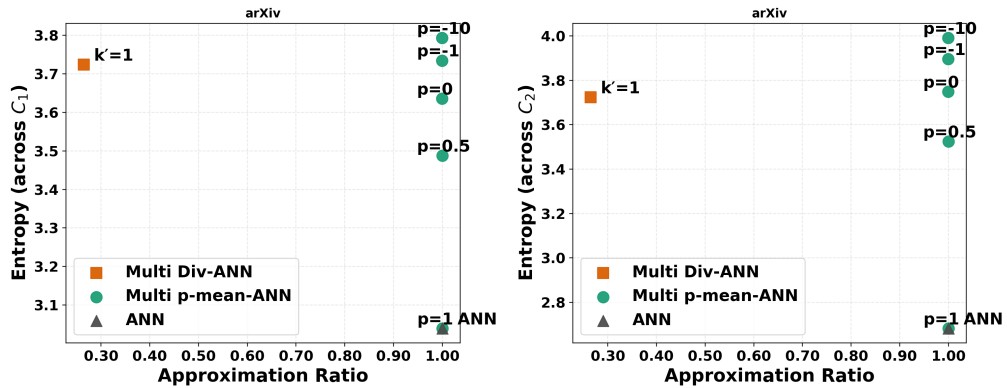

Figure 34: The plot reports the performance of $p$-mean-ANN with varying $p$ values for $k = 50$ in multi-attribute setting on `ArXiv` dataset.

within this set. That is, instead of starting out with $k$ neighbors for each $\ell \in [c]$ (as in Line 1 of Algorithm 1), the alternative here is to work with sufficiently many neighbors from the set $\cup_{\ell=1}^{c} D_\ell$.

We empirically show (in Tables 2 to 7) that this heuristic consistently achieves performance comparable to `p-Mean-ANN` across nearly all datasets and evaluation metrics. Since

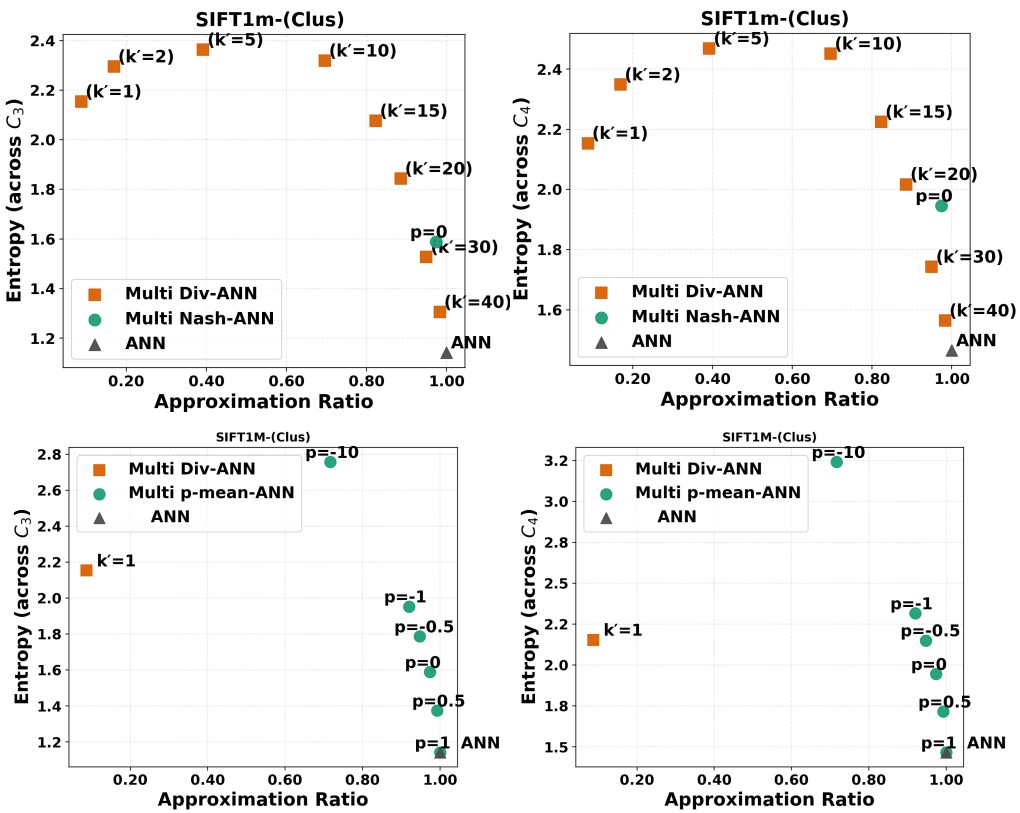

Figure 35: **Top Row**: The plots show approximation ratio versus entropy trade-offs for `Nash-ANN` against `Div-ANN` with varying values of $k'$ for attribute class $C_3$ **(left)** and $C_4$ **(right)**. **Bottom Row**: The plots show approximation ratio versus entropy trade-offs for $p$-mean-ANN, as $p$ varies, across attribute classes $C_3$ **(left)** and $C_4$ **(right)**. Both the rows correspond to $k = 50$ on `Sift1m-(Clus)` dataset in the multi-attribute setting.

$p$-`FetchUnion-ANN` retrieves a larger pool of vectors with high similarity, it achieves improved approximation ratio over $p$-`Mean-ANN`. This trend is evident in two datasets, namely `Deep1b-(Clus)` and `Sift1m-(Clus)`, although it comes at the cost of reduced entropy, which can be explained by the fact that in restricting its search to an initially fetched large pool of vectors, $p$-`FetchUnion-ANN` may miss out on a more diverse solution that exists over the entire dataset. Another important aspect of $p$-`FetchUnion-ANN` is that, because it retrieves all neighbors from the union at once, the heuristic delivers substantially higher throughput (measured as queries per second, QPS) and therefore lower latency. The results validating these findings are reported in Tables 8 and 9 for the `Sift1m-(Clus)` and `Amazon` datasets, respectively. In particular, it serves almost **10**× more queries on `Sift1m-(Clus)` and **3**× more queries on `Amazon` dataset. The latency values exhibit a similar trend with reductions of similar magnitude. In summary, these observations position the heuristic as a notably fast method for NaNNS and $p$-NNS, particularly when $c$ is large.

## F.6 COMPARISON WITH DIVERSITY BASED ON MINIMUM PAIRWISE DISTANCE

The current work studies diversity across specified attributes. Our comparisons focus on the hard-constrained approach, as it is the essential prior work that also addresses attribute diversity (Anand et al., 2025). In NNS, we found two additional threads of work on diversity and fairness. However, the normative concepts in these prior works are not attribute-centric. The diversity criterion developed in (Abbar et al., 2013a;b) requires an additional distance function (and not specifically attributes) to express diversity. The objective in (Abbar et al., 2013a) and (Abbar et al., 2013b) is to return, from the set of vectors that are at most $r$ distance away from the query, a subset of $k$ vectors that maximizes the minimum pairwise diversity distance.

| Metric | Algorithm | $p=-10$ | $p=-1$ | $p=-0.5$ | $p=0$ | $p=0.5$ | $p=1$ |
|---|---|---|---|---|---|---|---|
| Approx. Ratio | $p$-Mean-ANN | 0.865±0.045 | 0.909±0.029 | 0.922±0.027 | 0.938±0.023 | 0.961±0.018 | 1.000±0.000 |
| | $p$-FetchUnion-ANN | 0.907±0.033 | 0.912±0.030 | 0.921±0.027 | 0.935±0.024 | 0.958±0.019 | 1.000±0.000 |
| | ANN | | | 1.000±0.000 | | | |
| | Div-ANN ($k'$=1) | | | 0.813±0.053 | | | |
| Entropy | $p$-Mean-ANN | 5.644±0.000 | 5.382±0.135 | 5.252±0.153 | 5.058±0.178 | 4.687±0.227 | 2.782±0.684 |
| | $p$-FetchUnion-ANN | 5.364±0.156 | 5.333±0.149 | 5.261±0.150 | 5.099±0.171 | 4.736±0.221 | 2.782±0.684 |
| | ANN | | | 2.782±0.684 | | | |
| | Div-ANN ($k'$=1) | | | 5.594±0.049 | | | |

Table 2: Comparison of performance across $p$ values for `Amazon` at $k=50$.

| Metric | Algorithm | $p=-10$ | $p=-1$ | $p=-0.5$ | $p=0$ | $p=0.5$ | $p=1$ |
|---|---|---|---|---|---|---|---|
| Approx. Ratio | $p$-Mean-ANN | 0.985±0.010 | 0.985±0.010 | 0.985±0.010 | 0.986±0.009 | 0.989±0.008 | 1.000±0.001 |
| | $p$-FetchUnion-ANN | 0.989±0.007 | 0.989±0.007 | 0.989±0.007 | 0.990±0.006 | 0.991±0.006 | 1.000±0.001 |
| | ANN | | | 1.000±0.001 | | | |
| | Div-ANN ($k'$=1) | | | 0.293±0.007 | | | |
| Entropy | $p$-Mean-ANN | 3.793±0.002 | 3.793±0.002 | 3.793±0.002 | 3.793±0.002 | 3.793±0.002 | 2.790±0.510 |
| | $p$-FetchUnion-ANN | 3.704±0.167 | 3.704±0.166 | 3.704±0.166 | 3.704±0.166 | 3.704±0.166 | 2.790±0.510 |
| | ANN | | | 2.790±0.510 | | | |
| | Div-ANN ($k'$=1) | | | 3.799±0.029 | | | |

Table 3: Comparison of performance across $p$ values for `ArXiv` at $k=50$.

One way to adapt the distance-based diversity notion to the single-attribute setting is by considering Hamming distances. Specifically, any two vectors with the different attributes are set to have a diversity distance of 1 and vectors with the same attribute are set at 0 distance. The key issue with this adaptation is that the solutions it finds will always be at the extremes (See Figure 36, and 37 for $k = 10$, and $k= 50$ respectively). That is, if the prefetched pool consists of more than $k$ different attributes, then the subset picked by this approach will consist of one vector from each attribute (i.e., it is the same as the one via the hard-constraint formulation with $k' = 1$). Otherwise, if the prefetched pool consists of less than $k$ different attributes, then any size-$k$ subset will have minimum diversity distance 0. Hence, in this case, the selected subset can be arbitrary; say, the $k$ most similar vectors. This observation underscores that the notion from (Abbar et al., 2013a) and (Abbar et al., 2013b) (with Hamming distance as the diversity function) does not provide a comparison point stronger than `Div-ANN` with different values of $k'$.

We also note that, in the multi-attribute setting, the Hamming distance based diversity distance function is quite ad hoc. This restricts the use of the distance-based diversity notion in multi-attribute settings.

| Metric | Algorithm | $p=-10$ | $p=-1$ | $p=-0.5$ | $p=0$ | $p=0.5$ | $p=1$ |
|---|---|---|---|---|---|---|---|
| Approx. Ratio | $p$-Mean-ANN | 0.784±0.071 | 0.815±0.065 | 0.831±0.063 | 0.858±0.060 | 0.904±0.049 | 1.000±0.000 |
| | $p$-FetchUnion-ANN | 0.958±0.033 | 0.961±0.030 | 0.962±0.029 | 0.963±0.028 | 0.968±0.024 | 1.000±0.000 |
| | ANN | | | | 1.000±0.000 | | |
| | Div-ANN ($k'=1$) | | | | 0.286±0.041 | | |
| Entropy | $p$-Mean-ANN | 4.293±0.000 | 4.200±0.052 | 4.105±0.091 | 3.887±0.155 | 3.349±0.267 | 0.746±0.717 |
| | $p$-FetchUnion-ANN | 2.101±1.214 | 2.101±1.214 | 2.099±1.212 | 2.095±1.207 | 2.068±1.179 | 0.746±0.717 |
| | ANN | | | | 0.746±0.717 | | |
| | Div-ANN ($k'=1$) | | | | 4.191±0.234 | | |

Table 4: Comparison of performance across $p$ values for `Deep1b-(Clus)` at $k=50$.

| Metric | Algorithm | $p=-10$ | $p=-1$ | $p=-0.5$ | $p=0$ | $p=0.5$ | $p=1$ |
|---|---|---|---|---|---|---|---|
| Approx. Ratio | $p$-Mean-ANN | 0.958±0.019 | 0.960±0.017 | 0.961±0.016 | 0.963±0.014 | 0.969±0.010 | 1.000±0.000 |
| | $p$-FetchUnion-ANN | 0.958±0.019 | 0.960±0.017 | 0.961±0.016 | 0.963±0.014 | 0.969±0.010 | 1.000±0.000 |
| | ANN | | | | 1.000±0.000 | | |
| | Div-ANN ($k'=1$) | | | | 0.395±0.010 | | |
| Entropy | $p$-Mean-ANN | 4.293±0.000 | 4.292±0.005 | 4.288±0.010 | 4.275±0.020 | 4.217±0.068 | 2.070±0.208 |
| | $p$-FetchUnion-ANN | 4.293±0.001 | 4.292±0.005 | 4.288±0.010 | 4.275±0.020 | 4.217±0.068 | 2.070±0.207 |
| | ANN | | | | 2.070±0.207 | | |
| | Div-ANN ($k'=1$) | | | | 4.322±0.002 | | |

Table 5: Comparison of performance across $p$ values for `Deep1b-(Prob)` at $k=50$.

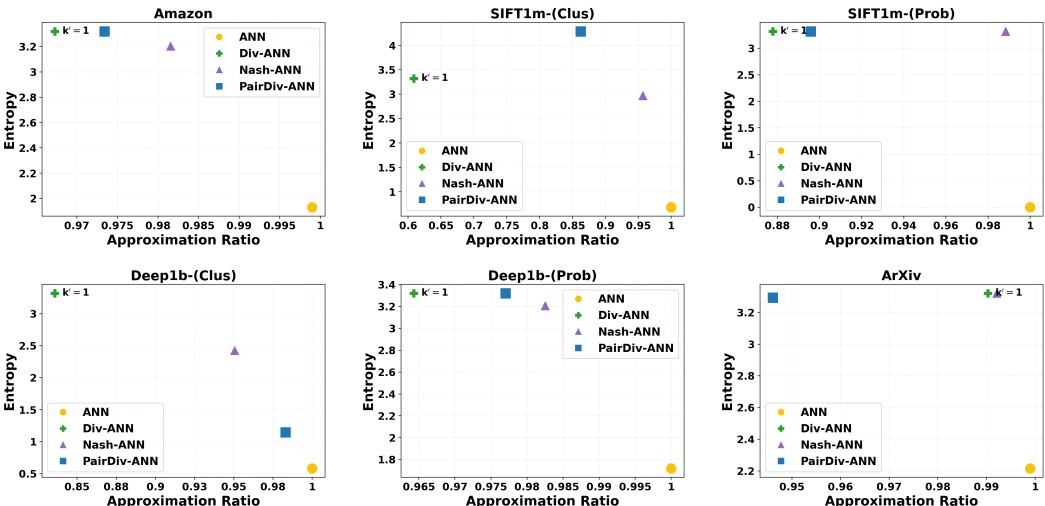

Figure 36: Comparison of ANN Variants on different datasets for $p=0$ and $k=10$.

| Metric | Algorithm | $p=-10$ | $p=-1$ | $p=-0.5$ | $p=0$ | $p=0.5$ | $p=1$ |
|---|---|---|---|---|---|---|---|
| Approx. Ratio | $p$-Mean-ANN | 0.749±0.051 | 0.810±0.045 | 0.812±0.043 | 0.846±0.036 | 0.932±0.028 | 1.000±0.000 |
| | $p$-FetchUnion-ANN | 0.979±0.014 | 0.980±0.013 | 0.980±0.013 | 0.981±0.012 | 0.983±0.011 | 1.000±0.000 |
| | ANN | | | | 1.000±0.000 | | |
| | Div-ANN ($k'=1$) | | | | 0.315±0.021 | | |
| Entropy | $p$-Mean-ANN | 4.285±0.012 | 4.293±0.002 | 4.293±0.001 | 4.197±0.045 | 3.506±0.275 | 0.892±0.663 |
| | $p$-FetchUnion-ANN | 2.235±0.802 | 2.238±0.802 | 2.239±0.802 | 2.239±0.802 | 2.231±0.800 | 0.892±0.663 |
| | ANN | | | | 0.892±0.663 | | |
| | Div-ANN ($k'=1$) | | | | 4.289±0.053 | | |

Table 6: Comparison of performance across $p$ values for `Sift1m-(Clus)` at $k=50$.

| Metric | Algorithm | $p=-10$ | $p=-1$ | $p=-0.5$ | $p=0$ | $p=0.5$ | $p=1$ |
|---|---|---|---|---|---|---|---|
| Approx. Ratio | $p$-Mean-ANN | 0.975±0.010 | 0.977±0.008 | 0.979±0.008 | 0.980±0.008 | 0.982±0.006 | 1.000±0.000 |
| | $p$-FetchUnion-ANN | 0.975±0.010 | 0.977±0.008 | 0.979±0.008 | 0.980±0.008 | 0.982±0.006 | 1.000±0.000 |
| | ANN | | | | 1.000±0.000 | | |
| | Div-ANN ($k'{=}1$) | | | | 0.404±0.004 | | |
| Entropy | $p$-Mean-ANN | 4.292±0.006 | 4.292±0.003 | 4.293±0.002 | 4.293±0.002 | 4.269±0.020 | 2.068±0.205 |
| | $p$-FetchUnion-ANN | 4.292±0.006 | 4.292±0.003 | 4.293±0.002 | 4.293±0.003 | 4.269±0.020 | 2.068±0.205 |
| | ANN | | | | 2.068±0.205 | | |
| | Div-ANN ($k'{=}1$) | | | | 4.322±0.005 | | |

Table 7: Comparison of performance across $p$ values for Sift1m-(Prob) at $k=50$.

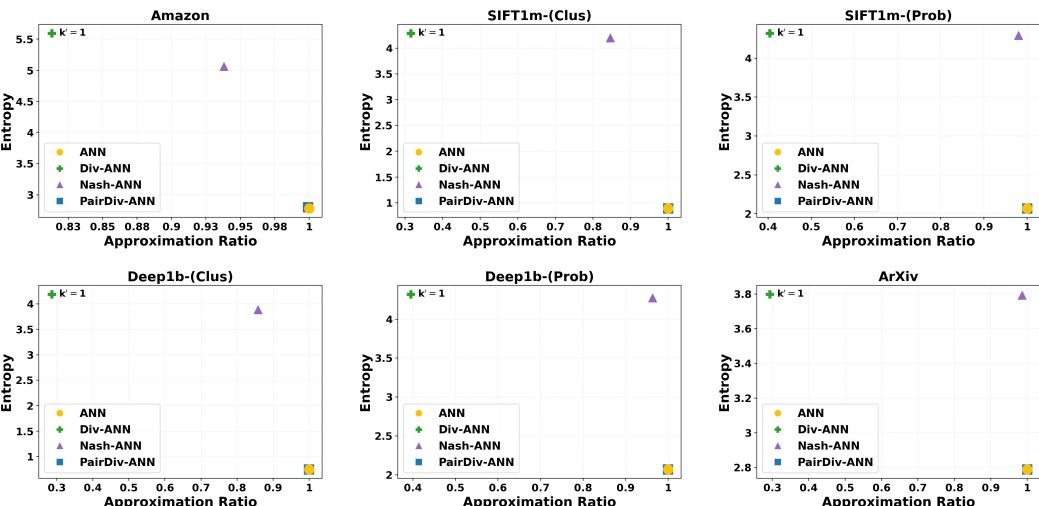

Figure 37: Comparison of ANN Variants on different datasets for $p=0$ and $k=50$. We set

| Metric | Algorithm | $p=-10$ | $p=-1$ | $p=-0.5$ | $p=0$ | $p=0.5$ | $p=1$ |
|---|---|---|---|---|---|---|---|
| Query per Second | $p$-Mean-ANN | 120.86 | 115.78 | 107.01 | 135.98 | 122.59 | 122.59 |
| | $p$-FetchUnion-ANN | 1324.53 | 1324.62 | 1337.28 | 1442.03 | 1443.38 | 1327.03 |
| Latency ($\mu s$) | $p$-Mean-ANN | 264566.00 | 276129.00 | 298804.00 | 230318.00 | 235144.00 | 260800.00 |
| | $p$-FetchUnion-ANN | 24133.80 | 24134.00 | 23907.00 | 22170.20 | 22149.30 | 28990.40 |
| 99.9th percentile of Latency | $p$-Mean-ANN | 484601.00 | 513036.00 | 478821.00 | 477925.00 | 482777.00 | 479132.00 |
| | $p$-FetchUnion-ANN | 52943.40 | 53474.70 | 54283.40 | 56128.70 | 53082.20 | 24088.70 |

Table 8: Comparison of performance on Queries per second and Latency across $p$ values on Sift1m-(Clus) dataset for $k=50$.

| Metric | Algorithm | $p=-10$ | $p=-1$ | $p=-0.5$ | $p=0$ | $p=0.5$ | $p=1$ |
|---|---|---|---|---|---|---|---|
| Query per Second | $p$-Mean-ANN | 198.08 | 195.97 | 199.08 | 179.03 | 171.22 | 189.31 |
| | $p$-FetchUnion-ANN | 620.27 | 610.62 | 551.02 | 608.76 | 572.57 | 591.76 |
| Latency ($\mu s$) | $p$-Mean-ANN | 161385.00 | 163121.00 | 160503.00 | 178555.00 | 186780.00 | 168856.00 |
| | $p$-FetchUnion-ANN | 51539.90 | 52362.30 | 58028.60 | 52521.60 | 55843.70 | 54030.80 |
| 99.9th percentile of Latency | $p$-Mean-ANN | 433434.00 | 407151.00 | 418147.00 | 421725.00 | 475474.00 | 404477.00 |
| | $p$-FetchUnion-ANN | 146632.00 | 144989.00 | 145620.00 | 145657.00 | 143627.00 | 146464.00 |

Table 9: Comparison of performance on Queries per second and Latency across $p$ values on Amazon dataset for $k=50$.

