# OpenReview forum: "Welfarist Formulations for Diverse Similarity Search"
_ICLR.cc/2026/Conference — ICLR 2026 Poster_

### Official Review · Reviewer_jBed · 2025-10-31

**Soundness:** 4
**Presentation:** 2
**Contribution:** 4
**Rating:** 10
**Confidence:** 4

**Summary:**

The paper proposes a new problem setting and algorithms for the problem of similarity search with diversity. An interesting formulation is given where similarity and diversity are combined in one objective. The objective is tunable so as to cover some extreme cases: only relevance, only diversity, and mixing the two using hyperparameters. Diversity is measured with respect to a set of predefined attributes. The paper makes a distinction when an item has a single attribute and when it has many attributes. In the former case an exact algorithm is given. In the latter case the problem is shown to be NP-hard and the greedy method is shown to give a (1-1/e) approximation. Also approximate similarity search can be incorporate to trade efficiency with solution quality.

**Strengths:**

S1. The problem formulation is quite elegant, providing a way to introduce diversity into the classic similarity search problem.
S2. Rigorous analysis and strong theoretical results.
S3. Good motivation and discussion.
S4. Thorough experiments, evaluating different aspects of the problem setting and the algorithms.

**Weaknesses:**

W1. The paper contains lots of discussion and motivation. On the one hand this is a strength, on the other hand, it gets quite repetitive and tiring. I would prefer a shorter discussion, avoiding excessive repetition, and presenting more technical results in the main paper instead of the appendix.
W2. In the intro, the use of geometric mean is motivated by the argument that the geometric mean is larger than the min value of an item and smaller than the arithmetic mean. This fact alone is not a strong motivation, as the fact that the values are different does not necessarily imply that the solutions of the respective problems will be different.
W3. The formulation is elegant, however, it needs to assume the presence of attributes, which might not always be the case. Another simpler notion of diversity, not covered by the setting of the paper, can be defined by employing the same similarity function used to compare the query with the data items. Here, we would require k items that are similar enough to the query but not similar to each other.

**Questions:**

Please answer the points mentioned above. Additionally, I noticed that the author(s) use the terms "diversity" and "fairness" almost interchangeably. I would like to know the views of the author(s) on this aspect.

---

> ### Author Response · Authors · 2025-11-18
> **Response for Reviewer jBed (Part I - [Ques], [W1], [W2])**
>
> We thank the reviewer for their thorough review and for appreciating the elegance of our formulation, as well as the strength of our results. Below, we will address the points **([Ques], [W1], [W2])** raised in the review. Kindly refer to the next comment for **[W3]**.
>
> &nbsp;
>
> - **[Ques]**: We primarily use the term “diversity” when discussing nearest neighbor search and use “fairness” in the context of welfare functions. Our aim was to convey that once we connect diversity in nearest neighbor search and fairness in welfare contexts, the two concepts are equivalent.
>
>     &nbsp;
>
> - **[W1]**: We appreciate the editorial feedback. The detailed introduction was somewhat intentional. The work makes novel conceptual contributions by bridging two distinct areas: nearest neighbor search and welfare economics. Hence, we felt it is important to articulate the reasoning behind the formulation, rather than just stating it (as in Definition 1). Such a self-contained exposition aims to ensure that a broad audience (including readers from both NNS and welfare economics communities) can appreciate the work.
>
>     &nbsp;
>
>     Still, in light of the reviewer’s feedback, we have abridged the introduction by removing repetitions, wherever possible. We have also moved proof of one of our main results (Theorem 1) into the main body; please see the updated version of the paper on OpenReview. To further incorporate this suggestion, we will utilize the extra page in the camera ready version to include additional  technical details.
>
>     &nbsp;
>
> - **[W2]**: *Our choice of Nash Social Welfare is not driven solely by the fact that the geometric mean sits between the arithmetic mean and the minimum value*. Rather, a key reason behind choosing Nash social welfare as the objective is that it is the only welfare function that **satisfies fundamental fairness axioms**, including Pigou-Dalton principle, Pareto optimality, independence of unconcerned agents, symmetry, and scale-freeness. We mention these points in the introduction (lines 088-090 of current OpenReview version and lines 090-092 of the original version), but we will highlight them more clearly to avoid any misunderstanding. We thank the reviewer for the feedback.

---

> ### Author Response · Authors · 2025-11-18
> **Response for Reviewer jBed (Part II - [W3])**
>
> - **[W3]**: We thank the reviewer again for appreciating the formulation.
>
>     &nbsp;
>
>     The reviewer is correct that our formulation requires the presence of attributes. Indeed, our motivation was to study a framework similar to [AIK+25] where attributes are explicitly available and diversity among attributes is a desideratum. *The framework captures numerous important applications, e.g., recommendation systems and online advertising*. For such applications, our work provides a well-founded formulation that strikes a balance between relevance and diversity across attributes.
>
>     &nbsp;
>
>     The alternatives suggested here in the review (and similar ones studied in [AAYIM13] and [AAYI+13]) do not explicitly address diversity across attributes, which is the focus of the current work. One can adopt [AAYIM13] and [AAYI+13] to attribute diversity, but even such extensions have inherent limitations, as detailed next.
>
>     &nbsp;
>
>     In particular, the notion of diversity studied in [AAYIM13] and [AAYI+13] requires an additional distance function (and not specifically attributes) to express diversity. In particular, the objective in their work is to return, from among the set of vectors that are at most $r$ distance away from the query, a subset of $k$ vectors that maximizes the minimum pairwise diversity distance.
>
>     &nbsp;
>
>     One way to adapt this distance-based diversity notion to the single-attribute setting is by considering Hamming distances. Specifically, any two vectors with the different attributes are set to have a diversity distance of $1$ and vectors with the same attribute are set at $0$ distance. *The key issue with this formulation is that the solutions it finds will always be at the extremes*: if the prefetched pool consists of more than $k$ different attributes, then the subset picked by this approach will consist of one vector from each attribute (i.e., it is the same as the one via the hard-constraint formulation with $k’=1$). Otherwise, if the prefetched pool consists of less than $k$ different attributes, then any size-$k$ subset will have minimum diversity distance $0$. Hence, in this case, the selected subset can be arbitrary; say, the $k$ most similar vectors.
>
>     &nbsp;
>
>     **That is, the notion from [AAYIM13] and [AAYI+13] (with Hamming distance as the diversity function) does not provide a comparison point stronger than the one (Div-ANN with different values of $k’$) already considered in the submission.**
>
>     &nbsp;
>
>     That said, in view of this feedback, we empirically compared our formulation with the alternative notion. The results (given in Appendix F.6 of the current OpenReview version and **tables below**) validate the above-mentioned observation that the Div-ANN (with different values of $k’$) is a more apt benchmark. We will include this discussion in the final version.
>
>     &nbsp;
>
>     In addition to not being attribute-centric, another concern with the notion from [AAYIM13] and [AAYI+13] is that (unlike NSW) this diversity definition does not originate from a normative basis; hence, it is unclear what fairness guarantees are satisfied by such a formulation.
>
>     &nbsp;
>
>
>     We also note that, in the multi-attribute setting, the Hamming distance based diversity distance function is quite ad hoc. This restricts the use of the distance-based diversity notion in multi-attribute settings.
>
>     &nbsp;
>
>     Although when aiming for a broader treatment of diversity, we agree with the reviewer that extending the NSW formulation to settings where attributes are not explicit is an important direction for future work. We note this in the Conclusion Section (Line 485).
>
>
> &nbsp;
> &nbsp;
>
> **Experimental Results**
>
> |      |    **Amazon Dataset (k=50)**    |     |
> |--------------------------|---------------|----------------------|
> | **Method ↓** | **Approx-Ratio ↑**| **Entropy ↑** |
> | Nash-ANN  | 0.938 ± 0.023 | 5.058 ± 0.178 |
> | ANN | 1.000 ± 0.000 | 2.782 ± 0.684 |
> | PairDiv-ANN [AAYIM13, AAYI+13] | 0.999 ± 0.008  | 2.801 ± 0.725 |
> | Div-ANN | 0.813 ± 0.053   | 5.594 ± 0.049 |
>
>
> | |   **SIFT1m (Prob) (k=10)**  | |
> |------------------------|--------------------|---------------------------|
> | **Method ↓**  |  **Approx-Ratio ↑** | **Entropy ↑**  |
> | Nash-ANN | 0.988 ± 0.032  | 3.3196 ± 0.223 |
> | ANN | 1.000 ± 0.008        | 1.7217 ± 0.315 |
> | PairDiv-ANN [AAYIM13, AAYI+13] |  0.896 ± 0.046 | 3.3219 ± 0.000 |
> | Div-ANN |  0.878 ± 0.047 | 3.3219 ± 0.000 |
>
> &nbsp;
> &nbsp;
>
> **References**
>
> [AIK+25] Anand, Piyush,  Piotr Indyk, Ravishankar Krishnaswamy, Sepideh Mahabadi, Vikas C. Raykar, Kirankumar Shiragur, Haike Xu. “Graph-Based Algorithms for Diverse Similarity Search.” ICML 2025.
>
> [AAYIM13] Abbar, Sofiane, Sihem Amer-Yahia, Piotr Indyk, and Sepideh Mahabadi. "Real-time recommendation of diverse related articles." WWW 2013.
>
>
> [AAYI+13] Abbar, Sofiane, Sihem Amer-Yahia, Piotr Indyk, Sepideh Mahabadi, and Kasturi R. Varadarajan. "Diverse near neighbor problem." SoCG 2013.

---

### Official Review · Reviewer_2kxy · 2025-11-01

**Soundness:** 2
**Presentation:** 1
**Contribution:** 2
**Rating:** 2
**Confidence:** 3

**Summary:**

This paper introduces a framework for incorporating diversity into Nearest Neighbor Search (NNS) by drawing on welfare theory from mathematical economics. Traditional NNS methods mainly focus on maximizing relevance, measured by similarity between a query and data points, while in more recent approaches, diversity requirements are specified using constraints. The main difference of the approach in the paper is that welfare requirements are proposed using welfare-based objectives grounded in Nash Social Welfare. More specifically, attributes of data points (such as color, brand, or seller) are modeled as “agents” in an economic system, where each agent’s utility depends on how well its associated attribute is represented among the selected neighbors. The collective welfare of the result set is then measured using a welfare function, such as the geometric mean in the case of NSW. Empirical evaluations on real and synthetic datasets provide evidence that the approach may have practical applications.

**Strengths:**

The main strength is that the chosen approach enforces fairness (diversity across attributes) without requiring ad hoc parameters or fixed quotas, and it adapts to the intent expressed in each query—for example, selecting more homogeneous results when the query is specific and more diverse ones when it is broad.

**Weaknesses:**

The main weakness of the paper is the presentation. The introduction is way too long. Key justifications of correctness are totally relegated to the appendix (all proofs of theorems). Additionally, currently very basic information such as the very formal definition of diversity being used by the authors is not explicitly highlighted. My recommendation is to

1) Shorten the introduction significantly to end at page 2. An introduction is usually expected to provide a brief summary of the results, a brief discussion of the main applications and appropriate connections with related work.

2) Provide formal definitions in such a way that the newly introduced concepts are highlighted. This makes the paper much easier to read because it clearly separates your original contribution from contributions from the literature. Since you are proposing a new way of quantifying diversity I would expect something like "Below we define our main notion of diversity..." or something similar and then a clear statement "Definition X (Diversity Mesure)".

With saved space, move some key insights necessary to prove the theorems to the main body of the paper. Currently all arguments are relegated to the appendix.

In summary, In my opinion, the current organisation of the paper is suboptimal and makes the evaluation of the contributions of the paper more difficult than necessary.

**Questions:**

No questions.

---

> ### Author Response · Authors · 2025-11-18
> **Response for Reviewer 2kxy**
>
> Here, we address the concern regarding “the introduction is way too long.”
>
> &nbsp;
>
> - **[1]**: The work makes novel conceptual contributions by bridging two distinct areas: nearest neighbor search and welfare economics. Hence, the intention behind the longer-than-usual introduction is to articulate the reasoning behind the formulation, rather than merely stating it (as in Definition 1). Such an exposition emphasizes the importance of defining objectives properly and intentionally goes beyond just tersely stating the formulation and results. Therefore, we believe the detailed introduction is a strength of our paper – this has been acknowledged by Reviewer jBed as well.
>
>     &nbsp;
>
>     Still, in view of this editorial feedback, we have abridged the introduction by removing repetitions, wherever possible. Further, for easy access, we have itemized and sign-posted the contributions at the end of the introduction. We have also moved proof of one of our main results (Theorem 1) into the main body. Please see the updated version of the paper on OpenReview with all of these changes.
>
>     &nbsp;
>
>     We will utilize the extra page in the camera ready version to include additional technical details.
>
>
>     &nbsp;
>
> - **[2]:**  Our formulation for diversity is succinctly expressed in Definition 1 (NashNNS). The updated version on OpenReview lists all the contributions at the end of the introduction; in particular, we point to Definition 1 here.  We note these contributions were mentioned at the end of the introduction in the earlier submitted version, albeit not in a bulleted manner.
>
>
>     &nbsp;
>
> - **[3]:** We have included the proof of Theorem 1 into the main body. As mentioned above, we will utilize the extra page in the camera ready version to include additional technical details.

---

### Official Review · Reviewer_nv7E · 2025-11-04

**Soundness:** 3
**Presentation:** 4
**Contribution:** 3
**Rating:** 8
**Confidence:** 4

**Summary:**

The paper investigates variants of ANNs with new constraints on Nash social welfare measures (NSW) and discusses single-attribute and multi-attribute settings. The problems are verified to be intractable -- and efficient algorithms with provable approximation ratios are provided. Benchmark tests have verified the advantages of the proposed algorithms in efficiency and quality.

**Strengths:**

S1. It's an interesting perspective to consider NSW for ANNs for fairness and diversity measures.
S2. Problems and Algorithms are justified with hardness analysis, matching provable guarantees, and cost analysis.
S3. Solutions with generality on oracles and ANN algorithms have been experimentally verified.

**Weaknesses:**

W1. The impact of correlated or contradictory utilities may warrant discussion.

W2. The discussion on connecting the solution to machine learning/representation learning could be discussed. Examples of other ML issues, or real-world scenarios that may benefit from the proposed algorithms, can be provided and tested.

**Questions:**

D1. The utilities may have a positive or negative correlation. Will it be ensured that a single optimal solution always exists under the NSW setting? How will the problem compare with other options for multi-criteria queries that seek Pareto-optimality, with dominance but not necessarily optimizing a single value?

D2. Consider providing examples of learning problems that could benefit from algorithmic and theoretical analysis.

D3. NN queries have been extensively studied;  how the proposed solution may be improved with existing techniques such as indexing, sampling, etc, for NaNNS and p-NNS would also be interesting.

---

> ### Author Response · Authors · 2025-11-18
> **Response for Reviewer nv7E**
>
> We thank the reviewer for appreciating the work. Next, we address the questions raised in the review.
>
> &nbsp;
>
> - **[D1]:** In our setting, the utilities are obtained via nonnegative similarities and, hence, are always nonnegative.
> Indeed, the NSW optimal solution might not be unique. However, multiplicity of optimal solutions does not impact the axiomatic guarantees and any of our results.
>
>     &nbsp;
>
>     We agree that Pareto optimality is important for multi-criteria queries. **It is relevant to note that the NSW (and $p$-mean) optimal solutions are always Pareto optimal in terms of the utility profiles**, and, as stated, NSW additionally satisfies many other fairness axioms such as the Pigou–Dalton principle, scale-freeness, symmetry, and independence of unconcerned agents.
>
>     &nbsp;
>
> - **[D2]**: *Any learning problem that utilizes NNS and wherein diversity is a desideratum can benefit from this work,* for example, retrieval-augmented generation (RAG) and recommendation systems.  We provide such examples in the introduction (Lines 034-039) and can add a few more such applications in the final version.
>
>     &nbsp;
>
> - **[D3]**: As mentioned in Corollary 2, Algorithm 1 for NaNNS and Algorithm 3 for p-NNS can leverage any existing ANN algorithm as a blackbox without incurring any additional loss in approximation over the underlying ANN method. That is, if the approximation ratio of the (black box) ANN method is $\alpha$, then we obtain an $\alpha$-approximate solution for the optimal Nash welfare. *Therefore, our algorithms can benefit from any future improvements in ANN methods.*
>
>     &nbsp;
>
>     In addition, the proposed heuristic (which first retrieves a sufficiently large set of similar vectors and then applies Algorithms 1 or 3 on them) also provides a direct way to build on any existing ANN pipeline. Our experiments show that this heuristic is comparable with the provable algorithm in terms of relevance and diversity, and, at the same time, it provides about a $10$ times increase in throughput and a similar decrease in latency; see Appendix F.5 for further details.

---

### Official Review · Reviewer_gr82 · 2025-11-08

**Soundness:** 3
**Presentation:** 3
**Contribution:** 3
**Rating:** 6
**Confidence:** 3

**Summary:**

This paper proposes a novel approach to nearest neighbor search (NNS) that balances relevance and diversity. It introduces welfare-based formulations based on Nash social welfare and p-means, which provide an adaptive, query-based balance between similarity and diversity. The effectiveness of these formulations is evaluated through experiments on several datasets, showing improved diversity while maintaining high relevance. The proposed algorithms, Nash-ANN and p-Mean-ANN, provide efficient solutions with provable guarantees.

**Strengths:**

**S1:** The paper provides a theoretical foundation by leveraging welfare functions from mathematical economics, particularly Nash social welfare, to address the challenge of diversity in NNS.

**S2:** The paper provides practical and efficient algorithms for solving the proposed welfare-based NNS problems.

**S3:** The paper's approach offers flexibility and adaptability by allowing the trade-off between relevance and diversity to be controlled through the parameter p in the p-mean welfare function. This allows practitioners to tailor the search results to specific task requirements, making the proposed methods versatile for different applications and user preferences.

**Weaknesses:**

**W1:** The proposed algorithms, both single-attribute and multi-attribute settings, are simple greedy-based algorithms. Although they are easy to implement and provide theoretical guarantees, they require multiple passes of linear scans over the dataset and thus become inefficient on large-scale datasets. Therefore, improving the efficiency of the proposed algorithms using ANN index structures is a critical issue.

**W2:** The proposed query formulation relies on parameter tuning, particularly the smoothing parameter $\eta$ and the exponent parameter $p$. Finding appropriate values for these parameters for specific queries can be challenging and may require extensive experimentation, which could be a practical limitation when used in practice.

**W3:** The paper primarily compares its methods against a hard-constrained approach and a standard NNS method. In the literature, many other formulations and methods for diversity-aware (as well as fairness-aware) NNS have been proposed. Further comparison with them is necessary to validate the effectiveness of the proposed formulation.

**Questions:**

See weaknesses.

---

> ### Author Response · Authors · 2025-11-18
> **Response for Review gr82  (Part I - [W1], [W2])**
>
> We thank the reviewer for their time and feedback. Below, we will address the points **([W1], [W2])** raised in the review and highlight the changes we have made in the OpenReview submission to incorporate the suggestions. **Kindly refer to the next comment for [W3]**.
>
> - **[W1]**: In the single-attribute case, **our algorithm does not necessarily require linear scans over the data**. As mentioned in Corollary 2, Algorithm 1 can be executed using any existing ANN indexing method as a black box. Here, Algorithm 1 requires as input $k$ vectors from each attribute class and we do not incur any additional loss in approximation over the underlying ANN method. That is, if the approximation ratio of the (black box) ANN method is $\alpha$, then we obtain an $\alpha$-approximate solution for the optimal NSW.
>
>     &nbsp;
>
>     For the multi-attribute case, Algorithm 2 (with provable guarantees) does require scans. However, it is possible to modify Algorithm 2 and leverage any sublinear-time ANN method: (i) For every attribute subset $L \subseteq [c]$, create an ANN data structure over vectors in $D_L := \{v \in P: \text{atb}(v) = L\}$ (note that $D_L$s are disjoint), and (ii) During the search for any given query, first retrieve top $k$ approximately most similar vectors from each $D_L$, using the ANN data structure, and then run Algorithm 2 on this collection of vectors. One can show that this modified algorithm has provable guarantees similar to Theorem 4.  Also, note that when the number, $U$, of distinct attribute tuples associated with the input vectors is small compared to $n$, this modified algorithm has a much better runtime of $O(k U)$.
>
>     &nbsp;
>
>     For addressing large-scale data sets, we do provide and empirically validate a heuristic that can be implemented on top of any existing ANN pipeline; see the Multi-Nash ANN description immediately preceding Section 4.1 and Appendix F.4. The heuristic is reminiscent of the above-mentioned modification of Algorithm 2 in that it prefetches a sufficiently large number of candidates, thereby avoiding repeated scans over the entire dataset.
>
>     &nbsp;
>
>     Finally, as stated in Section 5, an interesting direction for future work is to move beyond treating ANN as a black box and instead develop an indexing data structure tailored specifically for Nash ANN.
>
> &nbsp;
>
>
> - **[W2]**: We clarify that **our formulation does not rely on tuning the value of $p$**. We propose the Nash Nearest Neighbor Search (NaNNS) problem (Definition 1) as the objective that strikes a balance between relevance and diversity. However, practitioners might prefer finer control over the tradeoff between relevance and diversity. It is for such targeted tradeoffs that we generalize NaNNS to $p$-mean Nearest Neighbor Search. By setting the value of $p \in (-\infty, 1]$, a practitioner can derive a smooth tradeoff between the two extremes: complete relevance ($p=1$) and complete diversity ($p \to - \infty$). Note that $p$ is not required to be tuned on datasets but rather provided as an input, whose value can be selected by the practitioner based on the desired level of diversity, which is standard practice. We focus on the NaNNS problem ($p \to 0$), given that Nash social welfare satisfies relevant fairness and efficiency axioms. Still, the choice of $p$ provides a functional control, and we leave this choice to the practitioner for their use case.
>
>     &nbsp;
>
>     The smoothening constant $\eta$ serves to ensure non-zero values of the Nash social welfare. In fact, if $c \ll k$, in the single-attribute setting, $\eta$ can be ignored. Moreover, setting the $\eta$ value is straightforward; a small value on the order of the minimum distance between any pair of points in the dataset typically suffices. Our theoretical guarantees are *agnostic to the choice of $\eta$*, and our experiments suggest that the results are robust to a range of values of $\eta$. More details on the values of $\eta$ selected for our experiments can be found at the end of Appendix F.2.

---

> ### Author Response · Authors · 2025-11-18
> **Response for Review gr82 (Part II -  [W3])**
>
> - **[W3]**: The current work studies diversity across specified attributes. Our comparisons focus on the hard-constrained approach since it is the essential prior work that also addresses attribute diversity. In NNS, we find two additional threads of work on diversity and fairness. *However, as detailed below, the notions in these threads do not directly address attributes.*
>
>     &nbsp;
>
>     In particular, the notion studied in [HPM19, APS20, AHPM+21] is that of individual fairness: given a parameter $r$, every vector within a radius of $r$ of the query should have equal probability of being included in the returned set. Note that this definition is oblivious to any attribute information. In particular, if all the input vectors within a distance $r$ of the query have the same attribute, then this fairness notion fails to incorporate any diversity in terms of the attribute. Hence, this notion is not well-suited when studying diversity across attributes.
>
>     &nbsp;
>
>     The diversity notion developed in [AAYIM13, AAYI+13] requires an additional distance function (and not specifically attributes) to express diversity. The objective in [AAYIM13] and [AAYI+13] is to return, from among the set of vectors that are at most $r$ distance away from the query, a subset of $k$ vectors that maximizes the minimum pairwise diversity distance.
>
>     &nbsp;
>
>     One way to adapt the distance-based diversity notion to the single-attribute setting is by considering Hamming distances. Specifically, any two vectors with the different attributes are set to have a diversity distance of $1$ and vectors with the same attribute are set at $0$ distance. The key issue with this formulation is that the solutions it finds will always be at the extremes: if the prefetched pool consists of more than $k$ different attributes, then the subset picked by this approach will consist of one vector from each attribute (i.e., it is the same as the one via the hard-constraint formulation with $k’=1$). Otherwise, if the prefetched pool consists of less than $k$ different attributes, then any size-$k$ subset will have minimum diversity distance $0$. Hence, in this case, the selected subset can be arbitrary; say, the $k$ most similar vectors.
>
>
>     &nbsp;
>
>    **This observation underscores that the notion from [AAYIM13] and [AAYI+13] (with Hamming distance as the diversity function) does not provide a comparison point stronger than the one (Div-ANN with different values of $k’$) already considered in the submission.**
>
>     &nbsp;
>
>     Still, to incorporate the reviewer’s feedback, we empirically compared our formulation with this alternative notion. The results (given in Appendix F.6 of the current OpenReview version and **tables below**) validate the above-mentioned observation that the Div-ANN (with different values of $k’$) is a more apt benchmark. We will include this discussion in the final version.
>
>     &nbsp;
>
>     In addition to not being attribute-centric, another concern with the notion from [AAYIM13] and [AAYI+13] is that (unlike NSW) this diversity definition does not originate from a normative basis; hence, it is unclear what fairness guarantees are satisfied by such a formulation.
>
>     &nbsp;
>
>     We also note that, in the multi-attribute setting, the Hamming distance based diversity distance function is quite ad hoc. This restricts the use of the distance-based diversity notion in multi-attribute settings.
>
>     &nbsp;
>
>     We will include these discussions in the final version.
>
>
>     &nbsp;
>
> **Experimental Results**
>
> |             |    **Amazon Dataset (k=50)**    |     |
> |--------------------------|-----------------------|---------------------------------|
> | **Method ↓**       | **Approx-Ratio ↑**| **Entropy ↑**   |
> | Nash-ANN        | 0.938 ± 0.023            | 5.058 ± 0.178 |
> | ANN           | 1.000 ± 0.000            | 2.782 ± 0.684 |
> | PairDiv-ANN [AAYIM13, AAYI+13] | 0.999 ± 0.008  | 2.801 ± 0.725  |
> | Div-ANN        | 0.813 ± 0.053   | 5.594 ± 0.049   |
>
>
> |            |        **SIFT1m (Prob) (k=10)**  |   |
> |--------------------------|-----------------------|---------------------------------|
> | **Method ↓**             |  **Approx-Ratio ↑** | **Entropy ↑**   |
> | Nash-ANN        | 0.988 ± 0.032        | 3.3196 ± 0.223 |
> | ANN              | 1.000 ± 0.008        | 1.7217 ± 0.315 |
> | PairDiv-ANN [AAYIM13, AAYI+13] |  0.896 ± 0.046        | 3.3219 ± 0.000 |
> | Div-ANN           |  0.878 ± 0.047        | 3.3219 ± 0.000 |
>
> &nbsp;
> &nbsp;
>
> **References**
>
> [HPM19] Har-Peled et al. Near neighbor: Who is the fairest of them all?  NeurIPS 2019.
>
> [AHPM+21] Aumüller et al. Sampling a near neighbor in high dimensions—who is the fairest of them all?  ACM TODS 2022.
>
> [APS20] Aumüller et al. Fair near neighbor search: Independent range sampling in high dimensions. PODS 2020.
>
> [AAYIM13] Abbar et al. Real-time recommendation of diverse related articles.  WWW 2013.
>
> [AAYI+13] Abbar et al. Diverse near neighbor problem.  SoCG 2013.

---

> ### Comment · Reviewer_gr82 · 2025-11-19
>
> My concerns are mostly addressed by the rebuttal, and I will raise my score to 8.

---

> > ### Author Response · Authors · 2025-11-19
> >
> > We thank the reviewer for taking the time to go through the rebuttal and for increasing the score. We appreciate that the reviewer found the responses relevant. The reviewer's discussion has certainly helped us enhance our work.

---

### Meta-Review · Area_Chair_8q8x · 2025-12-26

**Summary:**

- The intro is too dense
- The proposed algorithm may need to scan data multiple times, which hurts efficacy.

**Reviewer Concerns:**

- The authors have reduced the length of the introduction and added more theoretical proofs to the main body.
- The authors have added a heuristic approach to improve the runtime of Nash-ANN.

**Reviewer Scores:**

Reviewer gr82 may increase its score since the rebuttal heavily addresses its concern.

---

### Decision · Program_Chairs · 2026-01-26

Accept (Poster)